# A memory model of rodent spatial navigation in which place cells are memories arranged in a grid and grid cells are non-spatial

**David E Huber\***

Department of Psychology and Neuroscience, University of Colorado Boulder, Boulder, United States

## eLife Assessment

This **important** paper provides **solid** evidence for an alternative conceptualization of the functional role of the place and grid cell network in the medial temporal lobe for memory as opposed to spatial processing or navigation. The theory is extensive, tightly integrating data on various spatial cell types. It accounts for many experimental results and generates strong predictions for future studies that will be of interest to researchers in this field. The impact of the work would be strengthened if future experiments reveal that grid cells do indeed encode specific nonspatial features.

**\*For correspondence:**
david.huber@colorado.edu

**Competing interest:** The author declares that no competing interests exist.

**Abstract** A theory and neurocomputational model are presented that explain grid cell responses as the byproduct of equally dissimilar hippocampal memories. On this account, place and grid cells are best understood as the natural consequence of memory encoding and retrieval; a precise hexagonal grid is the exception rather than the rule, emerging when the animal explores a large surface that is devoid of landmarks and objects. In the proposed memory model, place cells represent memories that are conjunctions of both spatial and non-spatial attributes, and grid cells primarily represent the non-spatial attributes (e.g. sounds, surface texture, etc.) found throughout the two-dimensional recording enclosure. Place cells support memories of the locations where non-spatial attributes can be found (e.g. positions with a particular sound), which are arranged in a hexagonal lattice owing to memory encoding and consolidation processes (pattern separation) as applied to situations in which the non-spatial attributes are found at all locations of a two-dimensional surface. Grid cells exhibit their spatial firing pattern owing to feedback from hippocampal place cells (i.e. a hexagonal pattern of remembered locations for the non-spatial attribute represented by a grid cell). Model simulations explain a wide variety of results in the rodent spatial navigation literature.

## Introduction

According to the human neuropsychological literature and primate studies, the primary function of the medial temporal lobe (MTL) is the encoding and retrieval of episodic memories (*Corkin, 1984*; *Squire and Zola-Morgan, 1991*). However, when studying rodent behavior, the MTL is often associated with spatial navigation (*Moser et al., 2008*; *O'Keefe, 1976*). There are some studies reporting spatial navigation cells in human MTL (*Doeller et al., 2010*; e.g. *Jacobs et al., 2013*) and some studies reporting non-spatial cell types in rodent MTL (e.g. *Aronov et al., 2017*), but these findings are exceptions to the general trends in the literature. It is possible that these general trends are accurate, with the MTL playing different roles in different species, as might be the case if primitive spatial networks were

coopted for representing concepts and memories in humans (*Buzsáki and Moser, 2013*; *Milivojevic and Doeller, 2013*). Alternatively, these general trends may reflect researchers' use of different behavioral tasks for each species out of convenience – it is difficult to ask rodents to recall events, and it is typically not possible to study humans wandering in large open spaces while recording brain activity. However, if the MTL is fundamentally a memory structure designed to remember *what* happened and *where* it happened, it may be that different behavioral tasks appear to identify different functions for the MTL because they focus on either the *what* or the *where* of the situation, rather than the conjunction of these attributes. Supporting this view, there is ample evidence that spatial location is a powerful episodic retrieval cue (*Godden and Baddeley, 1975*; *Roediger, 1980*).

Before presenting this theory in greater detail, I give an example in which a memory conjunction can produce a place cell response, depending on how one analyzes the situation. On the day that I turned 18, I was driving to my parents' home in eastern Massachusetts. Without warning, the front-right ball joint broke and the car tipped onto the road surface. Miraculously, the car safely drifted to a stop in the breakdown lane and I was unharmed. This is a vivid memory of a specific conjunction of attributes, including the exact place on Route 2 where the accident occurred. In the decades since, I have thought of this episode at various times. Of particular relevance to this memory account of hippocampal place cells, I *always* think of this episode when passing that location when traveling along Route 2. If you recorded from one of the hippocampal cells that represents this memory, that cell would appear to be a place cell for a specific location along Route 2. Furthermore, if you put an electrode in one of my cells that represents the concept of car accident, it would systematically respond at that location as well as at other locations of notable accidents (e.g. not one place, but multiple places, similar to a grid cell).

The proposal that hippocampus represents the multimodal conjunctions that define an episode is not new (*Marr et al., 1991*; *Sutherland and Rudy, 1989*) and neither is the proposal that hippocampal memory supports spatial/navigation ability (*Eichenbaum, 2017*). This view of the hippocampus is consistent with 'feature in place' results (*O'Keefe and Krupic, 2021*) in which hippocampal cells respond to the conjunction of a non-spatial attribute affixed to a specific location, rather than responding more generically to any instance of a non-spatial attribute. In other words, the what/where conjunction is unique. Furthermore, the uniqueness of the what/where conjunction may be the fundamental building block of spatial memory and navigation. In reviewing the hippocampal literature, (*Eichenbaum, 2017*) concludes that 'the hippocampal system is not dedicated to spatial cognition and navigation, but organizes experiences in memory, for which spatial mapping and navigation are both a metaphor for and a prominent application of relational memory organization'. However, this memory conjunction view of the MTL must be reconciled with the rodent electrophysiology finding that *most* cells in MTL appear to have receptive fields related to some aspect of spatial navigation (*Boccara et al., 2010*; *Diehl et al., 2017*; *Grieves and Jeffery, 2017*). In brief, if the majority of the cells in the rodent hippocampus and medial entorhinal cortex (mEC) are place cells (*O'Keefe, 1976*), grid cells (*Hafting et al., 2005*), head direction cells (*Taube et al., 1990*), conjunctive grid cells that are also sensitive to head direction (*Sargolini et al., 2006*), border/boundary cells (*Lever et al., 2009*; *Solstad et al., 2008*), and object-vector cells (*Høydal et al., 2019*), how can it be that the MTL is primarily a memory structure?

One possible answer to the apparent lack of non-spatial cells in MTL is to highlight the role of the lateral entorhinal cortex (LEC) as the source of non-spatial *what* information for memory encoding (*Deshmukh and Knierim, 2011*). LEC can be contrasted with mEC, which appears to only provide *where* information (*Boccara et al., 2010*; *Diehl et al., 2017*). Although it is generally true that LEC is involved in non-spatial processing, there is evidence that LEC provides some forms of spatial information (*Knierim et al., 2014*). The kind of non-spatial information provided by LEC appears to be in relation to objects (*Connor and Knierim, 2017*; *Wilson et al., 2013*). However, in a typical rodent spatial navigation study there are no objects within the enclosure. Thus, although the distinction between mEC and LEC is likely part of the explanation, it is still the case that rodent entorhinal input to hippocampus *appears* to heavily favor spatial information.

The paucity of non-spatial cells in rodent MTL could be explained if grid cells have been mischaracterized as spatial. I propose a memory account of the rodent spatial navigation literature in which each grid cell represents some non-spatial attribute (e.g. a particular sound) and place cells mark where the non-spatial attribute can be found (e.g. memory of locations where that sound can be

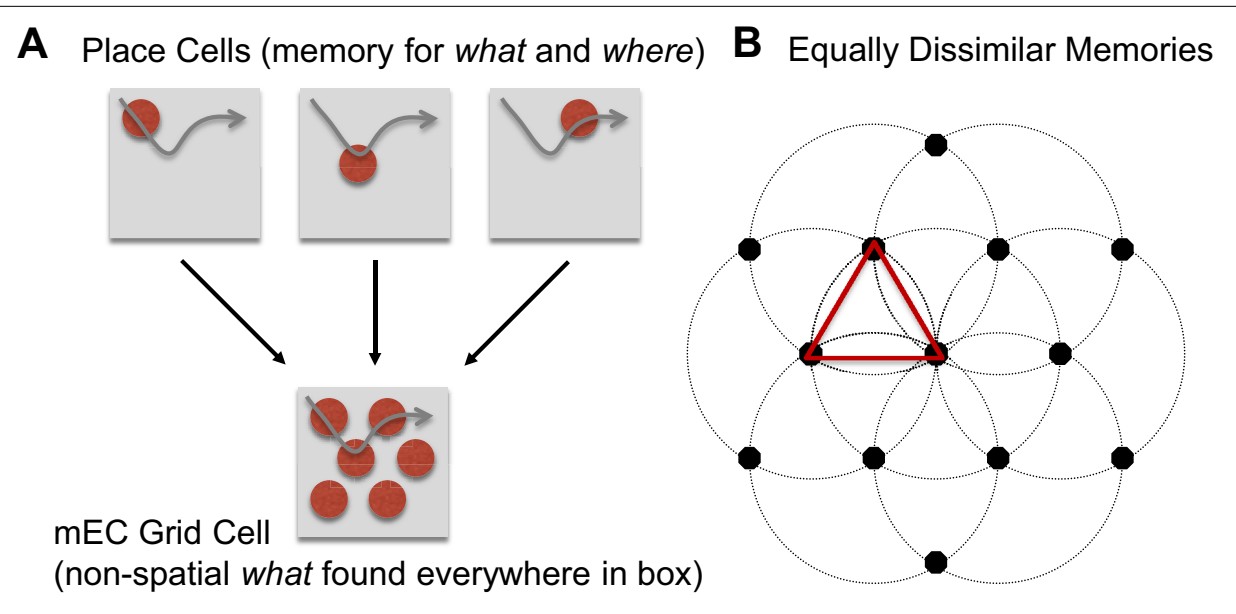

**Figure 1.** Proposed relationship between grid cells and place cells. Each place cell captures the conjunction of *what* happened and *where* it happened.
(**A**) When revisiting a position, the memory associated with that position is retrieved, providing feedback to the non-spatial *what* cells (e.g. a 60 Hz electronic noise), which reside in medial entorhinal cortex (mEC). Because the non-spatial attribute is common to all positions (e.g. the electronic noise occurs everywhere), and because various mechanisms ensure that cells avoid firing constantly at their maximum possible rate during the entire recording session, grid cells fire preferentially in the locations of each memory. (**B**) If memories are formed whenever the current situation is sufficiently different from prior situations, this results in a hexagonally arranged grid of place cells that tile the two-dimensional surface because the only thing that varies in a typical navigation experiment is X/Y location; each memory is formed whenever prior memories are sufficiently far away (the circles represent a fixed dissimilarity between memories).

heard). Because the non-spatial properties are not manipulated in a typical rodent navigation study, I give them the generic label K, and future studies are needed to determine the bottom-up non-spatial receptive fields that specify the non-spatial attributes K (e.g. see *Aronov et al., 2017* for an example in which some grid cells were also found to be sound-frequency cells). On this account, each place cell represents the conjunction of one or more non-spatial attributes and a specific location where those attributes can be found (which might be everywhere within the enclosure), and each grid cells represent the presence of a particular non-spatial attribute.

In naturalistic situations, different attributes (i.e. different K) are positioned in specific places in the environment (e.g. a particular location where a certain kind of food is reliably found). However, rodent navigation studies often use homogenous environments in which the non-spatial attributes are found everywhere. In this case, any attribute found in the environment (e.g. the 60 Hz hum of electronics), will be one that is found everywhere in the enclosure. If the non-spatial attribute is found everywhere in the enclosure, the memories of where that attribute can be found will uniformly tile the space of the enclosure. Furthermore, grid cells representing that attribute will be more strongly active in the place field centers of the memories that tile the space, owing to feedback from place cells – that is, there is a place cell that is the cause of each separate grid field, and the corresponding place cell becomes active owing to memory retrieval of the properties associated with that place.

In a typical rodent grid cell experiment, the *what* of the situation includes many possible factors (i.e. many K) found throughout a two-dimensional navigation surface – for example, sound, lighting, temperature, surface texture, etc. – as well as many episodic factors that are relatively constant during the recording session – for example, morning/evening, human experimenter, state of satiety, etc. Given the sameness of the non-spatial factors during the recording session, the only thing that differentiates one memory from another is *where* each memory occurred. This reduces the array of memories to a two-dimensional plane, resulting in a discrete grid of memories, with each place cell capturing a different memory in a different location, and each grid cell modulating its activity as the animal traverses the enclosure, with each location triggering an associated memory.

*Figure 1* shows the core hypothesis of this proposal, which assumes that the bottom-up receptive fields of grid cells in medial entorhinal cortex (mEC) are non-spatial attributes found throughout the two-dimensional surface. For instance, the animal experiences the same 60 Hz electronic noise at all positions in the enclosure. Because the sound occurs everywhere in the box, the mEC cell that detects that sound fires preferentially at the positions where the cell also receives feedback from place cells representing associated memories (*Figure 1A*). More specifically, an mEC grid cell experiences constant bottom-up excitatory input combined with top-down memory feedback that is stronger in particular remembered locations. Rather than firing at a constant high rate regardless of location, the cell adopts an adjusted firing threshold owing to inhibitory interneurons and divisive normalization (*Bhatia et al., 2019*; *Carandini and Heeger, 2011*; *Olsen et al., 2010*) such that the cell primarily modulates its activity as a function of top-down memory feedback (because the attribute is found everywhere, the only source of variability in the cell's response is the magnitude of memory feedback). The memories providing this feedback are encoded by hippocampal place cells, which reflect the conjunction of *what* happened (the non-spatial attribute of a particular grid cell) and *where* it happened (one position versus another position in the box). As the animal navigates (gray curved arrow), the current location cues prior memories that occurred in that location, which might be memories that were created just seconds or minutes ago ('here I am again in the middle of the box, which still has that same sound'). Thus, the finding that the mEC contains a high percentage of grid cells (*Boccara et al., 2010*) might be an artifact of an experimental paradigm in which nothing of interest varies aside from position.

Memory encoding is more likely to occur in novel situations (*Tulving et al., 1996*) whereas retrieval occurs in familiar situations (*Howard et al., 2005*), where 'familiar' includes situations that might have occurred just a few seconds ago. Combining these memory principles leads to a regularly spaced array of remembered positions (i.e. equally dissimilar memories) that is created 'on-the-fly' when navigating a novel two-dimensional plane that is unvarying in terms of its non-spatial attributes (*Figure 1B*). When first positioned in the enclosure, the animal creates a memory of that position conjoined with the non-spatial attributes of the recording session. This might be in the form of separate memories for each attribute K (e.g. particular sound), or it might be a complex multidimensional memory (e.g. the combination of sound, surface texture, etc.). Once the animal wanders sufficiently far from its initial position, the spatial attributes of the current situation mismatch recently encoded memories and the animal encodes a new memory. This memory formation process continues until the animal has fully explored the box. When not forming new memories, the animal retrieves recently encoded memories – in other words, the animal is either creating a memory of the non-spatial attributes associated with a new position or remembering what exists at recently visited positions, with the current position cueing memory retrieval. Because the non-spatial attributes are constant throughout the two-dimensional surface, this results in an array of discrete memory locations that is approximately hexagonal (as explained in the Model Methods, an 'online' memory consolidation process employing pattern separation rapidly turns an approximately hexagonal array into one that is precisely hexagonal). Because this array of memories is created on-the-fly, and because there is constant feedback from newly created memories, the hexagonal layout of memories will appear to exist instantly upon entry into the enclosure, as if it were a pre-existing representation designed to aid navigation. In summary, the grid array of memories is rapidly created as the animal explores a novel environment that is devoid of landmarks.

The foregoing account explains the core theory of this proposal. Providing an overview of the findings and model predictions, *Box 1* shows a list of results that are explained by this memory model as well as other models, a list of results that are uniquely explained by this model, and a list of predictions made by this model. Items on this list will be addressed in the model methods, simulation results, and discussion.

## Why model the rodent navigation literature with a memory model?

Spatial navigation is inherently a memory problem – learning the spatial arrangement of a new enclosure requires memory for the conjunction of *what* and *where*. This has long been realized and in the introduction to 'Hippocampus as a Cognitive Map', O'Keefe and Nadel wrote "We shall argue that the hippocampus is the core of a neural *memory system* providing an objective spatial framework within which the items and events of an organism's experience are located and interrelated" (emphasis added; *O'Keefe and Nadel, 1978*). Furthermore, in the last chapter of their book, they extended

## Box 1.

**Results explained by this memory model and some other models (no single other model explains all of these)**

- grid fields can be centered outside the box
- the population code of grid cells lies on a torus
- grid is aligned with the walls of the box
- the immediate existence of the grid pattern
- the slower learning of place fields
- dependency of grid cells on hippocampal feedback
- existence of grid cell modules
- hexagonal grid patterns for non-spatial representations
- grid pattern is disrupted in narrow passages (see Appendix)
- remapping of place cells (see Appendix)
- position dependency of place cell head direction sensitivity (see Appendix)

**Results uniquely explained by this memory model**

- some grid cells become head direction cells w/o hippocampal feedback
- grid cells that are also sensitive to sound frequency
- place cell head direction sensitivity increases in narrow passages (see Appendix)

**Unique predictions of this memory model, yet to be tested**

- sets of place cells (those representing the same properties) are arrayed in a grid
- all grid cells have non-spatial bottom-up receptive fields
- all grid cells are conjunctive with one or more non-spatial property
- all grid cells revert to bottom-up receptive field in the absence of hippocampal feedback
- place cells centered at the same location have complementary head direction sensitivity
- stabilization of place fields depends on learning the borders of the enclosure
- separate regions containing different non-spatial properties should produce separate grids
- memories rapidly consolidate to become neither too similar nor too dissimilar

cognitive map theory to human memory for non-spatial characteristics. However, in the decades since the development of cognitive map theory, the rodent spatial navigation and human memory literatures have progressed somewhat independently.

The ideas proposed in this model are an attempt to reunify these literatures by returning to the original claim that spatial navigation is inherently a memory problem. The goal of the current study is to explain the rodent spatial navigation literature using a memory model that has the potential to also explain the human memory literature. In contrast, most grid cell models (*Burak and Fiete, 2009*; *Bush et al., 2015*; *Castro and Aguiar, 2014*; *Couey et al., 2013*; *Fuhs and Touretzky, 2006*; *Guanella et al., 2007*; *Hasselmo, 2009*; *McNaughton et al., 2006*; *Mhatre et al., 2012*; *Solstad et al., 2006*; *Sorscher et al., 2023*; *Stepanyuk, 2015*; *Widloski and Fiete, 2014*) are domain-specific models of spatial navigation and as such, they do not lend themselves to explanations of human memory. Thus, the reason to prefer this model is parsimony. Rather than needing to develop a theory of memory that is separate from a theory of spatial navigation, it might be possible to address both literatures with a unified account.

This study does not attempt to falsify other theories of grid cells. Instead, this model reaches a radically different interpretation regarding the function of grid cells; an interpretation that emerges from viewing spatial navigation as a memory problem. All other grid cell models assume that an entorhinal

grid cell displaying a spatially arranged grid of firing fields serves the function of spatial coding (i.e. spatial grid cells exist to support a spatial metric). In contrast, the proposed memory model of grid cells assumes that the hexagonal tiling reflects the need to keep memories separate from each other to minimize confusion and confabulation – the grid pattern is the byproduct of pattern separation between memories rather than the basis of a spatial code.

It is now understood that grid-like firing fields can occur for non-spatial two-dimensional spaces. For instance, human entorhinal cortex exhibits grid-like responses to video morph trajectories in a two-dimensional bird neck-length versus bird leg-length space (*Constantinescu et al., 2016*). As a general theory of learning and memory, the proposed memory model of grid cells is easily extended to explain these results (e.g. relabeling the border cell inputs in the model as neck-length and leg-length inputs). However, there are other grid cell models that can explain both spatial grid cells as well as non-spatial grid-like responses (*Mok and Love, 2019*; *Rodríguez-Domínguez and Caplan, 2019*; *Stachenfeld et al., 2017*; *Wei et al., 2015*). Similar to this memory model of grid cells, these models are also positioned to explain both the rodent spatial navigation and human memory literatures. Nevertheless, there is a key difference between this model and other grid cell models that generalize to non-spatial representations. Specifically, these other models assume that grid cells exhibiting spatial receptive fields serve the function of identifying positions in the environment (i.e. their function is spatial). As such, these models do not explain why most of the input to rodent hippocampus appears to be spatial (*Boccara et al., 2010*; *Diehl et al., 2017*; *Grieves and Jeffery, 2017*). This memory model of grid cells provides an answer to the apparent paucity of non-spatial cell types in rodent MTL by proposing that grid cells with spatial receptive fields have been misclassified as spatial (they are *what* cells rather than *where* cells) and that place cells are fundamentally memory cells that conjoin *what* and *where*.

## The scope of the proposed model

The reported simulations explain why most mEC cell types in the rodent literature appear to be spatial (*Boccara et al., 2010*; *Diehl et al., 2017*; *Grieves and Jeffery, 2017*). Assuming that rodents can form non-spatial memories, rodent hippocampus must receive non-spatial input from entorhinal cortex. These simulations suggest that characterization of the rodent mEC cortex as primarily spatial might be incorrect if most grid cells (except perhaps head direction conjunctive grid cells) have been mischaracterized as spatial. Other literatures with other species find non-spatial representations in MTL (*Gulli et al., 2020*; *Quiroga et al., 2005*; *Wixted et al., 2014*) and non-spatial hippocampal memory encoding has been found in rodents (*Liu et al., 2012*; *McEchron and Disterhoft, 1999*). The proposed memory model is compatible with these results – the ideas contained in this model could be applied to non-spatial memory representations. However, surveys of cell types in rodent entorhinal cortex seem to indicate that most cells are spatial (*Boccara et al., 2010*; *Diehl et al., 2017*; *Grieves and Jeffery, 2017*). How can the rodent hippocampus encode non-spatial memories if most of its input is spatial? The goal of the reported simulations is to explain the apparent paucity of non-spatial cells in rodent entorhinal cortex by proposing that grid cells have been misclassified as spatial (see also *Luo et al., 2024*).

Given the simplicity of the proposed model, there are important findings that the model cannot address—it is not that the model makes the wrong predictions but rather that it makes no predictions. The role of running speed (*Kraus et al., 2015*) is one such variable for which the model makes no predictions. Similarly, because the model is a rate-coded model rather than a model of oscillating spiking neurons, it makes no predictions regarding theta oscillations (*Buzsáki and Moser, 2013*). The model is an account of learning and memory for an adult animal, and it makes no predictions regarding the developmental (*Langston et al., 2010*; *Muessig et al., 2015*; *Wills et al., 2012*) or evolutionary (*Rodríguez et al., 2002*) time course of different cell types. This model contains several purely spatial representations such as border cells, head direction cells, and head direction conjunctive grid cells and it may be that these purely spatial cell types emerged first, followed by the evolution and/or development of non-spatial cell types. However, this does not invalidate the model. Instead, this is a model for an adult animal that has both episodic memory capabilities and spatial navigation capabilities, irrespective of the order in which these capabilities emerged.

This model has the potential to explain context effects in memory (*Godden and Baddeley, 1975*; *Gulli et al., 2020*; *Howard et al., 2005*). According to this model, different grid cells represent

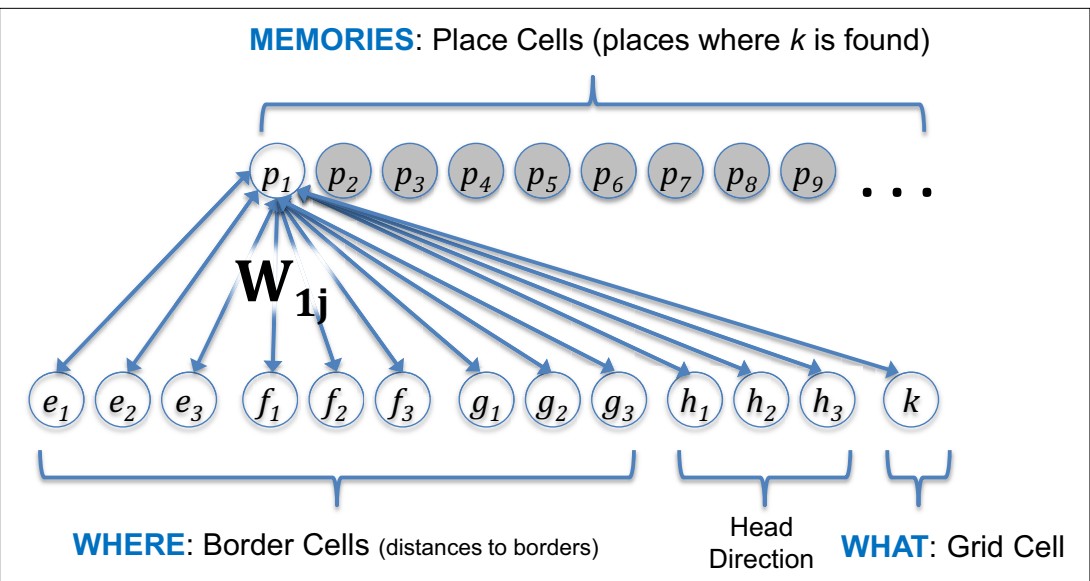

**Figure 2.** Assumed connectivity between place cells in hippocampus and border cells, head direction cells, and a grid cell in medial entorhinal cortex. When a memory is formed, a place cell (e.g. **p**₁) is recruited and the weights between the entorhinal inputs and the place cells are set equal to the inputs (e.g. $W_{1j}$ for weights to $p_1$), with $j$ ranging over the 13 entorhinal inputs to hippocampus. These weights are bidirectional, with feedback supporting memory recall. Because feedback modulates the response of the grid cell, this produces a higher firing rate at positions where the non-spatial attribute is remembered. Code for the model can be found at: https://github.com/dhuber1968/GridCellMemoryModel (copy archived at Zenodo: *Huber, 2025*).

different non-spatial characteristics and place cells represent the combination of these 'context' factors and location. In the simulation, just one grid cell is simulated but the same results would emerge when simulating hundreds of different non-spatial inputs provided that all of the simulated non-spatial inputs exist throughout the recording session. However, there is evidence that hippocampus can explicitly represent the passage of time (*Eichenbaum, 2014*), and time is assuredly an important factor in defining episodic memory (*Bright et al., 2020*). Thus, although the current model addresses unique combinations of *what* and *where*, it is left to future work to incorporate representations of *when* in the memory model.

## Model

In developing a model implementation of the theory outlined in the introduction, auxiliary assumptions (e.g. the shape of neural tuning functions, the manner in which X/Y locations are perceived, the mechanism for memory consolidation, etc.) are needed to address the findings listed in *Box 1*.

### Combinations of border cells specify position

The model is a bidirectional two-layer network (*Figure 2*) in which border cells (*Lever et al., 2009*; *Solstad et al., 2008*) – a.k.a., boundary cells – are the primary spatial input to hippocampus that determines place coding. This assumption is made because border cells provide an efficient representation of Euclidean space (e.g. if the animal knows how far it is from different walls of the enclosure, this already available information can be used to calculate location). Three border cell dimensions capture values along three allocentric directions of the two-dimensional surface. These position responses are combined with other entorhinal inputs (head direction and non-spatial grid cells) to form a multidimensional memory of where the non-spatial attributes occurred and which way the animal was oriented at the time of memory formation. When a new memory is formed, a hippocampal place cell (or more plausibly a population of cells) is recruited and the weights connecting the place cell to the entorhinal inputs are set equal to the current input values (*Grossberg and Grossberg, 1982*). This assumption is in keeping with the 'hippocampal indexing theory' of episodic memory (*Teyler and Rudy, 2007*) and is a natural outcome of Hebbian learning (*Hebb, 2005*).

These assumptions regarding border cells are based on the boundary vector cell (BVC) model of *Barry et al., 2006*. As in the BVC model, combinations of border cells encode where each memory occurred in the real-world X/Y plane. For the model shown in *Figure 2*, this population code of border cells is additionally combined with the response of head direction cells (*Taube et al., 1990*) to indicate where the animal was positioned and in which direction the animal was looking/headed. Thus, when including head direction, there is a three-dimensional space in which memories may vary: X, Y, and head direction.

Similar to the BVC model, each border cell receptive field is defined relative to a particular border, with a preferred allocentric distance to that border (e.g. 10 cm East of a border). However, unlike the BVC model, not all border cells are assumed to be parallel to a border. Instead, it is assumed that the border cells are pre-arranged into three non-orthogonal dimensions, tipped 60 degrees apart (as explained below, this arrangement provides a basis set that naturally calculates Euclidean distances between current position and remembered positions). When the animal first enters a square enclosure, one of the three dimensions aligns with the most salient wall, with the other two dimensions tipped at an angle relative to the borders.

Unlike the BVC model, the boundary cell representation is sparsely populated using a basis set of three cells for each of the three dimensions (i.e. nine cells in total), such that for each of the three non-orthogonal orientations, one cell captures one border, another the opposite border, and the third cell captures positions between the opposing borders (*Solstad et al., 2008*). However, this is not a core assumption, and it is possible to configure the model with border cell configurations that contain two opponent border cells per dimension, without needing to assume that any cells prefer positions between the borders (with the current parameters, the model predicts there will be two border cells for each between-border cell). Similarly, it is possible to configure the model with more than three cells for each dimension (i.e. multiple cells representing positions between the borders). To keep the model simple, the same number of cells was assumed for all dimensions and all dimensions were assumed to be circular (head direction is necessarily circular and because one dimension needed to be circular, all dimensions were assumed to be circular). Three cells per dimensions was chosen because this provides a sparse population code of each dimension, with few border cells responding between borders, while allowing three separate phases of grid cells within a grid cell module (in the model, a grid cell module arises from combination of a third dimension, such as head direction, with the real-world X/Y dimensions defined by border cells).

The assumed representational space is allocentric in the sense of aligning with the most salient characteristic of either the interior of the enclosure (e.g. a straight wall), the exterior to the enclosure (e.g. an external cue card), or some combination of the interior and exterior (e.g. an external cue card by virtue of its alignment with a straight wall). As such, the model can explain the finding that border cell representations rotate with rotation of the enclosure (but stay fixed relative to the experimental room) in the case of a circular enclosure that lacks any salient characteristics (*Hafting et al., 2005*). At the same time, the model can explain the finding that border cell representations do not rotate for a square environment, unless the rotation is 90, 180, or 270 degrees such that a different straight wall aligns with a salient aspect of the exterior (*Krupic et al., 2015*; *Savelli et al., 2017*).

## Circular basis set for each dimension

In the model, basis sets of three neural tuning functions (i.e. three cells, each with a neural tuning function centered on a different preferred input) provide an unbiased representation of each dimension (i.e. one basis set for each dimension). An unbiased representation is defined as one that can capture any position along the dimension with equivalent precision. For instance, consider the dimension of color, with the three types of color photoreceptors providing a basis set to identify all visible hues for human color vision. However, color discrimination/precision is not unbiased as a function of hue because the preferred wavelength of the short-wavelength photoreceptors is somewhat separated from the other two preferred wavelengths, which lie closer together (*Solomon and Lennie, 2007*). In contrast to the biased representation of color, which is constrained by the photochemical properties of photoreceptors, the model assumes that the preferred positions of different border cells (or head direction cells) in the basis set are placed at regularly spaced intervals.

Because memory formation sets the weights equal to entorhinal input responses, the important metric for assessing bias is the sum of squares of the entorhinal input. More specifically, memory

retrieval strength is equal to the sum of squares of the input when revisiting a circumstance that matches the circumstance of memory formation (i.e. the input response values will be equal to the weight values in such a circumstance). If this sum of squares of the entorhinal input is not constant with location, then certain pre-defined positions would have the capacity to produce stronger memories (e.g. a larger sum of squares) as compared to other pre-defined positions. Such a situation would bias memory retrieval for some locations as compared to other locations, regardless of prior experience. Thus, beyond equal spacing of preferred stimuli, an unbiased representation also requires a particular shape for the tuning curve (e.g. how quickly firing rate falls off with dissimilarity from preferring input) to provide a constant sum of squares. This is achieved by assuming a sine wave shape for the tuning curves.

Because head direction is necessarily a circular dimension, it was assumed that all dimensions are circular (a circular dimension is approximately linear for nearby locations). This assumption of circular dimensions was made to keep the model relatively simple, making it easier to combine dimensions and allowing application of the same processes for all dimensions. For instance, the model requires a weight normalization process to ensure that the pattern of weights for each dimension corresponds to a possible input value along that dimension. However, the normalization for a linear dimension is necessarily different than for a circular dimension. Because the neural tuning functions were assumed to be sine waves, normalization requires that the sum of squared weights add up to a constant value. For a linear dimension, this sum of squares rule only applies to the subset of cells that are relevant to a particular value along the dimension whereas for a circular dimension, this sum of squares rule is over the entire set of cells that represent the dimension (i.e. weight normalization is easier to implement with circular dimensions). Although all dimensions were assumed to be circular for reasons of mathematical convenience and parsimony, circular dimensions may relate to the finding that human observers have difficultly re-orienting themselves in a room depending on the degree of rotational symmetry of the room (*Kelly et al., 2008*). In addition, similar to other models (*Guanella et al., 2007*; *McNaughton et al., 2006*), this simplifying assumption allows the model to capture the finding that the population of grid cells lies on a torus (*Gardner et al., 2022*).

Each basis set that represents a dimension contains three equally spaced sine wave neural tuning functions. *Equation 1* is the neural response, $r$, of neuron, $i$, at dimension value, $d$ ($-1 < d < +1$). The preferred value, $p_i$, for the tuning function produces the largest response. The three preferred values for the three simulated neurons in the basis set are set to $-2/3$, 0, and $+2/3$, which span the range from $-1$ to $+1$. The value of 1 is added to the cosine so that the neural tuning functions are purely positive.

$$r_i\left(d\right) = \frac{\sqrt{2}}{3}\left\{1 + cosine\left[\pi\left(d - p_i\right)\right]\right\} \tag{1}$$

Setting the constant equal to the square root of 2 divided by 3 ensures that the sum of squares of the three sine waves is 1 across all values of the dimension (see *Figure 3A* for an example of the three sine waves). Furthermore, the sum of the three sine waves is a constant square root of 2, providing a constant level of input activity across the entorhinal inputs.

## Memory retrieval strength is proportional to Euclidean distance with three non-orthogonal dimensions

Animals can plan novel straight line paths to reach a known position and evidence suggests they do so by learning Euclidean representations of space (*Cheng and Gallistel, 2014*; *Normand and Boesch, 2009*; *Wilkie, 1989*). Thus, it was assumed that hippocampal place cells represent positions in Euclidean space (as opposed to non-Euclidean space, such a occurs with a city-block metric). To achieve Euclidean representations, the reported simulations used three non-orthogonal *E/F/G* circular dimensions (*Figure 3C*), which define a space that is a hexagonally connected three-dimensional torus. This can be compared to the hippocampal place cell representations that arise from a two-dimensional torus defined by orthogonal *X/Y* dimensions (*Figure 3A*). With three non-orthogonal dimensions, the similarity between memories is approximately Euclidean (*Figure 3D*). In contrast, with two orthogonal dimensions, the similarity between memories follows a city-block metric. As seen in *Figure 3B*, larger Euclidean distances along the x-axis produce multiple memory retrieval strength values along the y-axis when using two orthogonal dimensions.

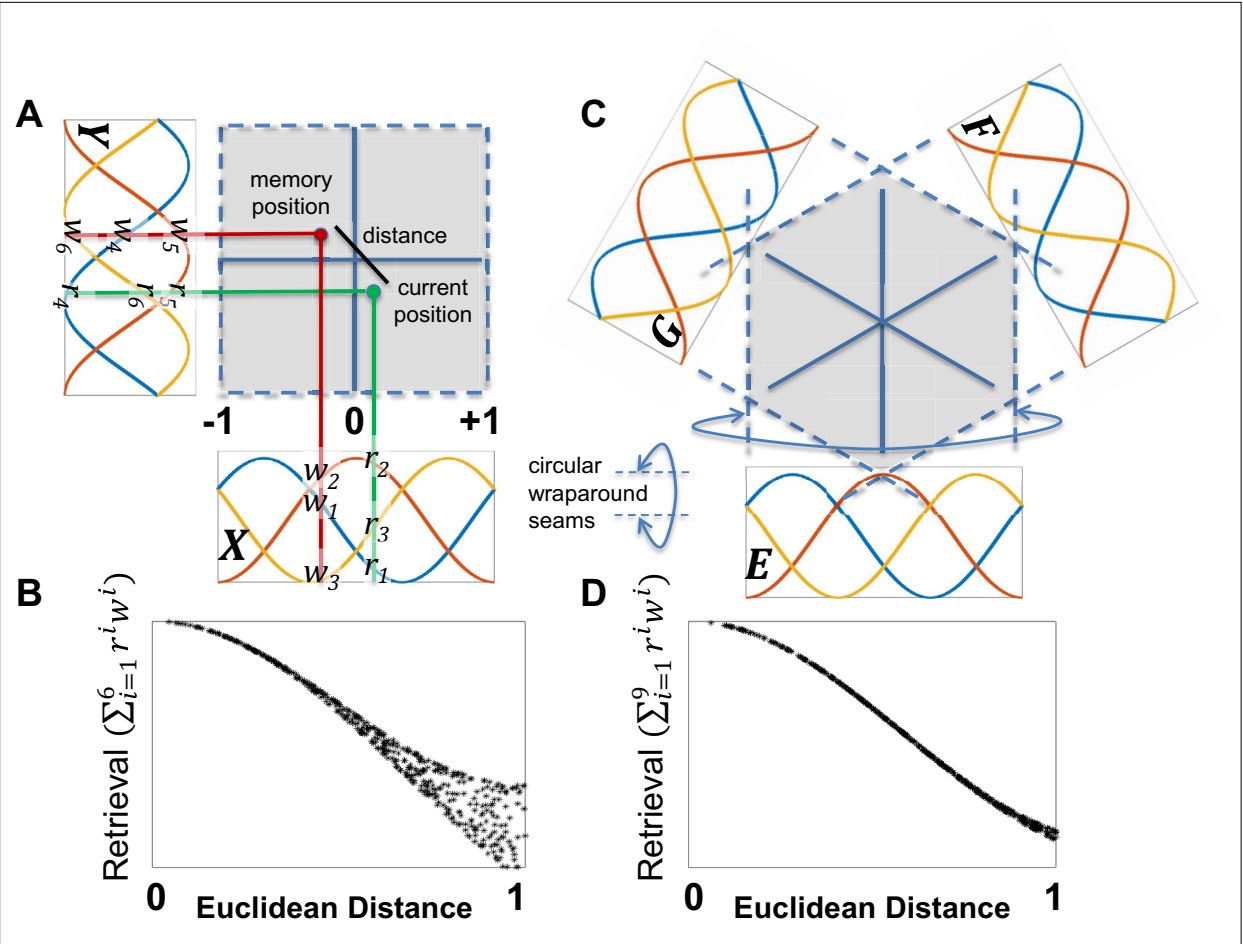

**Figure 3.** Comparison between memory retrieval with two orthogonal dimensions (*X/Y*) versus three non-orthogonal dimensions (*E/F/G*) that are 60 degrees apart. Each dimension is represented by a circular basis set with three equally spaced sine waves with a period of 2. When a place cell is learned, the weights connecting each sine wave input and the place cells are set equal to the input values. (**A**) In the case of orthogonal *X/Y* dimensions, this results in a pattern of 6 weight values ($w_1 - w_6$) across the two dimensions, as shown by the intersection of the red lines emanating from the position where the memory was formed (the red dot) and the three sine waves for each dimension. After memory formation, the current position (green dot) reactivates the memory based on the 6 current position response values ($r_1 - r_6$), summing the multiplication of the response values and the weight values. (**B**) The graph shows the result of randomly sampling 1000 different memory positions and retrieval positions, plotting retrieval strength as a function of Euclidean distance for each pair of positions. Retrieval strength is variable with Euclidean distance because the sum across the two orthogonal dimensions is a city-block metric (e.g.,the same Euclidean distance can map onto multiple city-block distances). (**C**) To capture Euclidean distances, three non-orthogonal dimensions (*E/F/G*) were used. (**D**) This produces a retrieval function that is approximately monotonic with Euclidean distance.

## Memory encoding and retrieval produces an approximate grid

Model simulations start the animal at a random location in the enclosure and at each time step, the animal adopts a new random goal direction as compared to the last time step. The simulated animal moves toward the new goal direction, but momentum dictates continuity across time steps. This results in a random curved path that eventually visits all positions with all head directions. Of note, the ability of the model to produce grid cell responses does not depend on this decision to simulate an animal taking a random walk – the same results emerge if the animal is more systematic in its path. All that matters for producing grid cell responses is that the animal visits all locations and that the animal takes on different head directions for the same location in the case of simulations that also include head direction as an input to hippocampal place cells. An example path with 1000 timesteps is shown in *Figure 4C*. A simulated recording session involves 10,000 timesteps. Using the 'Grid Score' measure developed by *Solstad et al., 2008*, even the first 1000 time steps produce a fairly accurate hexagonal grid of memory positions. The Grid Score measure compares the 2D spatial autocorrelation

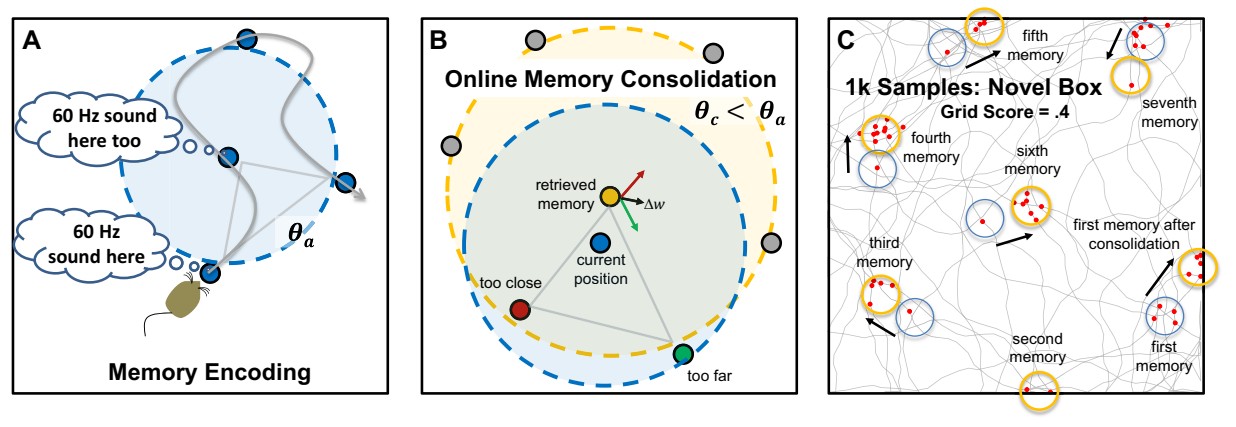

**Figure 4.** Memory encoding, memory consolidation, and an example sequence. (**A**) When first entering a novel environment, the animal creates a memory of the non-spatial attributes of that environment (e.g. the 60 Hz sound of electronics) at each location where the attribute is found. The gray curved arrow shows the random path taken by the simulated animal and the blue dots show the positions where memories are created. The activation threshold, $\theta_a$ (blue dashed circle), dictates whether previously created memories are retrieved, or if none are retrieved, a new memory is formed. This produces a minimum distance between memories. (**B**) The representations of memories are altered by an online consolidation process that produces unbiased memory representations that tile the environment (i.e. a cognitive map). In this process, the most strongly active memory (the yellow dot retrieved memory) is slightly altered in relation to competing memories that are also activated by the current position (the red and green dots). Other memories (gray dots) remain inactive because they are too dissimilar to the current position (outside the blue dashed circle centered on current position). After initial memory retrieval, the retrieved inputs are used to activate the competing memories and this strength of activation of competitors is compared to a consolidation threshold, $\theta_c$ (yellow dashed circle, centered on retrieved memory), which is smaller than the activation threshold, such that consolidation pushes memories to become maximally dissimilar (pattern separation). Competing memories that are more active than the consolidation threshold (red dot) push the weights of the retrieved memory away from the competing memory (red arrow). Competing memories that are less active than the consolidation threshold (green dot) pull the weights of the retrieved memory towards the competing memory (green arrow). For memories arranged in two real-world dimensions, this typically results in activation of three surrounding memories and consolidation makes the triangle formed by these memories an equilateral triangle (gray arrow). (**C**) An example path with 1000 simulated steps is shown, with the blue circles indicating the initial positions of memories and the yellow circles indicating memory positions after consolidation. The red dots show positions where the simulated grid cell fired. The firing threshold was set such that the cell fires 5% of the time, resulting in 50 positions where the grid cell fired. *Figure 4— video 1* shows the simulation in (**C**), including memories that flash yellow, red, and green as outlined in (**B**).

The online version of this article includes the following video for figure 4:

**Figure 4—video 1.** Memory formation and consolidation.

https://elifesciences.org/articles/95733/figures#fig4video1

of the spike rate map to one that is rotated by 60 or 120 degrees (sixfold symmetry for a hexagonal grid) versus one that is rotated 30, 90, or 150 degrees (fourfold symmetry for a square grid). Thus, a grid that is regular but square rather than hexagonal produces a negative Grid Scores. A Grid Score of zero can occur either if the spatial autocorrelation fails to exhibit any systematic grid or if the grid is halfway between square and hexagonal.

*Equation 2* is the activation, *act(p_i)*, of memory, *i*, represented by place cell $p_i$, based on input responses from the 12 entorhinal cells, $r_j$, that represent the current position (*E/F/G*) and head direction (*H*), and the previously learned weights, $w_{ij}$, between each input and the place cell. This equation is divided by 4 because there are four dimensions of input (*E/F/G/H*), resulting in a situation where memory activation is 1.0 if all weights exactly equal the input responses (i.e. perfect déjà vu). For simulations that do not include head direction, the summation in the numerator would be over nine weights (three dimensions *E/F/G*) and the constant in the denominator would be 3. The activation values for all place cells are then compared to the activation threshold, $\theta_a$, and if none is above threshold, a new memory is formed by recruiting a new place cell, setting $w_{ij} = r_j$ for all weights connecting inputs to the new place cell.

$$act\left(p_i\right) = \frac{\sum_{j=1}^{12} w_{ij} r_j}{4} \tag{2}$$

This mandates a minimum distance between memories at the time of memory formation (*Figure 4A*). In theory, the activation equation should also include the response of the grid cell, *k*, and the learned weight between grid cell and the place cell. However, because the non-spatial attribute is constant (i.e. the attribute is found everywhere on the surface), its inclusion would only shift all responses by the same constant and can be omitted without changing behavior of the model (i.e. if all activations are increased by a constant, but the activation threshold is also increased by the same constant, then memory formation and retrieval would be unchanged).

## Forming a cognitive map: online memory consolidation regularizes the grid

The encoding assumptions of the model produce a minimum spacing between memories. However, the initial lattice of remembered locations will be somewhat irregular (see *Ginosar et al., 2021* for an example of grid fields arranged irregularly, but with a minimum distance spacing), with some locations more densely surrounded by memories than others as dictated by the path taken by the animal during initial exploration. Similar to the assumption regarding unbiased basis sets of entorhinal inputs, it is advantageous to regularize the remembered locations because doing so provides an unbiased 'cognitive map' (*O'Keefe and Nadel, 1979*) that can be used to represent any location with equal precision. This regularization of the memory array could occur offline, as in traditional theories of systems consolidation (*O'Reilly and McClelland, 1994*). However, the proposed model assumes an online consolidation process that achieves regularization rapidly during initial exploration. Nevertheless, if the proposed consolidation process is a basic mechanism of learning, it will also occur during offline replay (*Ólafsdóttir et al., 2018*). The goal of the proposed consolidation learning is to ensure that every real-world position is surrounded by an array of memories that are neither too similar nor too dissimilar from each other; if all real-world positions are surrounded by a regular array of remembered locations, this provides a cognitive map by virtue of 'triangulation' (i.e. all real-world positions are surrounded by an equilateral triangle of positions with known attributes).

A summary of the consolidation process is provided before considering additional details. First, a set of memories is identified that surround the current real-world situation. Then, the most strongly active of the surrounding memories is adjusted in relation to the other surrounding memories in such a manner as to position the strongly active memory neither too far nor too close to the other surrounding memories. In the case of a real-world space in two-dimensions, the consolidation algorithm can be thought of as ensuring that all positions are surrounded by an equilateral triangle (aka, a 'regular simplex'), providing good triangulation to represent knowledge for the attributes of the environment. In the case of three dimensions, which occurs when including head direction, the algorithm ensures that each position is surrounded by a regular tetrahedron of surrounding memories. More abstractly, this consolidation process can be thought of as providing a 'covering map' of surrounding exemplars (*Kruschke, 1992*). Critically, even positions near the box borders obtain a surrounding array of memories, which occurs naturally during consolidation as some hippocampal place cells are pushed 'outside the box'. This occurs even though the animal is never given the opportunity to visit these positions. In other words, false memories for positions outside the box contribute to a cognitive map that supports knowledge for all positions within the box.

Online consolidation quickly repositions memories through nudges to the weights of each previously encoded memory (*Figure 4B*). The consolidation process uses what could be described as top-down delta-rule learning (*Rescorla and Wagner, 1972*; *Widrow and Hoff, 1960*): each competing memory that surround a retrieved memory provides a teaching signal to modify the weights of the retrieved memory. Unlike traditional delta-rule learning, which has a fixed learning rate, the learning rate sign and magnitude are modulated according to the BCM rule (*Bienenstock et al., 1982*), depending on similarity of each competing memory to the retrieved memory. Because the consolidation threshold, $\theta_c$, is smaller than the activation threshold that dictates initial memory formation, memories primarily push away from each other during the earliest stages of exploration, but after further exploration there is an equal mix of pushing and pulling once the enclosure has been fully explored. In other words, during early exploration, there is nothing known about what lies on the other side of each memory, and so there is room to push apart memories. However, once something is known about all positions in the enclosure, the memories are constrained on all sides.

In real systems, the proposed consolidation process might arise from several cycles of alternating bottom-up memory activation and top-down memory recall as coordinated by theta oscillations (see *Ólafsdóttir et al., 2016* for evidence that something similar might occur during offline consolidation). First, the entorhinal input corresponding to the real-world situation reminds the animal of the most similar memory (the retrieved memory), as well as a set of partially active competing memories. The retrieved memory re-activates the associated entorhinal inputs owing to top-down memory recall (pattern completion). For instance, a particular X/Y position reminds the animal of some non-spatial attribute that was encountered in a nearby location, and memory retrieval recalls the exact spatial position of that attribute by changing the entorhinal inputs from their real-world values to the recalled values. After this initial top-down memory retrieval, there is a second bottom-up cycle that is now based on the recalled entorhinal inputs. This second bottom-up cycle is used to identify the similarity of the competing memories in relation to the retrieved memory by assessing how strongly the competing memories become active based on the entorhinal inputs of the retrieved memory (critically, the designation of memories as belonging to the set of competing memories is based on the first bottom-up cycle that used real-world inputs such that the consolidation process is in relation to a set of memories that surround the current real-world situation). If the competing memories then re-activate their associated entorhinal inputs (i.e. a second top-down wave, or perhaps through additional cycles if the competing memories are assessed one at a time), this would provide the necessary entorhinal response values for the consolidation learning rule (more specifically, consolidation learning is based on the patterns of entorhinal inputs that correspond to the retrieved memory as well as each of the competing memories).

*Equation 3* specifies how the weights, $w_{RMj}$, connecting the entorhinal input, $j$, to the Retrieved Memory, $RM$, are changed by summing over the separate weight changes specified by each of the, $n$, Competing Memories, $CM$. There are typically three competing memories for the case of navigating in a space with real-world variation along three-dimensions (i.e. $X$, $Y$, and head direction $H$), corresponding to a surrounding tetrahedron when including the retrieved memory (a surround of four memories). The term inside the parentheses is the delta-rule for weight changes, which is the difference between the weight connecting the input, $j$, to the *retrieved* memory and the weight connecting the input, $j$, to the *competing* memory. This difference is then multiplied by the BCM learning rate for competing memory, $\alpha_{cm}$, based on how similar the competing memory is the retrieved memory (*Equation 4*). Specifically, the weight of the retrieved memory becomes more similar to the weight of the competing memory if $\alpha_{cm}$ is negative but more dissimilar if $\alpha_{cm}$ is positive. The learning rate is calculated by comparing how strongly the inputs corresponding to the retrieved memory (which are the weights of the retrieved memory) activate the competing memory place cell, $act(p_{CM} \mid w_{RM})$, as compared to the consolidation threshold, $\theta_c$. If these retrieved inputs activate the competing memory too strongly ($> \theta_c$), the competing memory is too similar (e.g. too close) and the retrieved memory's weights become dissimilar to the competing memory (pattern separation). If these retrieved inputs activate the competing memory too weakly ($< \theta_c$), the competing memory is too dissimilar (e.g. too far), and the retrieved memory's weights become more similar to the competing memory. If these inputs activate the competing memory to a value equal to the threshold, the competing memory is the desired dissimilarity from retrieved memory and there are no changes to the weights of the retrieved memory.

$$\Delta w_{RMj} = \sum_{CM=1}^{n} \alpha_{CM} \left( w_{RMj} - w_{CMj} \right) \tag{3}$$

$$\alpha_{CM} = act \left( p_{CM} | w_{RM} \right) - \theta_c \tag{4}$$

Code for model simulations can be found at https://github.com/dhuber1968/GridCellMemory-Model (copy archived at Zenodo: *Huber, 2025*) and *Figure 4—video 1* shows the first 1000 time steps for a simulation that does not include head direction (memories in a X/Y space), revealing the rapid formation of grid fields. A second movie highlights the slower learning of memories that vary in three dimensions (X, Y, and head direction), showing that although the memories roughly arrange in a hexagonal grid within the first 2000 time steps, the place cell memories continue to shift for tens of thousands of additional time steps, as revealed by the next 98,000 time steps run at high-speed (*Figure 9—video 1*). Pseudocode for each simulated time step is as follows:

1. Move to a new position according to momentum from the prior movement that is partially altered according to a new randomly sampled goal direction
2. Based on the new position and new head direction (*Equation 1*), activate memories (*Equation 2*) and compare to the activation threshold $\theta_a$
3. If no memories are above threshold, create a new memory by recruiting a place cell, setting memory weights equal to the current entorhinal input.
4. If just one memory is above threshold, do nothing (memory retrieval occurred, but no consolidation occurs)
5. If more than one memory is above threshold, note which memory is most active (the *retrieved memory*) and note which other memories are also activated in response to the current inputs (the *competing memories*). Consolidate the retrieved memory in terms of its similarity to the competing memories (*Equations 3; 4*), using the consolidation threshold $\theta_c$ to determine whether weight changes are towards or away from each competing memory
6. If consolidation occurred, normalize any updated weight values of the retrieved memory to ensure that they correspond to possible real-world inputs

In real systems, the final step of weight normalization might occur through post-learning feedback down to early perceptual regions of the cortex followed by a cycle back up to MTL, with a comparison between this (feedback) 'minus phase' versus (feedforward) 'plus phase' (*Hinton and Sejnowski, 1986*; *O'Reilly, 1996*). Alternatively, weight normalization might occur through a form of divisive normalization (*Carandini and Heeger, 2011*). In this case, the normalization ensures that the sum of squared weight values for each basis set is 1.0 and that the sum of weight values for each basis set is the square root of 2, as necessitated by the assumed basis set sine wave tuning functions. Rather than specifying how this normalization process operates for real neurons, the simulation takes a mathematical shortcut by using the arcsine function to recover the dimensional inputs that are closest to the updated weight values.

## Habituation, rate coding, and firing thresholds

In theory, a cell representing a non-spatial attribute found at all locations of an enclosure (aka, a grid cell in the context of this model), could fire constantly within the enclosure. However, in practice, cells habituate and rapidly reduce their firing rate by an order of magnitude when their preferred stimulus is presented without cessation (*Abbott et al., 1997*; *Tsodyks and Markram, 1997*). After habituation, the firing rate of the cell fluctuates with minor variation in the strength of the excitatory drive. In other words, habituation allows the cell to become sensitive to changes in the excitatory drive (*Huber and O'Reilly, 2003*). Thus, if there is stronger top-down memory feedback in some locations as compared to others, the cell will fire at a higher rate in those remembered locations rather than in all locations even though the attribute is found at all locations. In brief when faced with constant excitatory drive, the cell accommodates, and becomes sensitive to change in the magnitude of the excitatory drive. In the model simulation, this dynamic adaptation is captured by supposing that cells fire 5% of the time on-average across the simulation, regardless of their excitatory inputs.

The model does not directly implement spiking neurons. Instead, the simulated activation values can be thought of as proportional to the average firing rate of an ensemble of neurons with similar inputs and outputs (*O'Reilly and Munakata, 2000*). For instance, consider a set of several thousand spiking grid cells that are identical in terms of their firing fields. At any moment, some of these identically behaving cells will produce an action potential while others do not (i.e. the cells are not perfectly synchronized), but a snapshot of their behavior can be extracted by calculating average firing rate across the ensemble. The simulated cells in the model represent this average firing rate of identically behaving ensembles of spiking neurons. However, in comparing the output of the model to observed spike rate maps, an assumed firing threshold is needed. Consider for instance that there is always top-down activation to the non-spatial attribute cell (*k*) once the enclosure has been fully explored (e.g. all positions trigger memory retrieval indicating that presence of *k*). Nevertheless, a *k*-cell with a sufficiently high firing threshold will become silent when the animal is between preferred memory locations owing to slightly lower memory retrieval strength. The specific firing threshold for each cell is assumed to dynamically adjust to keep each cell at its preferred on-average firing rate over a relatively long timeframe. For the reported simulations of viewpoint-dependent memories (i.e. ones that include head direction), cells that are active 5% of the time were analyzed. In other words, the simulation unfolds as dictated by the model equations, with continuous activation values for each cell,

but when analyzing the results, the simulated steps that produced the top 5% of activation values (*Equation 5*) across the entire recording session were used to specify the spike map for the entorhinal cells. Unlike the entorhinal cells, spike maps for place cells were determined more directly; for place cells, there is a spike every time that place cell activation is greater than the activation threshold, $\theta_a$.

$$act_{mECi} = r_i\left(d\right) + R * act\left(p_{RM}\right) * w_{RMi} \tag{5}$$

*Equation 5* is the real-valued activation of entorhinal cell *i* (e.g. a border cell, head direction cell, or *k* cell) at the time of memory retrieval based on the current bottom-up input response as determined in *Equation 1*, combined with top-down memory feedback, which is weighted by the retrieval constant, *R*, multiplied by the activation of the place cell for the retrieved memory as determined by *Equation 2*, and the appropriate bidirectional weight between the entorhinal cell *i* and the retrieved memory. The feedback retrieval constant was set to 3 to ensure that the retrieved memory was able to re-instantiate the entorhinal inputs by overriding the perceptual input to the entorhinal cells (this assumption is not critical and lower values of *R* produce similar results).

Close consideration of the model details reveals an apparent inconsistency; if entorhinal inputs are only active 5% of the time, how can they provide the sine wave neural tuning response functions dictated by *Equation 1*? This apparent inconsistency is resolved by hypothesizing that different entorhinal cells have different preferred firing rates. For instance, in addition to 5% firing rate cells, other entorhinal cells might fire 10%, 50%, or even 90% of the time. The sine wave tuning function is assumed to reflect the summed activity for a population of entorhinal cells that have the same weight connections to hippocampus and same weight connections to cortical regions providing excitatory perceptual input to the entorhinal cortex; when adding up the responses of similarly connected entorhinal cells, the weighted sum is a sine wave. Thus, when activating hippocampal memories, it is the sine wave (population code) that dictates which memories are retrieved, but when analyzing the results in terms spike rate maps, specific examples of the population are considered; namely entorhinal cells with 5% firing rates.

The effect of considering cells with different firing rates will be quantitative, rather than qualitative. For instance, border cells with firing rates greater than 5% will fire not just immediately adjacent to the border, but also fire at greater distances from the border. Grid cells with firing rates greater than 5% will exhibit the same grid spacing, but their grid fields will be larger, with smaller gaps between the grid fields.

## Results
### Grid cell results
The results below report the 'core' behavior of the simulation model in terms of grid cells. The appendix reports additional simulations that relate to place cells and head direction sensitivity of place cells in light of enclosure geometry and changes in enclosure geometry (e.g. remapping). The additional simulations in the appendix also report the finding that hexagonal grid fails to emerge or is disrupted depending on enclosure geometry.

### Pattern separation: some grid and place fields centered outside the enclosure
Memory consolidation causes the positions of the place cells to jiggle and move slightly in an attempt to make memories equally similar/dissimilar from each other to reduce confusion and confabulation between memories (i.e. pattern separation). In the case of place cell memories that were formed near the borders of the enclosure, consolidation often results in the place cell center moving just outside the enclosure. Thus, as in often observed in real data, some place fields appear to be centered outside the enclosure and some grid fields also appear to be centered outside the enclosure, with just the edge of the firing field falling inside the enclosure (this behavior can be seen in the raw firing maps of all papers reporting grid cell activity).

### The population response of border cells specifies locations on a torus
The cosine tuning curves of the simulated border cells represent distance from the border on both sides the border (i.e. firing rate increases as the animal approaches the border from either the

inside or the outside of the enclosure). Experimental procedures do not allow the animal to experience locations immediately outside the enclosure, but these locations remain an important part of the hypothetic representation, particularly when considering the modification of memories through consolidation (i.e. a memory created inside the enclosure might be moved to a location outside the enclosure). This symmetry about the border cell's preferred location is needed to maintain an unbiased representation, with a constant sum of squares for the border cell inputs (see methods section). Rather than using linear dimensions, all dimensions were assumed to be circular to keep the model relatively simple. This assumption was made because head direction is necessarily a circular dimension and by having all dimensions be circular, it is easy to combine dimensions in a consistent manner to produce multidimensional hippocampal place cell memories. Thus, the border cells define a torus (or more accurately a three-torus) of possible locations. This provides a hypothetical space of locations that could be represented.

In light of the assumption to represent border cells with a circular dimension, when a memory is pushed outside the East wall of the enclosure, it would necessarily be moved to the West wall of the enclosure if the period of the circular dimension was equal to the width of the enclosure. If this were true, then the partial grid field on one side of the enclosure would match up with its remainder on the other side. Such a situation would cause the animal to become completely confused regarding opposite sides of the enclosure (a location on the West wall would be indistinguishable from the corresponding location on the East wall). To reduce confusion between opposite sides of the enclosure, the width of the enclosure in which the animal navigated (*Figure 5*) was assumed to be half as wide as the full period of the border cells. In other words, although the space of possible representations was a three-torus, it was assumed that the real-world two-dimensional enclosure encompassed a section of the torus (e.g. a square piece of tape stuck onto the surface of a donut). The torus is better thought of as 'playing field' in which different sizes and shapes of enclosure can be represented (i.e. different sizes and shapes of tape placed on the donut). Furthermore, this assumption provides representational space that is outside the box without such locations wrapping around to the opposite side of the box.

In addition to explaining outside-the-box grid fields, the use of circular border cells explains the finding that the population response of grid cells lies on a torus-like surface (*Gardner et al., 2022*). More specifically, the model predicts that place cells lie on a torus defined by circular border cells and because place cells are the cause the grid fields, the population code of grid cells also lies on a torus. This is one of the first computational models to explain the toroidal nature of grid cells (but see *Guanella et al., 2007*; *McNaughton et al., 2006*).

Because the three border cells are equally spaced with a period of 2, a box of width 1 results in a situation where the border cells most strongly prefer positions immediately outside of the box. For instance, as shown in *Figure 5*, the West border cell would be most active if the animal were 1/6 of a distance East of the West border (i.e. positions along the red arrow, which is placed just outside the box). Because the animal is never given the opportunity to explore outside the box, this particular cell primarily fires when the animal is immediately adjacent to the West border (the closest allowable position to the cell's preferred position). The other two border cell basis sets are at an angle to the recording box, preferring positions along tipped axes (the blue and green arrows show the preferred positions). These could be described as corner cells, although because they are at a 60 degree tilt rather than 45 degrees, they end up firing primarily along the top and bottom borders.

Border cells are not always uniform in their responsiveness along a particular border (*Solstad et al., 2008*), although it is not clear whether such inhomogeneities are consistent with the predicted corner cell responses seen in *Figure 5*. In addition, the model predicts the existence of cells that fire between the borders, but not at the border, although such cells are relatively rare (*Solstad et al., 2008*). However, these predictions of corner-cells and between-border cells are not crucial to the proposed memory theory of grid cells. For instance, if the model contained 6 different non-orthogonal dimensions, rather than 3, not only would it provide a more accurate approximation to Euclidean distances (see *Figure 3*), but it would be easier to find border cells that were closely aligned with the orientations of the borders. Similarly, rather than assuming 3 entorhinal cells for each dimension, which requires a 2-to-1 ratio of border cells to between-border cells, it is possible to configure the model with just 2 opposing border cells for each dimension, in which case there are no between-border cells. These assumptions regarding the number of non-orthogonal dimensions and the number of cells per

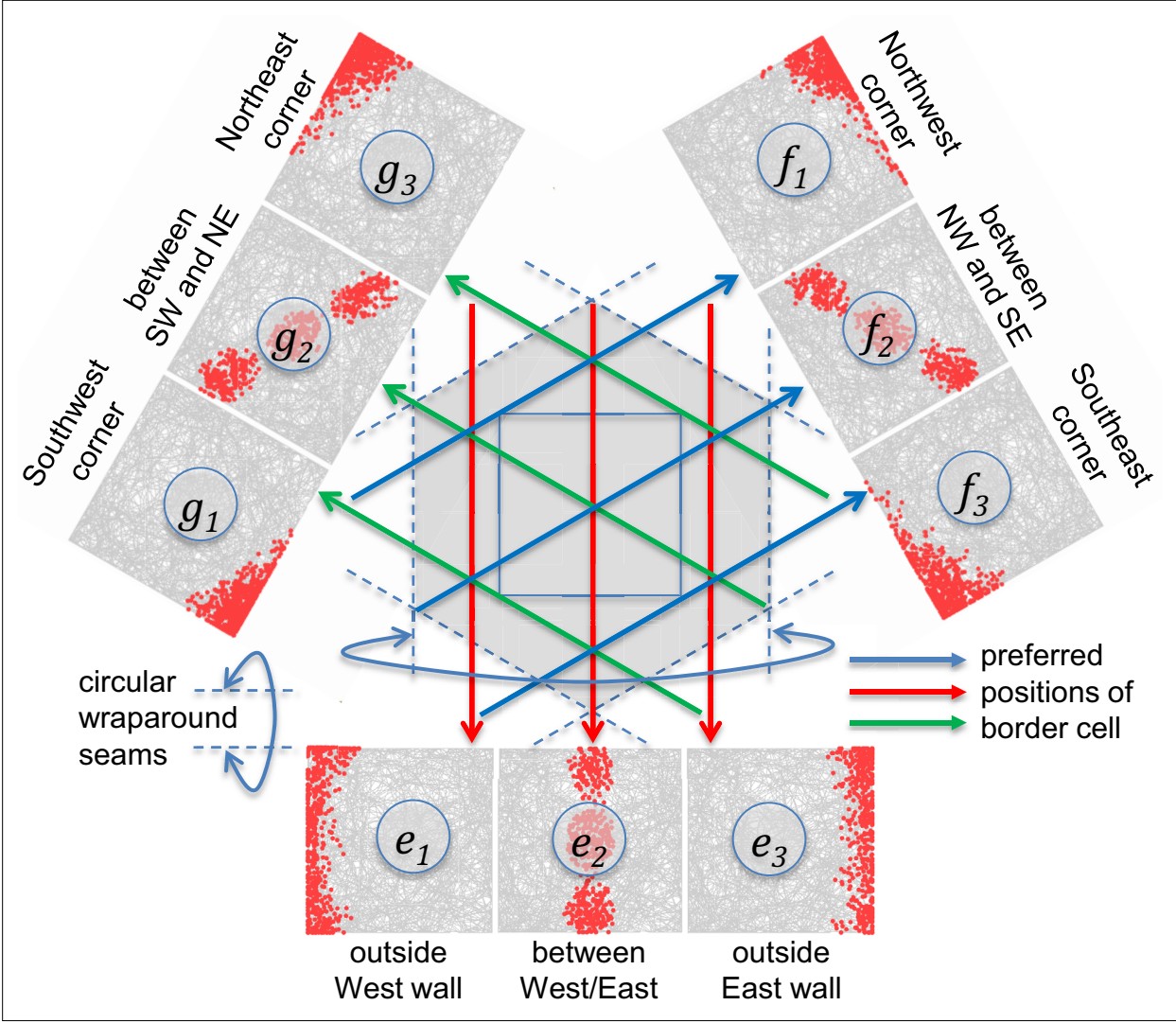

**Figure 5.** Position of square recording enclosure relative to circular border cell dimensions. The nine outer graphs show simulated spike rate maps for the border cells under the assumption that each border cell fires 5% of the time based on the summation of its bottom-up and top-down inputs (*Equation 5*). The enclosure was assumed to be half as wide as the full period of the circular border cells, minimizing confusion between opposite borders. The dashed lines indicate wraparound seams such that each dashed line is connected to the opposite dashed line to create a cylinder for that dimension, resulting in a hexagonally connected 3-torus for the entire space across the three non-orthogonal dimensions. The gray curved lines within each of the 9 border cell graphs show the path of a simulated animal across 10,000 steps. The red dots (500 per graph) show positions where the simulated border cell fired. The graphs for the G and F dimensions are rotated to align the graphs with the corresponding allocentric directions. The preferred positions for each of the nine cells are indicated by the entire line length of the red, green, or blue arrows that point to the corresponding firing map. The letter labels inside each graph indicate the simulated cell using the same labeling scheme as in *Figure 2*.

dimension were made primarily to keep the model simple and computationally efficient. The core of the model is its prediction that place cells are conjunctions of *what* and *where* (i.e. memories) and that grid cells are non-spatial attributes (*what*).

## Grid fields are immediately apparent and align with enclosure walls

Behavior of the model was first explored without head direction, based on the real-world two-dimensional inputs (*X/Y*) as captured by the three non-orthogonal basis sets (*E/F/G*), in combination with a single non-spatial cell *k*. *Figure 6* shows firing maps for the non-spatial cell during initial recording in a novel box (i.e. the first 10,000 simulation steps) as well as the same simulation after 10 prior sessions (familiar box) of experience (e.g. recording after 100,000 prior steps). To explore how the model behaves, this was done with three different consolidation thresholds, which affect

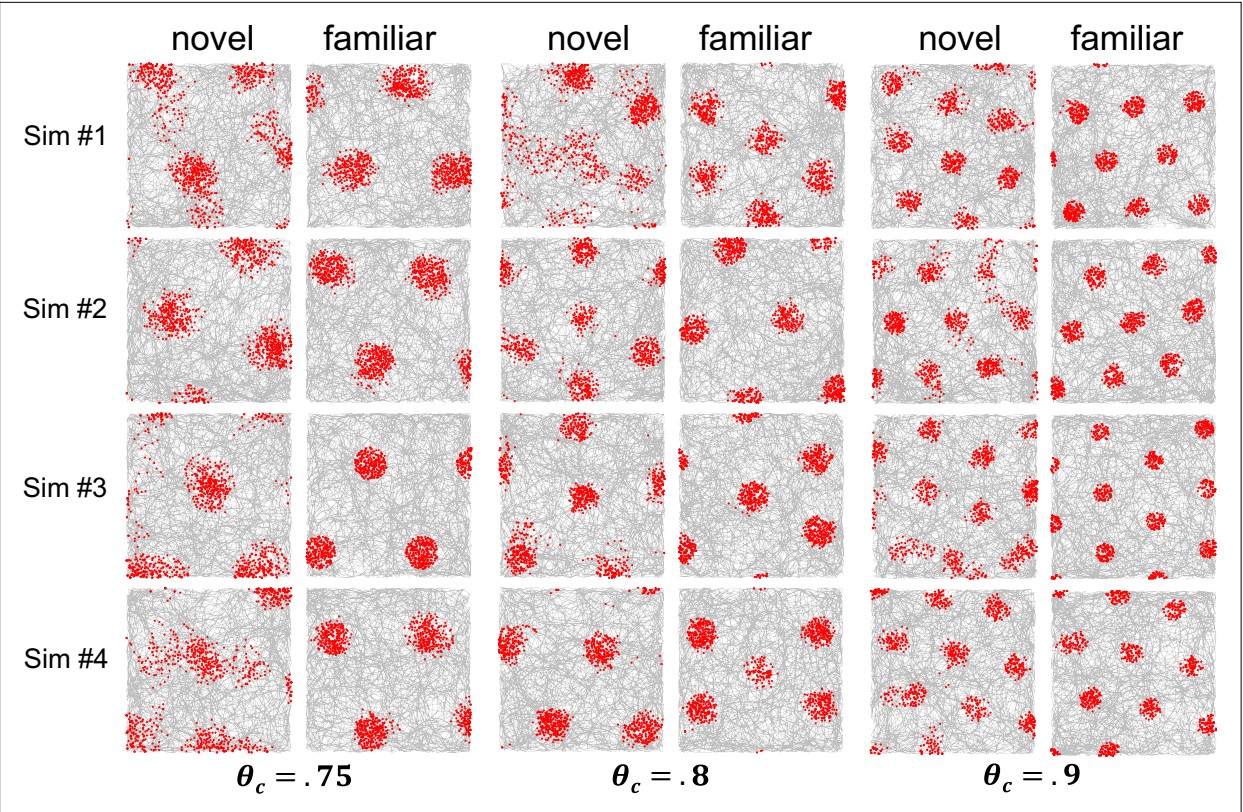

**Figure 6.** Results for simulated grid cells representing a non-spatial attribute common to a set of place cell memories when not including head direction in the place cell memories (allocentric memories). These simulations are an exploration of how the model behaves with different parameter values. In each case, the first four simulations are shown, regardless of outcome. Results are shown when adopting one of three different consolidation thresholds, $\theta_c$, which produce different spacings between memories. The corresponding activation thresholds, $\theta_a$, were 0.86, 0.9, and 0.92 to make sure that memories were created and activated with a somewhat closer spacing than that dictated by the consolidation threshold. Each pair of novel and familiar firing maps is the same simulation, with the novel firing map showing the first 10,000 simulated steps in a novel environment and the familiar firing map showing 10,000 simulated steps after 100,000 prior steps (e.g. after the equivalent of 10 sessions of prior experience).

grid spacing. For these simulations, the results from cells with a 10% firing rate (rather than 5%) were analyzed to ensure coverage of all grid fields for the case of a closely spaced grid. As discussed in the model methods, the model assumes a population of similarly connected entorhinal cells that collectively produce the sine wave tuning functions (*Equation 1*). As such, the choice to analyze cells with different firing rates amounts to considering grid cells that have larger versus smaller grid fields, but with equivalent spacing/orientation between grid fields. In this case, 10% firing rate grid cells were selected for their property of exhibiting grid field sizes that allow easy visual assessment of the grid pattern.

As seen in *Figure 6*, in some cases (e.g. the bottom-right two graphs) the orientation and position of the grid hardly changes with additional experience whereas in other cases (e.g. the bottom-middle two graphs) the position and/or orientation of the grid changed. Because the consolidation threshold remained the same with experience, the grid spacing remained the same. It has been reported that grid spacing tends to shrink with experience (*Barry et al., 2012*) and this could be captured by gradually increasing the consolidation threshold. In some cases, the novel grid was somewhat smeared out for some of the grid fields, reflecting the ongoing consolidation of memories. This lack of grid regularity for a novel environment has been reported (*Barry et al., 2012*). In nearly all cases, there were grid fields that appeared to be partially outside the box, as is typically observed in the firing maps of grid cells.

In real data, the grid array tends to align with a straight wall of the enclosure (*Krupic et al., 2015*) and this is true of many of the simulations in *Figure 6*. To better quantify this result, an additional 20 simulations were performed at each consolidation threshold after 30 prior sessions experience

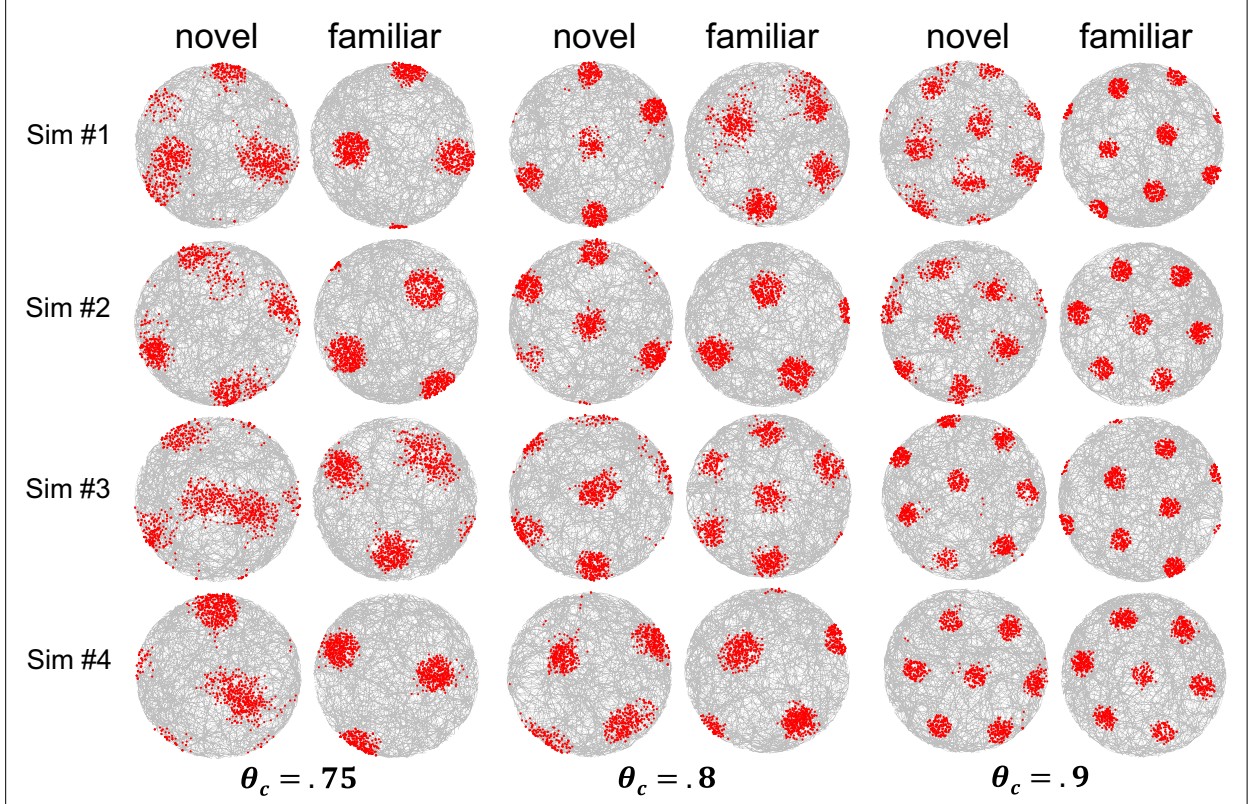

**Figure 7.** Simulation results when using the same parameters and settings as in *Figure 6* for a circular enclosure.

(i.e. excessive prior experience to ensure stabilization of consolidation). For the 0.75 consolidation threshold (far spacing), 10 simulations produced a grid that was horizontally oriented and 5 were vertically oriented. The remaining five simulations produced a grid with four main firing fields that were arrayed in square arrangement rather than a hexagonal arrangement. For the 0.8 consolidation threshold (intermediate spacing), all grids were hexagonal, with 2 of the simulations producing a grid that was horizontally oriented and the remaining 18 producing a grid that was vertically oriented. For the 0.9 consolidations threshold (close spacing), all grids were hexagonal, with six of the simulations producing a grid that was horizontally oriented, while six simulations produced a grid that was vertically oriented. The remaining eight simulations were tipped at some orientation other than vertical or horizontal. In summary, nearly all grids were hexagonal and the grid usually, but not always, aligned with the vertical or horizontal axis of the square recording enclosure, with this being more likely to occur with a far grid spacing (i.e. more widely separated hippocampal place cell memories).

It is not clear from these results whether the tendency of the grid to align with the box reflects the geometry of the box or the assumption that border cell dimension *E* aligns with one of the box borders. To address this question, another set of similar simulations was run with a circular enclosure, but with the dimension *E* still in the horizontal direction, to see if the grid still tended to be horizontally or vertically oriented. As seen in *Figure 7*, hexagonal grids readily emerge in a circular enclosure (i.e. the grid does not require straight walls). Overall, the orientation of the grids still seemed to align with the vertical or horizontal directions, where the definition of vertical and horizontal is now relative to the underlying border cell dimension *E*. However, this alignment occurred slightly less often as compared to the square enclosure. To quantify this observation, 20 simulations were run at each of the three grid spacings after 30 sessions of prior experience. For the 0.75 consolidation threshold (far spacing), 10 simulations produced grids that were horizontally oriented and 2 were vertically oriented. An additional seven were tipped at random angles and the remaining simulation produced a square grid that was horizontally/vertically aligned. For the 0.8 consolidation threshold (intermediate spacing), four simulations produced horizontally oriented grids and 10 produced vertically oriented grids. An additional three were tipped at random angles and the remaining three failed to produce

a hexagonal grid (the grid was either a square grid or four place fields arranged in a Y pattern). For the 0.9 consolidations threshold (close spacing), one simulation produced a grid that was horizontally aligned and three were vertically aligned. For the remaining 16 simulations, it was tipped at some orientation other than vertical or horizontal.

In summary, these simulations show that the grid is immediately established and that the grid has a tendency to align with the underlying border cell directions (e.g. the border cell direction *E* or perpendicular to *E*), particularly with a widely spaced grid in the case of a square enclosure (i.e. the geometry of the enclosure also seemed to play a role in light of the reduced vertical/horizontal alignment within a circular enclosure). Thus, the finding that the grid tends to align with walls of a square enclosure *Krupic et al., 2015* follows from the assumption that one of the underlying border dimensions aligns with the walls of a square enclosure.

Additional analyses revealed that this tendency to align with border cell dimensions is caused by weight normalization (**Step 6** in the pseudocode). Specifically, connection weights cannot be updated above their maximum nor below their minimum allowed values. This results in a slight tendency for consolidated place cell memories to settle at one of the three peak values or three trough values of the sine wave basis set. Consolidation weight changes may attempt to push a weight value below zero or above the maximum allowed value. However, weight normalization will bring the weight back into the allowable range. As a result, it is slightly more difficult for consolidation to push a weight over the peak or under the trough of the tuning curve (but this can occur owing to the other two cells in a basis set, which would be adjusting their weights in a more linear portion of the tuning curve). This 'stickiness' at one of six peak or trough values for each basis set is very slight and only occurred after many consolidation steps. In terms of biological systems, there is an obvious lower-bound for excitatory connections (i.e. it is not possible to have an excitatory weight connection that is less than zero), but it is not clear if there is an upper-bound. Nevertheless, it is common practice with deep learning models include an upper-bound for connection weights because this reduces overfitting (*Srivastava et al., 2014*) and there may be similar pressures for biological systems to avoid excessively strong connections.

## Grid cell modules reflect the dense packing of memories in three dimensions: an example with X, Y, and head direction

According to this memory model of spatial navigation, place cells are the conjunction of location, as defined by border cells, and one or more properties that are remembered to exist at that location. Such memories could, for instance, allow an animal to remember the location of a food cache (*Payne et al., 2021*). The next set of simulations investigates behavior of the model when one of the to-be-remembered properties is head direction at the time when the memory was formed (e.g. the direction of a pathway leading to a food cache). Indicating that head direction is an important part of place cell representations, early work on place cells in mazes found strong sensitivity to head direction, such that the place field is found in one direction of travel but not the other (*McNaughton et al., 1983*; *Muller et al., 1994*). Place cells can exhibit a less extreme version of head direction sensitivity in open field recordings (*Rubin et al., 2014*), but the nature of the sensitivity is more complicated, depending on location of the animal relative to the place field center (*Jercog et al., 2019*).

It is possible that some place cell memories do not receive head direction input, as was the case for the simulations reported in *Figure 6*/**7** – in those simulations, place cells were entirely insensitive to head direction, owing to a lack of input from head direction cells. However, removal of head direction input to hippocampus affects place cell responses (*Calton et al., 2003*) and grid cell responses (*Winter et al., 2015*), suggesting that head direction is a key component of the circuit. Furthermore, if place cells represent episodic memories, it seems natural that they should include head direction (i.e. viewpoint at the time of memory formation).

In the simulations reported next, head direction is simply another property that is conjoined in a hippocampal place cell memory. In this case, a head direction cell should become a head direction conjunctive grid cell (i.e. a grid cell, but only when the animal is heading in a particular direction), owing to memory feedback from the hexagonal array of hippocampal place cell memories. When including head direction, the real-world dimensions of variation are across three dimensions (X, Y, and head direction) rather than two, and consolidation will cause the place cells to arrange in a

three-dimensional volume. The simulation reported below demonstrates that this situation provides a 'grid module'.

One of the key findings in the grid cell literature is the existence of anatomically arranged modules, with nearby grid cells having the same orientation, spacing, and distortion (i.e. the degree to which the grid is slightly elongated in a particular direction), but different spatial phases (*Stensola et al., 2012*; *Yoon et al., 2013*). In other words, the grid of nearby cells is often identical but shifted. Some theories suggest that these grid cell modules provide a two-dimensional Fourier basis set for location coding (*Rodríguez-Domínguez and Caplan, 2019*; *Stachenfeld et al., 2017*; *Wei et al., 2015*). However, the proposed non-spatial memory model of grid cells suggests a different interpretation – rather than tiling a two-dimensional space, grid cell modules might emerge from the consolidation of memories into a three-dimensional volume. For memories that differ from each other in three real-world dimensions, with a fixed minimum distance between memories, their arrangement is equivalent to the dense packing of equal-sized spheres in a three-dimensional volume (e.g., putting marbles in a glass). In this case, the arrangement of memories should follow the Kepler conjecture (*Hales, 2005*), which states that there are two ways in which equal-sized spheres can be arranged with the densest possible packing. Both solutions are hexagonal in nature, containing separate layers of spheres that are hexagonally arranged with the same orientation and spacing, but with different phases: face-centered cubic packing (FCP), entails three different hexagonal layers, whereas hexagonal close packing (HCP) entails two different hexagonal layers.

Consider the 3D volume defined by *X*, *Y*, and head direction, *H*. If memories include head direction (i.e. memories of where a non-spatial attribute can be found and the head direction at the time of memory formation), then consolidation may lead to a close-packed FCP or HCP arrangement. In this case, the activity of different head direction cells might capture different hexagonal layers of the dense packed lattice of memories. If so, each head direction cell will exhibit a grid pattern of the same orientation and spacing as other head direction cells, but with the spatial phase of the grid shifted as dictated by FCP or HCP geometry. In the reported simulation, the three-dimensional memories naturally settled on the FCP solution (for related theoretical proposals that involve 3D hexagonal tiling, see *Mathis et al., 2015*; *Stella and Treves, 2015*), with three layers that were separately captured by the three different head direction cells in the head direction basis set (*Figure 8*).

*Figure 8* shows a typical result for the non-spatial attribute cell (*Figure 8A*) and the three head direction cells (*Figure 8B*) while navigating in a familiar box. The six border-cell firing maps for this particular simulation were plotted in *Figure 5*. The non-spatial attribute cell produced a grid field that was rotated by 90 degrees relative to the grid fields of the head direction cells and a grid spacing that was more closely spaced by a factor equal to the square root of three, reflecting the superposition of FCP layers (*Figure 8C*). This coordination between grid cells of different grid spacings was recently been observed (*Waaga et al., 2022*), and in the model this coordination arises from memory retrieval for the same set of memories. The place cell memories were arranged into 6 different FCP layers (a repeating sequence of three layers), providing an allocentric representation of a familiar box based on combinations of viewpoint-dependent memories (i.e. based on place cell memories that include head direction). More specifically, each location is remembered from two opposite viewpoints (*Figure 8D*). Two of the three head direction cells exhibited the same grid orientation and spacing but the grid fields were shifted. The third head direction cell would have produced something similar except that its place cell memories were mostly outside-the-box (the spike rate maps show hints of the outside-the-box memories near the borders). In keeping with this result for simulated head direction cell $h_2$, it has been reported that some mEC cells that initially appear to have a single place cell firing field are in truth grid cells when the box borders are removed – removing the borders reveals that there are additional grid fields outside the box (*Savelli et al., 2008*).

In summary, this simulation produced a head direction grid cell module in combination with a view independent (i.e. allocentric) grid array for the non-spatial cell (places where the non-spatial property *k* can be found), with this second array exhibiting a closer spacing by a factor of the square root of 3 and an orientation that is rotated by 90 degrees. The wider grid spacing corresponds to the 'layers' taken from the three-dimensional FCP solution and the closer grid spacing is the 'projection' through the layers (e.g. when collapsing across head direction).

Is this layer/projection pairing of grid modules compatible with empirical data? *Stensola et al., 2012* simultaneously recorded as many as 186 mEC grid cells in individual rats, finding approximately

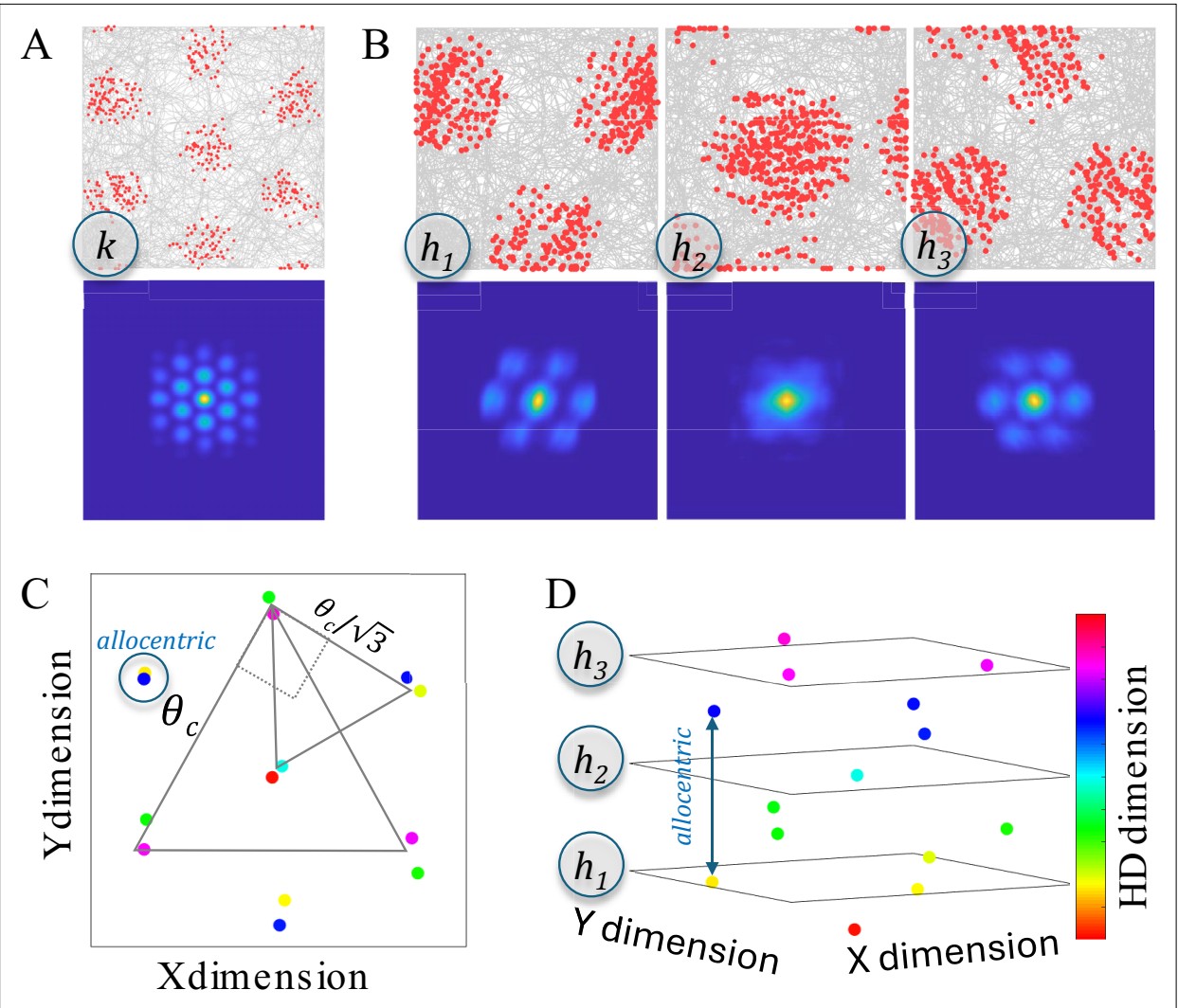

**Figure 8.** Results from a simulation that included head direction (0.8 consolidation threshold). These results are for a familiar box (10 sessions of prior experience). The plotted firing maps (red dots for spikes) and spatial autocorrelation (blue for low correlation and yellow for high correlation) maps are labeled according to the specific cell labels used in *Figure 2*. The border cell results for this particular simulation are shown in *Figure 5*. (**A**) The grid fields for the non-spatial attribute, $k$, are vertically aligned. (**B**) In contrast the non-spatial attribute, the grid fields for the head direction cells are horizontally aligned. Cells $h_1$ and $h_3$ have the same grid orientation and spacing as each other, as would cell $h_2$, except that for $h_2$, its corresponding place cell memories are mostly outside the box. As in real grid cell modules, the grid fields for each of the three head direction cells ($h_1$ to $h_3$) are shifted relative to each other. (**C**) The head direction grid spacing is the square root of 3 larger than the non-spatial attribute grid spacing, reflecting the superposition of face-centered cubic packing layers. (**D**) For this grid spacing, there were 6 HD layers, such that every third layer produced a nearly identical spatial phase of HD-sensitive place cell memories, except that the preferred HD was 180 degrees opposite. This provides an allocentric arrangement of the place cell memories whereby the non-spatial attribute at each location was remembered from two 180 degree opposite viewpoints (head directions). The colored dots in the three-dimensional plot are based on the final positions of the 14 memories at the end of the simulation, as calculated from the weight matrix for each place cell, with the color of the dots showing the preferred head direction of the corresponding memory.

four different grid modules for each animal. The ratio of grid spacing between modules was found to be approximately the square root of 2 when comparing modules that had grid spacing values that were close in value. This can be contrasted with the square root of 3 relationship predicted by the FCP grid module produced in this simulation. However, an alternative interpretation of the Stensola data would place the modules into interleaved sets of two (each pair of grid modules consisting of one head direction dependent and one allocentric module) with the two modules in a pair differing in orientation by 90 degrees and differing in grid spacing by the square root of 3.

Consider the possibility that one mEC grid modules is based on head direction (viewpoint) in remembered positions relative to the enclosure borders (e.g. learning the properties of the enclosure,

such as the metal surface) while a different grid module is based on head direction in remembered positions relative to landmarks exterior to the enclosure (e.g. learning the properties of the experimental room, such as the sound of electronics that the animal is subject to at all locations). This might explain why a deformation of the enclosure (moving one of the walls to form a rectangle rather than a square) caused *some of the grid modules but not others* to undergo a deformation of the grid pattern in response to the deformation of the enclosure wall (see also *Barry et al., 2007*). More specifically, suppose that the movement of one wall is modest and after moving the wall, the animal views the enclosure as being the same enclosure, albeit slightly modified (e.g. when a home is partially renovated, it is still considered the same home). In this case, the set of non-orthogonal dimensions associated with enclosure borders would still be associated with the now-changed borders and any memories in reference to this border-determined space would adjust their positions accordingly in real-world coordinates (i.e. the place cells would subtly shift their positions owing to this deformation of the borders, producing a corresponding deformation of the grid). At the same time, there may be other sets of memories that are in relation to dimensions exterior to the enclosure. Because these exterior properties are unchanged, any place cells and grid cells associated with the exterior-oriented memories would be unchanged by moving the enclosure wall.

Similar to head direction, there are other factors that vary during the recording session, which may explain grid modules that do not depend on head direction. For instance, in the current simulation, it was assumed that the animal adopted a new random goal direction at every time step, independent of head direction (e.g. the animal decides to alter its course slightly towards direction, *g*, for the next step, even though its current head direction is, *h*). To assess whether goal direction might form a grid module, another simulation was run in which the third dimension was based on a basis set of three goal direction cells ($g_1$ to $g_3$) rather than head direction. The results were essentially identical to that shown in *Figure 8*, swapping the labels of $h_1$ to $h_3$ for $g_1$ to $g_3$, except that the goal direction cells were insensitive to head direction (close examination of the firing maps in *Figure 8B* reveals that they tend to occur on trajectories that align with the preferred head direction of the corresponding head direction cell, but this was not true of the goal direction simulation).

## Grid fields appear immediately in a novel environment, whereas place cell positions take time to stabilize

The head direction grid module results reported above were based on a familiar environment, with 10 sessions worth of prior experience. However, grid field patterns are often seen immediately upon entry into a novel environment, although the pattern may be somewhat less regular for a novel environment (*Barry et al., 2012*). In keeping with this finding, the two-dimensional allocentric simulations (i.e. the simulations in *Figure 6*, *Figure 7* that did not include head direction) produced a grid immediately in a novel environment. To ascertain whether the grid is also immediately apparent when including head direction, the same head direction simulation as reported in *Figure 8* was run multiple times, but with different amounts of prior experience before recording the results. As shown in *Figure 9A* and *Figure 9B*, the non-spatial attribute cell (*k*), produced an immediately apparent grid pattern. Additional sessions of prior experience served to make the grid more precise as revealed by the grid score measure. However, the results were fundamentally different for the head direction conjunctive grid cells (i.e. head direction cells that become head direction conjunctive grid cells). For the head direction cells, there was no apparent grid without prior experience (grid score of 0), but additional sessions of prior experience produced reliable grid scores for the head direction cells (the graph plots the grid score for the head direction cell with the highest grid score of the three head direction cells, $h_{max}$, considering that some of the head direction cells fail to produce a grid owing to outside-the-box memories, as was the case for cell $h_2$ in *Figure 8B*).

Why was the grid immediately apparent for the non-spatial attribute cell whereas the grid took considerable prior experience for the head direction cells? The answer relates to memory consolidation and the shifting nature of the hippocampal place cells. Head direction cells only produced a reliable grid once the hippocampal place cells (aka, memory cells) assumed stable locations. During the first few sessions, the hippocampal place cells were shifting their positions owing to pattern separation and consolidation. But once the place cells stabilized, they provided reliable top-down memory feedback to the head direction cells in some places but not others, thus producing a reliable grid arrangement to the firing maps of the head direction cells. In other words, for the head direction cells,

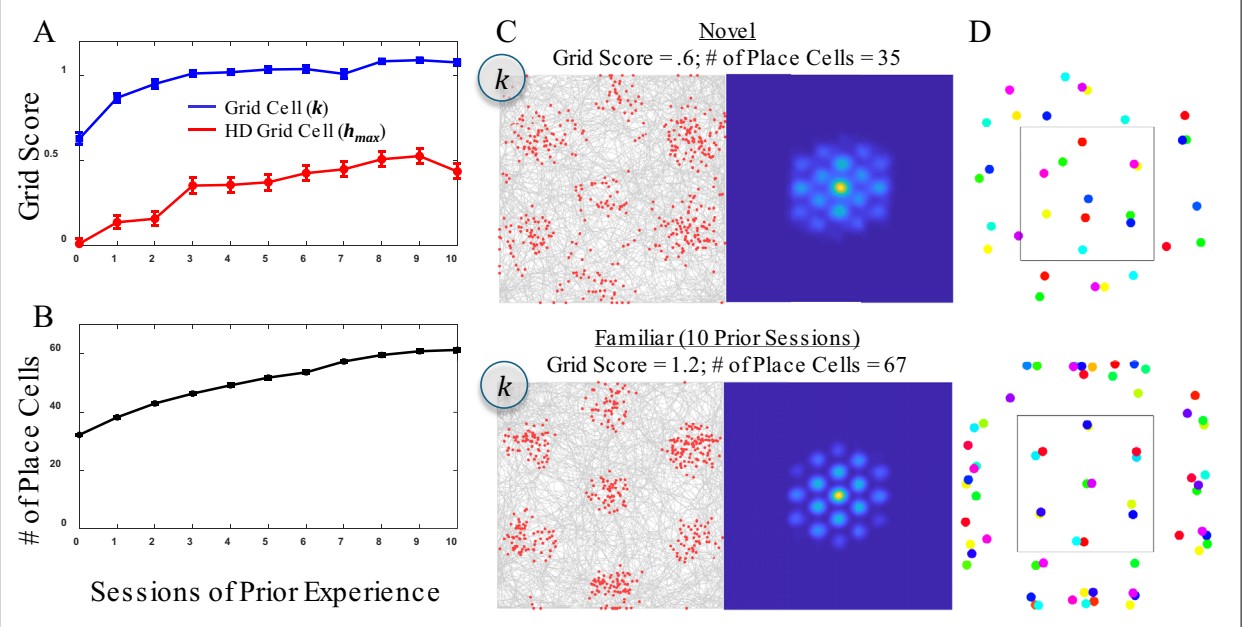

**Figure 9.** Simulations depicting the learning of non-spatial grid fields (**k**) and head direction conjunctive grid field (**h**), using the same parameter values as in *Figure 8*. (**A**) Average grid scores across 100 simulations, with prior experience ranging from novel (no prior experience) to 10 sessions of prior experience, show that the grid field of the non-spatial cell, *k*, was immediately apparent whereas the head direction grid cells required prior experience before grid fields stabilized. Because one of the three head direction grid cells tended to have a single central grid field, with potentially outside the box additional grid fields (see cell $h_2$ in *Figure 8*), the maximum grid score ($h_{max}$) from the three head direction cells was used. Error bars are plus and minus one standard error of the mean. (**B**) A plot showing the number of place cell memories created by the end of the simulation reveals that the number of memories grew in a similar manner to the stabilization of the head direction grid fields. (**C**) Representative firing map and spatial autocorrelation results for a novel environment and a familiar one reveal that that the non-spatial cell's grid fields were regular but less precise in a novel environment, reflecting the shifting nature of the place fields. (**D**) Plots showing the final positions of all place cell memories for the representative simulations, reveal that the main effect of prior experience was formation of outer memories that conform to the shape of the box; as the animal learned head direction representations for the borders of the box, the grid cell module based on head direction stabilized. The color of the dots represents the head direction associated with each memory using the same color scale as *Figure 8*. *Figure 9—video 1* shows memory formation and consolidation in this situation that includes head direction.

The online version of this article includes the following video for figure 9:

**Figure 9—video 1.** Place cell stabilization.

https://elifesciences.org/articles/95733/figures#fig9video1

the grid only appeared once the place cells stabilized. This slow stabilization of place fields is a known property (*Bostock et al., 1991*; *Frank et al., 2004*).

In the simulation, the place cells did not stabilize until a sufficient number of place cells were created (*Figure 9C*). Specifically, these additional memories were located immediately outside the enclosure, around all borders (*Figure 9D*). These 'outside the box' memories served to constrain the interior place cells, locking them in position despite ongoing consolidation. This dynamic can be seen in a movie showing a representative simulation. The movie shows the positions of the head direction sensitive place cells during initial learning, and then during additional sessions of prior experience as the movie speeds up (see link in *Figure 9* capture).

Why did the non-spatial grid cell (*k*) produce a grid immediately, before the place cells stabilized? As discussed in relation to *Figure 8*, the non-spatial grid cell is the projection through the 3D volume of real-world coordinates that includes X, Y, and head direction. Each grid field of a non-spatial grid cell reflects feedback from several place cells that each have a different head direction sensitivity (see for instance the allocentric pairs of memories illustrated in *Figure 8C and D*). Thus, each grid field is the average across several memories that entail different viewpoints and this averaging across memories provides stability even if the individual memories are not yet stable. This average of unstable memories produces a blurry sort of grid pattern without any prior experience.

A final piece of the puzzle relies on the same mechanism that caused the grid pattern to align with the borders as reported in the results of *Figures 6 and 7*. Specifically, there are some 'sticky' locations with ongoing consolidation because the connection weights are bounded. Because weights cannot go below their minimum or above their maximum, it is slightly more difficult for consolidation to push or pull connection weights over the peak value or under the minimum value of the tuning curve. Thus, the place cells tend to linger in locations that correspond to the peak or trough of a border cell. There are multiple peak and trough locations but for the parameter values in this simulation, the grid pattern seen in *Figure 9C* shows the set of peak/trough locations that satisfy the desired spacing between memories. Thus, the average across memories shows a reliable grid field at these locations even though the memories are unstable.

In summary, these results indicate that some grid patterns are learned more slowly than others in a completely novel environment (i.e. an environment that is completely foreign to the animal). More specifically, head direction conjunctive grid cells require stabilization of the place cell memories in the face of ongoing consolidation of view-dependent memories whereas grid cells that are not sensitive to head direction may reflect on-average memory feedback from head direction sensitive place cells (i.e. several place cells with the same place field, but with different head direction sensitivity). However, it should be noted that these simulations were for a 'blank slate' animal that had no prior experience with any similar enclosures. This is unlikely to be the situation in the real world, and it may be that the empirical finding that head direction conjunctive grid cells can be found immediately reflects memories of similar experiences in similar enclosures rather than very recent memories from the last few minutes or seconds.

## Consolidation of interior place cells depends on exterior place cells: the cognitive map emerges after learning specific views of the boundaries

*Figure 9B* shows that the main effect of prior experience is the addition of new place cell memories, which produce more precise grid fields for the non-spatial attribute cells (*Figure 9C*); these new place cells result in stability (i.e. consolidation no longer moves the positions of the place cells). The place field locations for these additional memories were primarily outside of the box (*Figure 9D*). Considering that place fields are the cause of individual grid fields for the model, this may relate to the finding that encounters with boundaries can serve to stabilize grid fields (*Hardcastle et al., 2015*). More specifically, memories initially formed inside the box in positions adjacent to the borders push against each other during consolidation owing to pattern separation (*Chanales et al., 2017*), causing some memories to move outside the box. This doesn't necessarily mean that the animal remembers positions outside the box. Instead, these memories could be thought of as characterizations of the borders; by placing their preferred positions outside the box, these memories become highly selective for positions immediately inside the borders of the box (i.e. the inside border locations are the only allowable real-world circumstances that come closest to their preferred inputs). By analogy, the faces of famous individuals are more easily recognized when drawn with impossibly exaggerated facial features (*Mauro and Kubovy, 1992*), suggesting that the memory representation is a pattern-separated impossible exaggeration that uniquely identifies that individual. In addition to being highly selective to the borders of the box, these exaggerated border memories constrain the interior memories, which stabilizes interior place cells, revealing the grid fields of the head direction cells.

The effect of outside-the-box place cells is further explored by considering in detail the firing maps of the place cells (*Figure 10*) for a familiar box (this particular simulation is different than the one shown in *Figure 9*, but used the same parameters as *Figure 9*). Similar to the 'familiar' results shown in *Figures 9 and 10A* shows the final positions for the 48 place cells that were learned after 10 sessions of prior experience, as well as the firing maps for the three head direction cells (cells $h_1$, $h_2$, and $h_3$ using the cell labeling from *Figure 2*). Although similar results were show in *Figure 9*, the results for this particular simulation are shown such that the preferred location of each place cell (*Figure 10A*) can be directly compared to the observed place field firing maps (*Figure 10B*) of each place cell. This comparison highlights how the outside-the-box place cells constrain the nature of the cognitive map.

The place cells were grouped into two types according to observed firing rate: ones that fired less than 2.5% of the time versus ones that fired more than 2.5% of the time (for place cells, firing rates reflect the portion of stimulated steps for which the cell was above the activation threshold). The cutoff of 2.5% was chosen because this cleanly divided the place cells into those with preferred

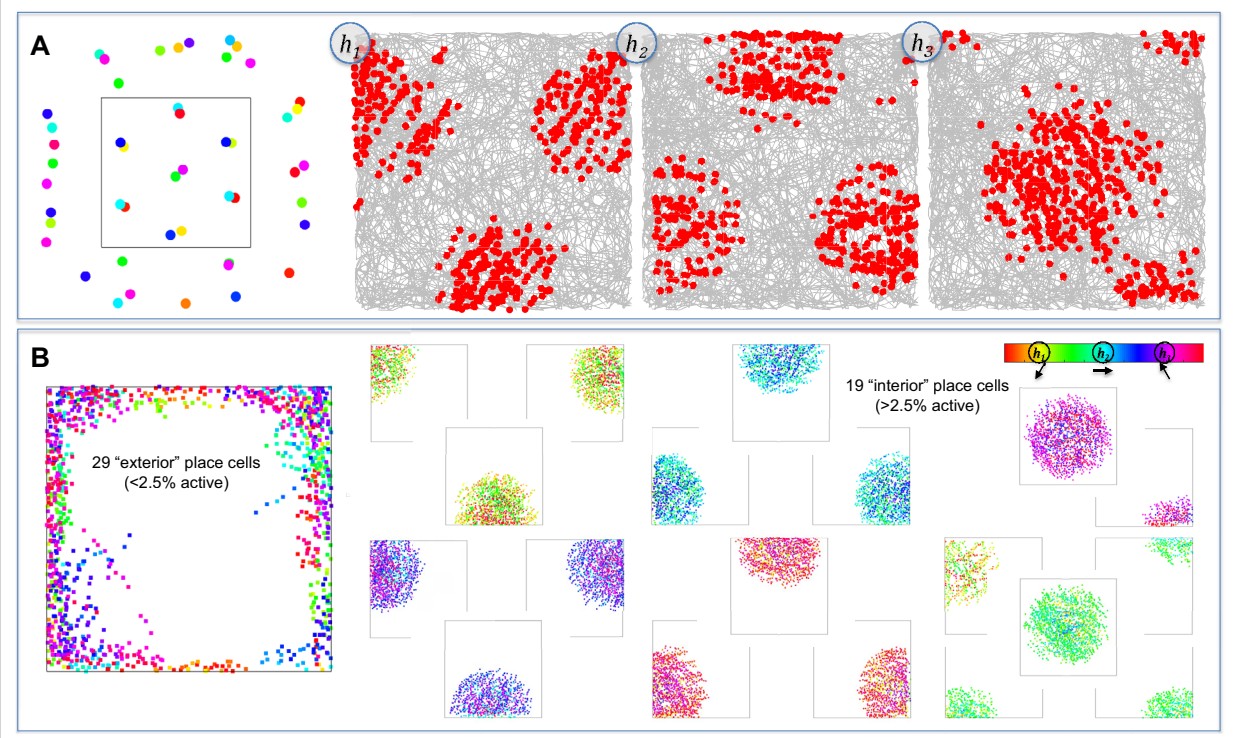

**Figure 10.** Using the same parameters as *Figures 8 and 9*, a comparison of place cell firing maps for interior versus exterior place cells demonstrates how head direction sensitivity depends on location and how interior place cells give rise to head direction conjunctive grid cells. (**A**) The final positions (preferred positions and preferred head directions) of the 48 place cell memories and head direction cell firing maps. (**B**) Firing maps of the place cells. The place cells were divided into 29 'exterior' place cells that were active less than 2.5% of the time (these are accumulated onto the same graph on the left) versus 19 'interior' place cells (one graph per cell on the right). The eight interior place cells in the top two rows of were the cause of the head direction conjunctive grid cell firing map shown immediately above in panel. The color bar that relates head directions to colors includes the labels for the head direction cells ($h_1$, $h_2$, and $h_3$) and the adjacent arrows depict the three head directions. The 11 interior place cells in the bottom two rows selected for head directions that were between the preferred head directions of the head direction cells (i.e. the other three hexagonal layers from the six-layer hexagonal close packing). The color of each spike for the place cells shows the head direction of the simulated animal at the time of the spike. For the place field center of the interior place cells, head direction sensitivity was weak (i.e. a greater range of colors in the center of each place field), whereas the firing maps for the exterior place cells were highly view dependent (consistent color for each cluster of spikes).

positions outside the box versus inside the box. As shown in left-side of *Figure 10B*, the less active place cells only fired when the animal was near a border, as expected considering that these place cells had their preferred locations outside-the-box. Therefore, these low activity place cells are termed 'exterior' place cells, which capture memories for specific locations near a border, as seen from a particular viewpoint. This viewpoint dependency is specified by the color coding of the spikes shown in *Figure 10B*, which shows that the exterior place cells only fire when the animal is not only close to the border, but also when the animal has its head pointed in a very specific direction (i.e. each cluster of spikes, which correspond to a particular exterior place cell, has a very specific color).

The right-hand side of *Figure 10B* shows the 'interior' place cells. These 19 cells are arranged in a precise hexagonal pattern, and they are the source of hexagonal grid fields exhibited by the head direction cells. The firing maps for the eight hippocampal place cells in the top two rows plot the results for head direction-sensitive place cells that have the same preferred head direction as the head direction cell plotted immediately above each graph. In contrast, the 11 place cells in the bottom two rows are sensitive to head directions that were 180 degrees opposite (i.e. head directions that were between the preferred head directions of the three head direction cells). Unlike the exterior place cell firing maps, which were highly selective for head direction, these interior place cells were less selective, firing to a range of head direction values, particularly when the animal was in the center of the place field (i.e. at the center of each cloud of spikes, the head direction colors take on hues other than the preferred head direction of the corresponding head direction cells).

## Discussion

These results demonstrate that a memory model can explain many of the findings in the rodent electrophysiology literature that were previously assumed to reflect the operations of an entorhinal/hippocampal navigation system. By explaining navigation with a memory model, this provides unified account of the rodent navigation and human memory literatures. Furthermore, the model explains cell surveys of the rodent entorhinal that *appear* to indicate that most of the information flowing into rodent hippocampus is spatial. In the proposed account, the function of cells that exhibit a hexagonal spatial grid has been misclassified as spatial (see also *Luo et al., 2024*). In the model, place cells are memory cells, representing conjunctions of *what* and *where,* and grid cells are non-spatial *what* cells that fire at specific locations where the non-spatial attribute is remembered. In a typical rodent navigation study, there are diffuse non-spatial properties that may become associated with locations throughout the recording enclosure (e.g. electronic noise, lighting, fear, attached wires, etc.). Because there are many such properties, there are many grid cells. Some of the cells in entorhinal cortex are truly spatial on this account, such as head direction cells (including head direction conjunctive grid cells) and border cells. However, according to this memory model, grid cells without head direction sensitivity are non-spatial. These grid cells only appear to be spatial because their activity was recorded while the animal navigated an enclosure in which the non-spatial attributes were the same at all locations. The uniformity of the non-spatial attributes leads to a situation in which memories become arrayed in a hexagonal grid owing to pattern separation.

In a typical rodent spatial navigation study, the non-spatial attributes are well-controlled, existing at all locations regardless of the enclosure used during testing (hence, a grid cell in one enclosure will be a grid cell in a different enclosure). Because labs adopt standard procedures, the surfaces, odors (e.g. from cleaning), external lighting, time of day, human handler, electronic apparatus, hunger/thirst state, etc. might be the same for all recording sessions. Additionally, the animal is not allowed to interact with other animals during recording and this isolation may be an unusual and highly salient property of all recording sessions. Notably, the animal is always attached to wires during recording. The internal state of the animal (fear, aloneness, the noise of electronics, etc.) is likely similar across all recording situations and attributes of this internal state are likely represented in the hippocampus and entorhinal input to hippocampus. According to this model, hippocampal place cells are 'marking' all locations in the enclosure as places where these things tend to happen.

In terms of the animal's internal state, all locations in the enclosure may be viewed as equally aversive and unrewarding, which is a memorable characteristic of the enclosure. Reward, or lack thereof, is arguably one of the most important non-spatial characteristics and application of this model to reward might explain the existence of goal-related activity in place cells (*Hok et al., 2007*; although see *Duvelle et al., 2019*), reflecting the need to remember rewarding locations for goal directed behavior. Furthermore, if place cell memories for a rewarding location activate entorhinal grid cells, this may explain the finding that grid cells remap in an enclosure with a rewarded location such that firing fields are attracted to that location (*Boccara et al., 2019*; *Butler et al., 2019*). Studies that introduce reward into the enclosure are an important first step in terms of examining what happens to grid cells when the animal is placed in a more varied environment.

### Key predictions made by this memory model

Future studies should investigate what happens to grid cells when rodents navigate in truly naturalistic settings that include varied objects, varied food sources, varied surfaces, other animals, etc. Do grid cells persist in such a naturalistic environment (perhaps some of them will persist, considering that the animal is still subjected to the electronic recording apparatus and other lab procedures)? Or will some grid cells lose their regularity, or even become silent, considering that there is no longer uniformity to the non-spatial attributes? The results of *Barry et al., 2012* demonstrate that the grid pattern expands and becomes less regular in a novel environment. Nevertheless, the novel environment in that study was uncluttered rather than naturalistic.

A key prediction of this account is that in the absence of memory retrieval from hippocampal place cells, each mEC grid cell should revert to its underlying bottom-up receptive field. For instance, in the absence of top-down memory retrieval feedback, a head direction conjunctive grid cell should become a head direction cell at all positions rather than just at grid field positions. Such a finding was reported by *Bonnevie et al., 2013*. More generally, this kind of result should occur for all grid

cells. For instance, if the bottom-up receptive field for a grid cell is a tuning function for a preferred sound frequency, then that cell should exhibit sound frequency sensitivity in all positions in absence of hippocampal feedback rather than only responding in remembered positions. In brief, this account assumes that *all grid cells are conjunctive grid cells* except that the nature of the non-spatial attribute that makes them conjunctive is not yet known for many grid cells (see *Aronov et al., 2017*, for an example of what might be sound conjunctive grid cells).

This model predicts that place cell fields should be arrayed in a hexagonal grid for situations that produce grid cell firing. This does not mean that *all* place cell fields should be part of the *same* grid. Instead, the key prediction is that *sets* of place cells should be arrayed in a grid where a set of place cells is defined as place cells that represent the same attribute in conjunction with location information (e.g. all places with a particular sound). The attribute that links a set of place cells could be spatial, such as with head direction (i.e. a remembered viewpoint at all locations), and one way to test this idea would be to find sets of place cells that are sensitive to the same head direction, with a test of whether their corresponding place fields are arrayed in a grid. A different approach to testing this prediction could determine co-activity between grid cells and place cells (e.g., find the set of place cells that are co-active with a particular grid cell). Once a set of place cells is identified in this manner, the place field centers of the set should be arrayed in the same manner as the grid cell used to identify the set. In other words, there should be a one-to-one correspondence between each grid field of a particular grid cell and a corresponding place field.

The one-to-one correspondence grid fields and place fields arises from hippocampal feedback in the model. In other words, the place fields are the cause of the grid fields. However, the model is bidirectional and recurrent. For instance, memory consolidation of the place cells arises from cycles of bottom-up and top-down activation between entorhinal cells and hippocampal place cells. As such, changes in the properties of the grid cells may entail changes in the properties of place cells. This recurrence might explain the finding that manipulations of grid spacing are associated with an increase in the size of place fields (*Mallory et al., 2018*). For this memory model, place cell field size directly relates to the stability of place cells. If the place cells are actively moving their place centers with ongoing consolidation (e.g. moving to acquire a larger spacing between memories), they will appear to have larger place fields as they shift position. In the simulation results, this occurred with a novel enclosure (see for instance the larger grid field sizes, which arise from unstable place cells, for the novel network results in *Figure 6*, *Figure 7*), and to a lesser extent with remapping *Appendix 1—figure 3*.

According to this model, hexagonally arranged grid cells should be the exception rather than the rule when considering more naturalistic environments. In a more ecologically valid situation, such as with landmarks, varied sounds, food sources, threats, and interactions with conspecifics, there may still be remembered locations were events occurred or remembered properties can be found, but because the non-spatial properties are non-uniform in the environment, the arrangement of memory feedback will be irregular, reflecting the varied nature of the environment. This may explain the finding that even in a situation where there are regular hexagonal grid cells, there are often irregular non-grid cells that have a reliable multi-location firing field, but the arrangement of the firing fields is irregular (*Diehl et al., 2017*). For instance, even when navigating in an enclosure that has uniform properties as dictated by experimental procedures, they may be other properties that were not well-controlled (e.g. a view of exterior lighting in some locations but not others), and these uncontrolled properties may produce an irregular grid (i.e. because the uncontrolled properties are reliably associated with some locations but not others, hippocampal memory feedback triggers retrieval of those properties in the associations locations).

In this memory model, there are other situations in which an irregular but reliable multi-location grid may occur, even when everything is well controlled. In the reported simulations, when the hippocampal place cells were based on variation in X/Y (as defined by Border cells), nothing else changed as a function of location, and the model rapidly produced a precise hexagonal arrangement of hippocampal place cell memories. When head direction was included (i.e. real-world variation in X, Y, and head direction), the model still produced a hexagonal arrangement as per face-centered cubic packing of memories, but this precise arrangement was slower to emerge, with place cells continuing to shift their positions until the borders of the enclosure were sufficiently well learned from multiple viewpoints. If there is real-world variation in four or more dimensions, as is likely the case in a more

ecologically valid situation, it will be even harder for place cell memories to settle on a precise regular lattice. Furthermore, in the case of four dimensions, mathematicians studying the 'sphere packing problem' recently concluded that densest packing is irregular (*Campos et al., 2023*). This may explain why the multifield grid cells for freely flying bats have a systematic minimum distance between firing fields, but their arrangement is globally irregular (*Ginosar et al., 2021*). Assuming that the memories encoded by a bat include not just the three real-world dimensions of variation, but also head direction, the grid will likely be irregular even under optimal conditions of laboratory control.

This memory model predicts that grid-like responses do not require spatial navigation for their existence. Instead, grid-like responses might occur in any situation involving a highly familiar two-dimensional space in which nothing of interest varies aside from the two dimensions. In such cases, the memories that represent points in the two-dimensional space arrange themselves into an equally spaced hexagonal pattern owing to consolidation and pattern separation. If this pattern separation consolidation process is a general property of cortical neurons, grid-like responses may be found throughout the cortex. This might explain why well-learned non-spatial two-dimensional spaces such as cartoon bird neck/leg length (*Constantinescu et al., 2016*) and pine/banana odors (*Bao et al., 2019*) produce grid-like responses in areas other than the MTL. It might also explain why visual search in the two dimensions of the visual field can produce grid-like responses in humans (*Julian et al., 2018*) and primates (*Killian et al., 2012*).

## Other grid cell models assume that spatial grid cells have a spatial function

The regularity of grid cell responses is fascinating and unparalleled in neuroscience – the existence of the grid pattern assuredly tells us something important about the operation of the nervous system and, correspondingly, there have been dozens of models aimed at explaining how the regular grid pattern emerges. These models are briefly considered to highlight the most important differences between them and this memory model of grid cells.

The first wave of grid cell models built the grid cell responses in a purely bottom-up fashion, proposing that place cells learn their positions as defined by a population code of grid cells (*Hasselmo, 2009*; *Mhatre et al., 2012*; *Solstad et al., 2006*). Subsequent work called into question these models (although see *Lian and Burkitt, 2022*), based on the finding that inactivation of the hippocampus eliminates grid responses (*Bonnevie et al., 2013*) and the finding that place cells exist before grid cells during development (*Bjerknes et al., 2014*). To accommodate such results, a second wave of grid cell models assumed that place cells serve as a guide for the development/emergence of grid cell responses (*Castro and Aguiar, 2014*; *Stepanyuk, 2015*; *Widloski and Fiete, 2014*). However, it is less clear why the grid cells are needed in these models, considering that place cells could directly provide spatial knowledge, although some have proposed that grid cells are useful for charting novel navigational paths (*Bellmund et al., 2016*; *Bush et al., 2015*; *Sorscher et al., 2023*). Another class of grid cell models propose that the grid emerges from intrinsic attractor dynamics within entorhinal cortex (*Burak and Fiete, 2009*; *Couey et al., 2013*; *Fuhs and Touretzky, 2006*; *Guanella et al., 2007*; *McNaughton et al., 2006*), explaining why the grid can exist even during sleep (*Gardner et al., 2019*). However, in these models, the function of entorhinal grid cells is nevertheless assumed to support spatial representations.

Recently, there are several models that take a broader information-theoretic view of the MTL by asking what kinds of representations are useful for representing different kinds of spaces in a parsimonious manner (*Bellmund et al., 2018*; *Mok and Love, 2019*; *Rodríguez-Domínguez and Caplan, 2019*; *Wei et al., 2015*), or useful for predictive coding (*Stachenfeld et al., 2017*). These models can capture some of the grid cell results presented in the current simulations, including extension to non-spatial grid-like responses (e.g. grid field that cover a two-dimensional neck/leg length bird space). Furthermore, these models may be able to explain memory phenomena similar to the model proposed in this study. However, unlike the proposed model, these models assume that the function of entorhinal grid cells that exhibit spatial X/Y grid fields during navigation is to represent space. In contrast, the memory model proposed in this study assume that the function of spatial X/Y grid cells is to represent a non-spatial attribute; the only reason they exhibit a spatial X/Y grid is because memories of that non-spatial attribute are arranged in a hexagonal grid owing to the uncluttered/unvarying nature of the enclosure. Thus, these model do not explain why most of the input to rodent

hippocampus appears to be spatial (*Boccara et al., 2010*; *Diehl et al., 2017*; *Grieves and Jeffery, 2017*), whereas the proposed model can explain this situation as reflecting the miss-classification of grid cells with a spatial arrangement as providing spatial input to hippocampus.

## Conclusions

Returning to the original question posed in this study – how can the function of the MTL be the creation and retrieval of episodic memories if most of the cells in MTL are spatial? The proposed theory and model present a potential answer. The appearance of rodent MTL as primarily concerned with navigation rather than memory may be spurious if grid cells have been misclassified as spatial. Instead, place cells may be 'memory cells' that combine spatial location with many other dimensions into complex conjunctions representing episodic memories, and the precise hexagonal firing pattern of grid cells might reflect memory encoding for non-spatial attributes ('what') that are found throughout the two-dimensional surface.

## Acknowledgements

I thank Trygve Solstad and Rosie Cowell for many discussions during the development of this model, Nina Kazanina for providing feedback on a draft of this report, Josh Jacobs for creating an initial version of *Figure 1A*, and Tim Xia for his work reanalyzing the data of *Jercog et al., 2019*.

## Additional information

### Funding

No external funding was received for this work.

### Author contributions

David E Huber, Conceptualization, Software, Formal analysis, Investigation, Visualization, Writing – original draft, Project administration, Writing – review and editing

### Author ORCIDs

David E Huber ⓘ https://orcid.org/0000-0002-7709-7993

Reviewer #1 (Public review): https://doi.org/10.7554/eLife.95733.3.sa1
Reviewer #3 (Public review): https://doi.org/10.7554/eLife.95733.3.sa2
Author response https://doi.org/10.7554/eLife.95733.3.sa3

## Additional files

### Supplementary files

MDAR checklist

### Data availability

Computer code for simulating the model and two videos showing model behavior are linked in Figures 2, 4, and 9, respectively. The code has been archived at Zenodo (*Huber, 2025*).

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

# Appendix 1

## Place cell results

### Head direction sensitivity of place cells depends on position relative to borders and place field center

For the parameter values used in *Figures 8–10*, the place cells were fairly sensitive to head direction. When including head direction in the place cell memories, it may be that different parameter values produce place cells that are less sensitive to head direction. If the activation/consolidation threshold parameters are low enough, a hippocampal place cell can become active solely based on location, regardless of head direction. To examine situations with relatively low head direction sensitivity, a new simulation was run by setting the consolidation threshold and activation thresholds to the widely spaced grid values from *Figure 6*, *Figure 7* but with the inclusion of head direction in the hippocampal place cell memories.

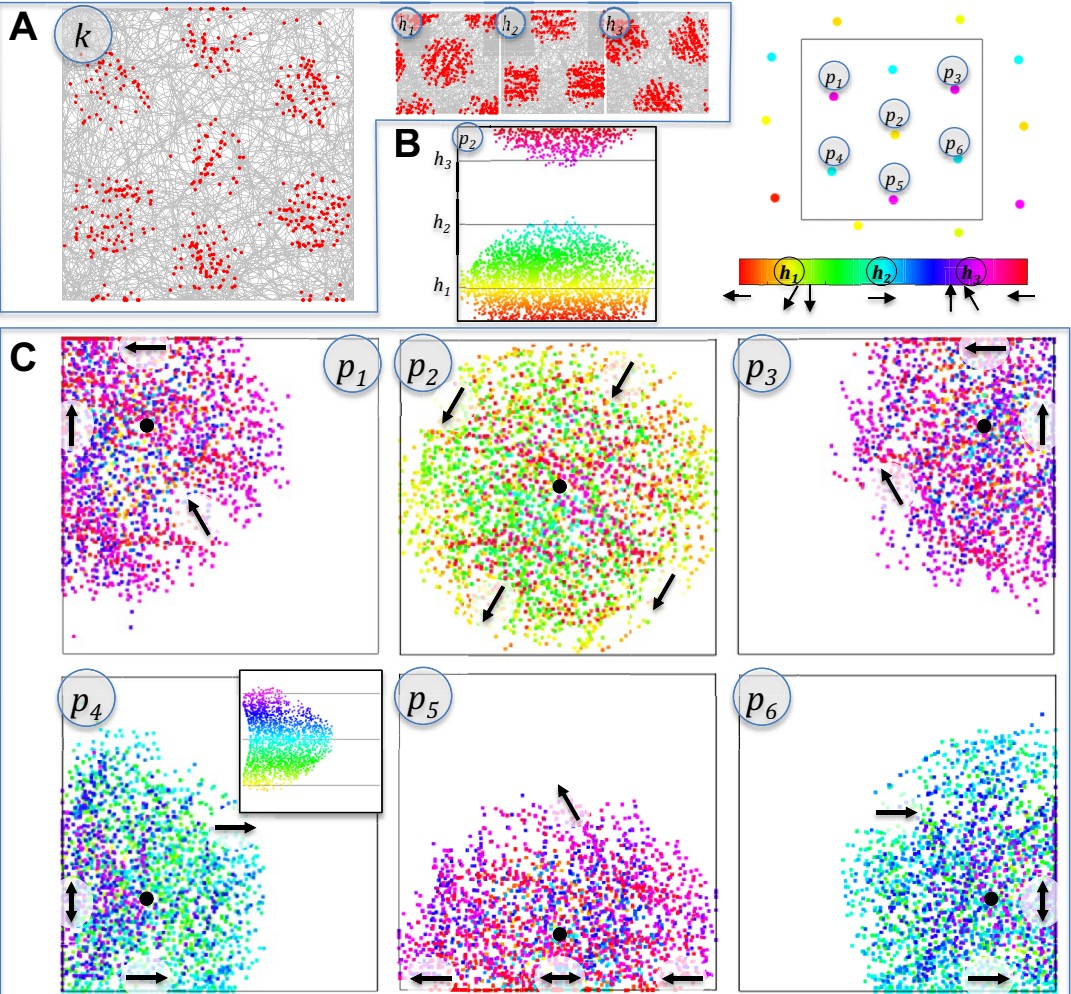

**Appendix 1—figure 1.** Simulation results using the wide-spacing activation/consolidation threshold parameter values from *Figure 6/ Figure 7* but when including head direction in the hippocampal place cell memories. (**A**) The non-spatial attribute cell ($k$) and head direction cells ($h_1 – h_3$) produced similar grid field patterns to those seen in *Figure 9/ Figure 10*. (**B**) The right-hand graph shows the final preferred positions of hippocampal place cells, with color indicating preferred head direction. The wider spacing was satisfied through three rather than six layers along the head direction dimension (only three different colors). Six interior place cells ($p_1 – p_6$) are selected for additional analyses. The lower-left graph plots x-position by head direction for the firing map of cell $p_2$, collapsing over y-position: the cell was active for a full 2/3 of head directions at its place field center. The color of each spike indicates the head direction of the spike according to the color map. (**C**) Head direction sensitivity
*Appendix 1—figure 1 continued on next page*

*Appendix 1—figure 1 continued*

of the place cells (shown in six firing map plots for each of six place cells) for the six interior place cells. The black dots show approximate place field centers, and the black arrows show approximate head direction sensitivity in different locations. For cell $p_2$ head direction sensitive increases with distance from the center. For the other five cells, a different pattern emerges along the borders of the box. Along the borders, some head directions are not empirically observed, as shown by the inset graph for cell $p_4$, which plots x-position by head direction. The absence of data for a subset of head directions gives the spurious appearance of head direction sensitivity that is in line with the borders in one direction or the other, or possibly both directions.

As seen in *Appendix 1—figure 1A*, which shows results from a familiar environment, the firing maps for the non-spatial attribute cell (a.k.a. grid cell) and the three head direction cells were very similar to those seen in *Figures 8–10* despite the larger grid spacing that arises from a lower consolidation threshold. The pressure to place memories farther apart was satisfied by arranging the place cell memories into three face-centered cubic packing hexagonal layers rather than six layers. In other words, this change of parameter values resulted in a wider spacing between the memory layers along the head direction dimension. This is shown in *Appendix 1—figure 1B*, where the plot of the final positions of the memories are non-overlapping rather than producing two oppositely oriented memories in an allocentric pair (the simulation resulted in 45 memories, although the exterior place cell memories are clipped off in the graph to make it easier to see the face-centered cubic packing of the interior 17 memories).

These results demonstrate that the head direction grid module can emerge with other parameter settings. Furthermore, these results indicate that there is more than one way in which an allocentric cognitive map can emerge – even though this simulation did not produce allocentric pairings of oppositely oriented place cells in the same position, the interior place cells were largely insensitive to head direction when the animal was in the exact center of the place field. This is shown in the graph labeled $p_2$ in *Appendix 1—figure 1B*, which plots the head direction of each spike for place cell $p_2$ as a function of x-position, collapsing over y-position. This particular cell preferred the center of the enclosure and responded to a full 2/3 of the possible head directions at the place field center. This occurred even though the head direction of the memory was in direction $h_1$ (the yellow color on the head direction color map).

*Appendix 1—figure 1C* plots the firing maps for six of the interior place cells, using the labels from *Appendix 1—figure 1B*, with head direction coded by color. The black dots show the approximate place field centers for each place cell, and it is at these positions that the cell is relatively insensitive to head direction (i.e. responding to a full 2/3 of possible head directions). But for positions farther from the black dots, the place cell becomes more selective in its head direction, provided that the position is not along a border (discussed below). In other words, these place cells are active if the animal is in the preferred location regardless of head direction, or active for non-preferred positions provided that the head direction matches the preferred direction.

The situation near the borders is more complicated owing to missing data (*Muller et al., 1994*; *Peyrache et al., 2017*). Specifically, there is often a failure to observe head directions pointing towards or away from border positions if the animal tends to run alongside the borders (this happens in the simulation owing to momentum). The problem of missing data is highlighted with the black arrows in *Appendix 1—figure 1C*, which show head direction sensitivity in different locations of the place field. For instance, place cell $p_4$ appears to prefer both up and down head directions along the West border even though the preferred head direction of this place cell memory was to the East. This occurred because head direction towards the West was never observed at the West border. The inset graph shows the same firing map, but with axes of x-position and head direction, collapsing over y-position, highlighting the absence of East-facing head direction samples at the lowest (i.e. Westernmost) x-positions.

The complicated nature of hippocampal place cell head direction sensitivity seen in the simulation results in *Appendix 1—figure 1* may provide an alternative explanation of the results reported by *Jercog et al., 2019*, which were taken to indicate that head direction sensitivity of hippocampal place cells is relative to a reference point. The reference point model in that study described a number of different behaviors of place cells, including: (1) place cells that seemed to prefer the animal heading towards or away from the place field center; (2) place cells that preferred the animal heading in a circular direction clockwise or counterclockwise around the pace field; and (3) place

cells that preferred the animal heading in a particular direction that was far outside the enclosure. Some of these behaviors are approximated by the simulation results in *Appendix 1—figure 1*, and future work could directly compare this account to the reference point model.

## Head direction sensitivity of place cells depends on enclosure geometry

As reported in *Figure 10B*, the outside-the-box exterior place cells are more selective for head direction considering that the animal is not allowed to visit these cells' preferred locations: that is, the animal is only ever located in the periphery of these cells' true place fields, where head direction sensitivity is greater. This is equally true for the simulation in *Appendix 1—figure 1*. This aspect of model may help explain the empirical result that place cells seem to exhibit greater head direction sensitivity when the animal navigates a familiar narrow passage, such as occurs in a radial arm maze or an elevated track (*McNaughton et al., 1983*; *Muller et al., 1994*). Using the same parameters as *Appendix 1—figures 2* reports the results of an animal navigating a familiar but extremely narrow enclosure (only 5% as tall as it is wide). In this case, nearly all place cells (14 of the 15 memories) were exterior place cells and, as a result, nearly all place cells exhibited strong head direction sensitivity, preferring travel in one direction or the other along the narrow passage (blue-East or red-West), but not in both directions. In brief, place fields for narrow passages have greater head direction sensitivity because the 'true' place field center has been consolidated to a position outside the passage.

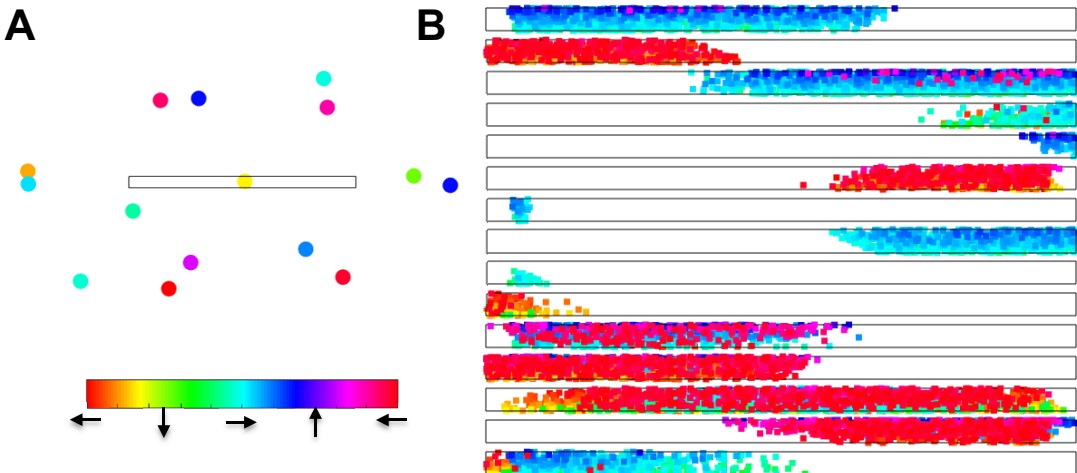

**Appendix 1—figure 2.** Place cell responses using same parameters as *Appendix 1—figure 1* but with an enclosure that is 5% as tall as it is wide, simulating behavior in a highly familiar narrow passage, such as an arm of a radial arm maze or an elevated track. (**A**) The positions of all 15 consolidated memories are shown, with the color indicating preferred head direction according to the color map. (**B**) The firing maps show the place field for each place cell with spike color indicating head direction. Even though these same parameter values produced place cells that were relatively insensitive to head direction in the open field enclosure of *Appendix 1—figure 1*, in this case each place cell appeared to only prefer one direction or the other (blue or red) along the narrow passage.

As seen in *Appendix 1—figure 2*, because all but one of the place cells was exterior when the simulated animal was constrained to a narrow passage, the hippocampal place cell memories were no longer arranged in a hexagonal grid. This disruption of the grid array for narrow passages might explain the finding that the grid pattern (of grid cells) is disrupted in the thin corner of a trapezoid (*Krupic et al., 2015*) and disrupted when a previously open enclosure is converted to a hairpin maze by insertion of additional walls within the enclosure (*Derdikman et al., 2009*).

## Remapping of place cells with changes in enclosure geometry

It is beyond the scope of this study to provide a full model of the situations that cause hippocampal place cells to remap their place field positions (*Geva-Sagiv et al., 2016*; *O'Keefe and Nadel, 1979*) or keep their place field positions but change their firing rate (*Leutgeb et al., 2005*). However, because this is a memory model, it holds the potential to explain such effects (see *Sanders et al., 2020* for a recent review of the remapping literature and application of a belief state model). According to the proposed memory account of place cells, a similar enclosure or a similar context (i.e. non-

spatial attributes) as compared to prior experiences can trigger memory retrieval (see *Maurer and Nadel, 2021* for a similar proposal). For instance, similar locations might trigger memory retrieval, but perhaps to a lesser degree (i.e. rate remapping) or perhaps the set of retrieved memories might entail a completely different arrangement if those memories occurred in a similar context but with a different enclosure (i.e. global remapping). Furthermore, as the new enclosure/context becomes familiar, new memories (place fields) may be created and the positions of old memories may be changed owing to consolidation and pattern separation.

As currently implemented, the only non-spatial contextual input to the model is the non-spatial grid cell (the $k$ cell). Future work could expand this non-spatial input by including basis sets of different contextual inputs (e.g. a basis set for sound, $k_1$-$k_3$, a basis set for surface texture, $l_1$-$l_3$, etc.), in which case the model could apply to situations where the geometry and location of the enclosure are kept the same, but the non-spatial characteristics are changed. Nonetheless, the current model can be applied in its current form to changes in enclosure geometry, which has been found to produce both rate remapping (*Leutgeb et al., 2005*) as well global remapping, depending on how drastically the geometry is changed (*Wills et al., 2005*).

Using the same parameter values as for *Figures 8–10*, *Appendix 1—figure 3A* shows the grid cell firing map ($k$), hippocampal place cell memory positions (color coded for head direction), head direction grid module firing map ($h_1$-$h_3$), and border cell firing maps ($e_1$-$e_3$, $f_1$-$f_3$, and $g_1$-$g_4$) for a recording after 10 sessions of prior experience with a circular enclosure. Critically, the same non-spatial attribute is found for the familiar circular enclosure and the novel square enclosure and both enclosures have same width/height. This situation might correspond to several days of testing with the circular enclosure followed by testing in a square enclosure that is placed in the same broader context (i.e. the same experimental testing room) as prior testing. Prior to the change, it can be seen that the familiar circular enclosure produced a six-layer face-centered head direction grid module, similar to the results reported in *Figures 8–10*. The border cell firing maps are shown to highlight that as currently configured, the model tends to produce somewhat patchy border cell responses (i.e. multiple separated hotspots along a line), particularly for the interior border cells ($e_2$, $f_2$, and $g_2$). This occurs because the border cells receive stronger memory retrieval feedback in the locations of hippocampal place cell memories. This behavior is parameter dependent; for instance, border cells with a higher firing rate will be more 'filled in' between the hotspots where memory feedback occurs.

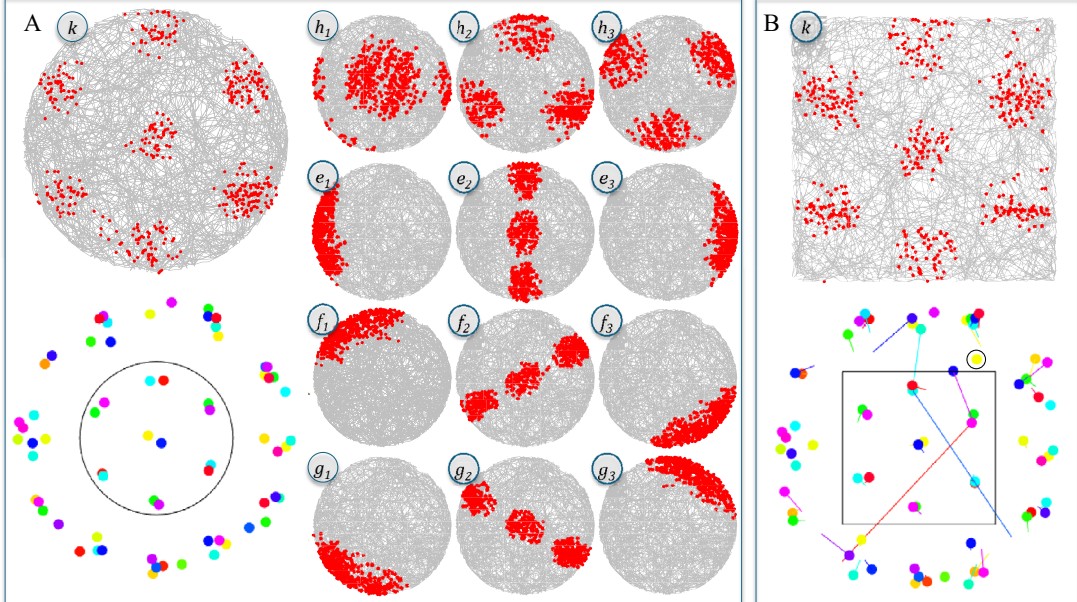

**Appendix 1—figure 3.** Simulation results for a familiar circular enclosure followed by a change to a square enclosure placed in the same context, containing the same non-spatial attribute. (**A**) The memory locations (dots colored by head direction) and firing maps for the familiar circular enclosure show a six-layer face-centered cubic packing for the head direction grid module (**h₁–h₃**). The border cells ($e_1$-$e_3$, $f_1$-$f_3$, and $g_1$-$g_4$) captured one side or the other except for the between-border cells, which show a patchy response owing to memory feedback. (**B**) Upon

*Appendix 1—figure 3 continued*

entry into a novel square enclosure that contained the same non-spatial attribute as the circular enclosure and was the same width/height as the circular enclosure, the head direction grid module was immediately apparent (in contrast to the results reported in *Figure 9*). This occurred because the memories created in the circular environment were recalled in the square environment and the geometry was similar enough that very few of the retrieved memories changed their locations. The 'tails' emanating from each place cell memory location show the shift from the prior location in the circular enclosure to the new location in the square enclosure. The color of the tail indicates the prior head direction of the memory if head direction of the memory changed with consolidation (e.g. the blue memory on the upper border has a pink tail). As shown, there was a subtle change in the positions of the exterior place cells to accommodate the novel square shape. Two exterior place cells migrated to interior positions (long red and blue tails) and, correspondingly, two interior place cells migrated to exterior positions (cyan and pink tails). The length of the tails is sometimes misleading owing to the circular nature of the border cells (in truth, the two very long red and blue tails migrated in a wraparound fashion, which was a much smaller change). Finally, one new memory was formed (yellow memory encircled in black).

*Appendix 1—figure 3B* is a recording of the same simulated animal after switching to a novel square enclosure that contains the same non-spatial attribute and has the same width/height as the familiar circular enclosure. Of note, experiments that switched animals between square and circular enclosures have produced global remapping and, correspondingly, rotation of the grid cell responses (*Fyhn et al., 2007*). However, to keep things simple in this simulation, it was assumed that the border cells lined up in the same manner for both enclosures (i.e. the *E* dimension was East-West for both enclosures), such as might occur if the animal was primarily attuned to a salient characteristic of the testing room rather than the enclosure walls. But if the border cells had changed their alignment with the new enclosure (e.g. if the *E* border dimension aligned with the North-South borders), then the place cells would remain in their same positions relative to the now-rotated borders (i.e. no remapping relative to the enclosure) and the corresponding grid cells would also retain their same alignment relative to the enclosure.

As seen in the non-spatial grid cell firing map and in the memory locations at the end of the recording session, the head direction grid module was immediately apparent in this novel environment (unlike the results plotted in *Figure 9*, for a 'blank slate' simulated animal with no prior experience with any similar enclosures). Thus, the immediate appearance of head direction grid modules may reflect memory retrieval of similar enclosures. In brief, because the shape of the novel square enclosure was similar to the circular enclosure (i.e. of similar height and width), and because both enclosures were in the same experimental room and in the same position, and because both enclosure contained the same non-spatial attribute, the exterior place cells that were learned for the circular enclosure were readily adapted to make sense of the square enclosure, and this gave stability to the hexagonal 3D arrangement of the hippocampal place cells. In other words, this situation is analogous to rate remapping results in which previously learned memories from similar enclosures placed in the same global context produce the same place fields, but with different firing rates (*Leutgeb et al., 2005*).

The current simulation did not actually produce a substantial change in firing rates of the place cells considering that the non-spatial context of the situation was assumed to be identical between the circular enclosure and the change to the square enclosure. Despite the adaptation of memories from the circular enclosure to the novel square enclosure, there were some changes. More specifically, 12 of the 14 interior place cells remained in their same positions. However, two of the previously interior place cells moved to the exterior (the cyan and pink tails) and, correspondingly, two of the previously exterior place cells moved to the interior, filling the now vacant memory locations (red and blue tails, which actually took the shorter path wrap-around direction, rather than the long paths shown in the figure). In addition, one new memory was formed (the encircled yellow dot). Another important aspect of this partial remapping is the set of subtle changes in the exterior place cells to accommodate the square shape.

Using the same parameters and same situation of recording in a familiar enclosure before a change to a novel enclosure placed in the same global context and containing the same non-spatial attribute, *Appendix 1—figure 4* shows a simulation that is more accurately be described as global remapping (*Geva-Sagiv et al., 2016*; *O'Keefe and Nadel, 1979*). In this case, the familiar enclosure was a rectangle that was only 25% as tall as the square. As seen in *Appendix 1—figure 4A*, the exterior place cells conformed to the shape of the rectangle. When moved to the square enclosure (*Appendix 1—figure 4B*), this gave the animal access to previously unexplored regions

within the global context (the regions above the North wall of the rectangular enclosure and below the South wall of the rectangular enclosure). As a result of learning and consolidation for these previously unexplored regions, the exterior place cells above and below the remembered enclosure were pushed a substantial distance to accommodate the larger square shape. Furthermore, four new interior memories were created (as well as one new exterior memory) and some of the interior memories changed from one interior position to a different interior position. By the end of the recording in the novel square enclosure, the memories formed into a six-layer face-centered head direction grid module, demonstrating that even with global remapping, memories from other enclosures can be adapted to provide a rapid understanding of a novel enclosure. However, in this case, this occurred in combination with a greater degree of new memory formation and more radical changes in the positions of place cells than the remapping that occurred with the change from a circular enclosure to a square enclosure.

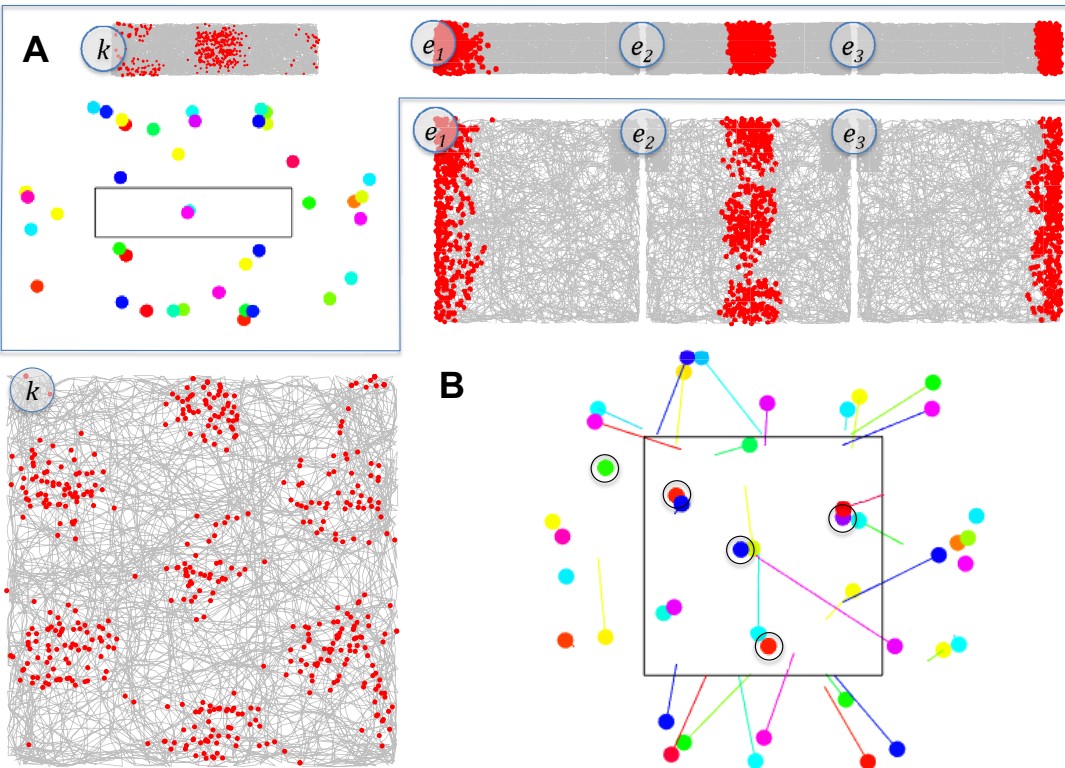

**Appendix 1—figure 4.** Simulation results for a familiar rectangular enclosure followed by a change to a square enclosure placed in the same context, containing the same non-spatial attribute. (**A**) As with the narrow passage in *Appendix 1—figure 2*, the familiar rectangular enclosure was too narrow to produce a highly regular hexagonal grid. (**B**) In switching to the much larger square enclosure, most place cells changed their locations (global remapping) and five new memories were formed. Despite the considerable disruption in the place fields, the memories from the rectangular enclosure allowed for the rapid establishment of the hexagonal grid, including head direction conjunctive grid cells (see *Appendix 1—figure 3* caption for additional information).

Finally, one notable behavior of the border cells in *Appendix 1—figure 4* is that East/West border cells appeared to elongate their responses when the East and West borders were elongated to create the novel square enclosure. Exactly this result has been reported in the literature (*Solstad et al., 2008*), and in the model it occurs because the border cells are more accurately described as an allocentric playing field in which different enclosures can be represented, with a tendency for the playing field to align with borders. In other words, the border cells define a toroidal space (a donut) on which knowledge for the enclosure can be created (e.g. placing memory 'sprinkles' on the donut in the shape of the enclosure).

