## [Editor Report · eLife Assessment]

This **important** paper provides **solid** evidence for an alternative conceptualization of the functional role of the place and grid cell network in the medial temporal lobe for memory as opposed to spatial processing or navigation. The theory is extensive, tightly integrating data on various spatial cell types. It accounts for many experimental results and generates strong predictions for future studies that will be of interest to researchers in this field. The impact of the work would be strengthened if future experiments reveal that grid cells do indeed encode specific nonspatial features.

---

## [Referee Report · Reviewer #1 (Public review)]

Huber proposes a theory where the role of the medial temporal lobe (MTL) is memory, where properties of spatial cells in the MTL can be explained through memory function rather than spatial processing or navigation. Instantiating the theory through a computational model, the author shows that many empirical phenomena of spatial cells can be captured, and may be better accounted through a memory theory. It is an impressive computational account of MTL cells with a lot of theoretical reasoning and aims to tightly relate to various spatial cell data.

In general, the paper is well written, and has been greatly improved after revision for clarity and situating the model in the context of the literature. Below are a few responses to the author's rebuttal.

(2 & 3) In response to my previous review point 2 and 3, the author has now added "According to this model, hexagonally arranged grid cells should be the exception rather than the rule when considering more naturalistic environments." It is good to know that it captures data that show non-grid like responses in more complex and realistic environments. However, the model still focuses on explaining the spatial firing aspect of grid cells even though they are not supposed to be spatial. I noted in my previous review, "If it's not encoding a spatial attribute, it doesn't have to have a spatial field. For example, it could fire in the whole arena". The author notes inhibitory drive and habituation. Habituation happens, but then spatial cell responses are supposed (or assumed) to be still strong after many visits to that environment. More generally, I am more convinced that grid-like and spatial coding are a special case - both in navigation and memory. In a way I believe the author agrees, though the work here focuses on capturing spatial properties (which is understandable given the literature). In conclusion, though there may be theoretical disagreements, I find the points the author raises fair.

(4) The difference between mEC and lEC or PRC for encoding non-spatial vs spatial attributes is still not clear to me - though not crucial for the point of this paper.

(5) Thank you for providing a video - this makes it extremely clear how learning occurs.

---

## [Referee Report · Reviewer #3 (Public review)]

The author presents a novel theory and computational model suggesting that grid cells do not encode space, but rather encode non-spatial attributes. Place cells in turn encode memories of where those specific attributes occurred. The theory accounts for many experimental results and generates useful predictions for future studies. The model's simplicity and potential explanatory power will interest others in the field. There are, however, a few weaknesses outlined below which undermine the theory.

Main criticisms:

(1) A crucial assumption of the model is that grid cells express grid-like firing patterns if and only if the content of experience is constant in space. It is difficult to imagine a real world example that satisfies this assumption. Odors and sounds are used as examples. While they are often more spatially diffuse than an object on the ground, odors and sounds have sources that are readily detectable and thus are not constant in space. Animals can easily navigate to a food source or to a vocalizing conspecific. This assumption is especially problematic because it predicts that all grid cells should become silent when their preferred non-spatial attribute (e.g. a specific odor) is missing. I'm not aware of any experimental data showing that grid cells become silent. On the contrary, grid cells are known to remain active across all contexts that have been tested, including across sleep/wake states. Unlike place cells, grid cells have never been shown to turn off. Since grid cells are active in all contexts, their preferred attribute must also be present in all contexts, and therefore they would not convey any information about the specific content of an experience. The author lists many attributes that could in theory be constant in a laboratory setting, but there is no data I'm aware of that shows this is true in practice. As it stands, this crucial assumption of the model remains mere speculation.

(2) The proposed novelty of this theory is that other models all assume that grid cells encode space. This is not quite true of models based on continuous attractor networks, the discussion of which is essentially absent. More specifically, attractor models focus on the importance of intrinsic dynamics within entorhinal cortex in generating the grid pattern. While this firing pattern is aligned to space during navigation and therefore can be used a representation of that space, the neural dynamics are preserved even during sleep. Similarly, it is because the grid pattern does not strictly encode physical space that grid-like signals are also observed in relation to other two-dimensional continuous variables.

(3) The use of border cells or boundary vector cells as the main (or only) source of spatial information in the hippocampus is not well supported by experimental data. Border cells in entorhinal cortex are not active in the center of an environment. Boundary-vector cells can fire farther away from the walls, but are not found in entorhinal cortex. They are located in the subiculum, a major output of the hippocampus. While the entorhinal-hippocampal circuit is a loop, the route from boundary-vector cells to place cells is much less clear than from grid cells. Moreover, both border cells and boundary-vector cells (which are conflated in this paper) comprise a small population of neurons compared to grid cells.

Minor comments:

(1) There is substantial theoretical and experimental work supporting the idea that grid cell modules instantiate continuous attractor networks, yet this class of models is largely ignored:

p. 7 "In contrast, most grid cell models (Bellmund et al., 2016; Bush et al., 2015; Castro & Aguiar, 2014; Hasselmo, 2009; Mhatre et al., 2012; Solstad et al., 2006; Sorscher et al., 2023; Stepanyuk, 2015; Widloski & Fiete, 2014) are domain specific models of spatial navigation"

The following references should be added:

McNaughton, B. L., Battaglia, F. P., Jensen, O., Moser, E. I. & Moser, M.-B. Path integration and the neural basis of the 'cognitive map'. Nat. Rev. Neurosci. 7, 663-678 (2006).

Fuhs, M. C. & Touretzky, D. S. A spin glass model of path integration in rat medial entorhinal cortex. J. Neurosci. 26, 4266-4276 (2006).

Burak, Y. & Fiete, I. R. Accurate path integration in continuous attractor network models of grid cells. PLoS Comput. Biol. 5, e1000291 (2009).

Guanella, A., Kiper, D. & Verschure, P. A model of grid cells based on a twisted torus topology. Int. J. Neural Syst. 17, 231-240 (2007).

Couey, J. J. et al. Recurrent inhibitory circuitry as a mechanism for grid formation. Nat. Neurosci. 16, 318-324 (2013).

(Note: the Bellmund et al. (2016) citation is likely a mistake and was intended to be Bellmund et al. (2018).)

(2) The author claims in two places that this model is the first to explain that grid cell population activity lies on a torus. While it may be the first explicit computational account of why grid cell activity is mapped onto a torus, these claims should be moderated for clarity, for example by adding "but see McNaughton et al. (2006) and others".

Box 1. Results Uniquely Explained by this Memory Model - the population code of grid cells lies on a torus

p.11 "In addition, this simplifying assumption allows the model to capture the finding that the population of grid cells lies on a torus (Gardner et al., 2022), although I note that the model was developed before this result was known."

(3) Lateral entorhinal cortex is largely ignored in this memory model. It should be considered that the predominance of spatial representations reported in the literature is due to historical reasons. Namely, the discovery of hippocampal place cells spurred interest in looking upstream for the source of spatial information, which was later abundantly clear in medial entorhinal cortex. Lateral entorhinal cortex is relatively understudied, but is known to encode odors, objects, and time in a way that medial entorhinal cortex does not. It is therefore confusing to assume that these attributes are instead encoded by grid cells.

---

## [Author Response]

The following is the authors’ response to the original reviews.

**Public Reviews**

**Reviewer #1 (Public Review):**
(1) Although the theory is based on memory, it also is based on spatially-selective cells.Not all cells in the hippocampus fulfill the criteria of place/HD/border/grid cells, and place a role in memory. E.g., Tonegawa, Buszaki labs' work does not focus on only those cells, and there are certainly a lot of non-pure spatial cells in monkeys (Martinez-Trujillo) and humans (iEEG). Does the author mainly focus on saying that "spatial cells" are memory, but do not account for non-spatial memory cells? This seems to be an incomplete account of memory - which is fine, but the way the model is set up suggests that *all* memory is, place (what/where), and non-spatial attributes ("grid") - but cells that don't fulfil these criteria in MTL (Diehl et al., 2017, Neuron; non-grid cells; Schaeffer et al., 2022, ICML; Luo et al., 2024, bioRxiv) certainly contribute to memory, and even navigation. This is also related to the question of whether these cell definitions matter at all (Luo et al., 2024). The authors note "However, this memory conjunction view of the MTL must be reconciled with the rodent electrophysiology finding that most cells in MTL appear to have receptive fields related to some aspect of spatial navigation (Boccara et al., 2010; Grieves & Jeffery, 2017). The paucity of non-spatial cells in MTL could be explained if grid cells have been mischaracterized as spatial." Is the author mainly talking about rodent work?

There is a new section in the introduction that deals with these issues, titled ‘Why Model the Rodent Navigation Literature with a Memory Model?’ That section reads:

“Spatial navigation is inherently a memory problem – learning the spatial arrangement of a new enclosure requires memory for the conjunction of what and where. This has long been realized and in the introduction to ‘Hippocampus as a Cognitive Map’, O’Keefe and Nadel (1978) wrote “We shall argue that the hippocampus is the core of a neural memory system providing an objective spatial framework within which the items and events of an organism's experience are located and interrelated” (emphasis added). Furthermore, in the last chapter of their book, they extended cognitive map theory to human memory for non-spatial characteristics. However, in the decades since the development of cognitive map theory, the rodent spatial navigation and human memory literatures have progressed somewhat independently.

The ideas proposed in this model are an attempt to reunify these literatures by returning to the original claim that spatial navigation is inherently a memory problem. The goal of the current study is to explain the rodent spatial navigation literature using a memory model that has the potential to also explain the human memory literature. In contrast, most grid cell models (Bellmund et al., 2016; Bush et al., 2015; Castro & Aguiar, 2014; Hasselmo, 2009; Mhatre et al., 2012; Solstad et al., 2006; Sorscher et al., 2023; Stepanyuk, 2015; Widloski & Fiete, 2014) are domain specific models of spatial navigation and as such, they do not lend themselves to explanations of human memory. Thus, the reason to prefer this model is parsimony. Rather than needing to develop a theory of memory that is separate from a theory of spatial navigation, it might be possible to address both literatures with a unified account.

This study does not attempt to falsify other theories of grid cells. Instead, this model reaches a radically different interpretation regarding the function of grid cells; an interpretation that emerges from viewing spatial navigation as a memory problem. All other grid cell models assume that an entorhinal grid cell displaying a spatially arranged grid of firing fields serves the function of spatial coding (i.e., spatial grid cells exist to support a spatial metric). In contrast, the proposed memory model of grid cells assumes that the hexagonal tiling reflects the need to keep memories separate from each other to minimize confusion and confabulation – the grid pattern is the byproduct of pattern separation between memories rather than the basis of a spatial code.

It is now understood that grid-like firing fields can occur for non-spatial twodimensional spaces. For instance, human entorhinal cortex exhibits grid-like responses to video morph trajectories in a two-dimensional bird neck-length versus bird leg-length space (Constantinescu et al., 2016). As a general theory of learning and memory, the proposed memory model of grid cells is easily extended to explain these results (e.g., relabeling the border cell inputs in the model as neck-length and leg-length inputs). However, there are other grid cell models that can explain both spatial grid cells as well as non-spatial grid-like responses (Mok & Love, 2019; Rodríguez-Domínguez & Caplan, 2019; Stachenfeld et al., 2017; Wei et al., 2015). Similar to this memory model of grid cells, these models are also positioned to explain both the rodent spatial navigation and human memory literatures. Nevertheless, there is a key difference between this model and other grid cell models that generalize to non-spatial representations. Specifically, these other models assume that grid cells exhibiting spatial receptive fields serve the function of identifying positions in the environment (i.e., their function is spatial). As such, these models do not explain why most of the input to rodent hippocampus appears to be spatial (Boccara et al., 2010; Diehl et al., 2017; Grieves & Jeffery, 2017). This memory model of grid cells provides an answer to the apparent paucity of nonspatial cell types in rodent MTL by proposing that grid cells with spatial receptive fields have been misclassified as spatial (they are *what* cells rather than *where* cells) and that place cells are fundamentally memory cells that conjoin *what* and *where*.”

(2) Related to the last point, how about non-grid multi-field mEC cells? In theory, these also should be the same; but the author only presents perfect-look grid cells. In empirical work, clearly, this is not the case, and many mEC cells are multi-field non-grid cells (Diehl et al., 2017). Does the model find these cells? Do they play a different role? As noted by the author "Because the non-spatial attributes are constant throughout the two-dimensional surface, this results in an array of discrete memory locations that are approximately hexagonal (as explained in the Model Methods, an "online" memory consolidation process employing pattern separation rapidly turns an approximately hexagonal array into one that is precisely hexagonal). " If they are indeed all precisely hexagonal, does that mean the model doesn't have non-grid spatial cells?

Grid cells with irregular firing fields are now considered in the discussion with the following paragraphs

“According to this model, hexagonally arranged grid cells should be the exception rather than the rule when considering more naturalistic environments. In a more ecologically valid situation, such as with landmarks, varied sounds, food sources, threats, and interactions with conspecifics, there may still be remembered locations were events occurred or remembered properties can be found, but because the non-spatial properties are non-uniform in the environment, the arrangement of memory feedback will be irregular, reflecting the varied nature of the environment. This may explain the finding that even in a situation where there are regular hexagonal grid cells, there are often irregular non-grid cells that have a reliable multi-location firing field, but the arrangement of the firing fields is irregular (Diehl et al., 2017). For instance, even when navigating in an enclosure that has uniform properties as dictated by experimental procedures, they may be other properties that were not well-controlled (e.g., a view of exterior lighting in some locations but not others), and these uncontrolled properties may produce an irregular grid (i.e., because the uncontrolled properties are reliably associated with some locations but not others, hippocampal memory feedback triggers retrieval of those properties in the associations locations).

In this memory model, there are other situations in which an irregular but reliable multilocation grid may occur, even when everything is well controlled. In the reported simulations, when the hippocampal place cells were based on variation in X/Y (as defined by Border cells), nothing else changed as a function of location, and the model rapidly produced a precise hexagonal arrangement of hippocampal place cell memories. When head direction was included (i.e., real-world variation in X, Y, and head direction), the model still produced a hexagonal arrangement as per face-centered cubic packing of memories, but this precise arrangement was slower to emerge, with place cells continuing to shift their positions until the borders of the enclosure were sufficiently well learned from multiple viewpoints. If there is real-world variation in four or more dimensions, as is likely the case in a more ecologically valid situation, it will be even harder for place cell memories to settle on a precise regular lattice. Furthermore, in the case of four dimensions, mathematicians studying the “sphere packing problem” recently concluded that densest packing is irregular (Campos et al., 2023). This may explain why the multifield grid cells for freely flying bats have a systematic minimum distance between firing fields, but their arrangement is globally irregular (Ginosar et al., 2021). Assuming that the memories encoded by a bat include not just the three real-world dimensions of variation, but also head direction, the grid will likely be irregular even under optimal conditions of laboratory control.”

(3) Theoretical reasons for why the model is put together this way, and why grid cells must be coding a non-spatial attribute: Is this account more data-driven (fits the data so formulated this way), or is it theoretical - there is a reason why place, border, grid cells are formulated to be like this. For example, is it an efficient way to code these variables? It can be both, like how the BVC model makes theoretical sense that you can use boundaries to determine a specific location (and so place cell), but also works (creates realistic place cells).

The motivation for this model is now articulated in the new section, quoted above, titled ‘Why Model the Rodent Navigation Literature with a Memory Model?’ Regarding the assumption that border cells provide a spatial metric, this assumption is made for the same reasons as in the BVC model. Regarding this, the text said: “These assumptions regarding border cells are based on the boundary vector cell (BVC) model of Barry et al. (2006). As in the BVC model, combinations of border cells encode where each memory occurred in the realworld X/Y plane.”. A new sentence is added to model methods, stating: “This assumption is made because border cells provide an efficient representation of Euclidean space (e.g., if the animal knows how far it is from different walls of the enclosure, this already available information can be used to calculate location).”

But in this case, the purpose of grid cell coding a non-spatial attribute, and having some kind of system where it doesn't fire at all locations seems a little arbitrary. If it's not encoding a spatial attribute, it doesn't have to have a spatial field. For example, it could fire in the whole arena - which some cells do (and don't pass the criteria of spatial cells as they are not spatially "selective" to another location, related to above).

Some cells have a constant high firing rate, but they are the exception rather than the rule. More typically, cells habituate in the presence of ongoing excitatory drive and by doing so become sensitive to fluctuations in excitatory drive. Habituation is advantageous both in terms of metabolic cost and in terms of function (i.e., sensitivity to change). This is now explained in the following paragraph:

“In theory, a cell representing a non-spatial attribute found at all locations of an enclosure (aka, a grid cell in the context of this model), could fire constantly within the enclosure. However, in practice, cells habituate and rapidly reduce their firing rate by an order of magnitude when their preferred stimulus is presented without cessation (Abbott et al., 1997; Tsodyks & Markram, 1997). After habituation, the firing rate of the cell fluctuates with minor variation in the strength of the excitatory drive. In other words, habituation allows the cell to become sensitive to changes in the excitatory drive (Huber & O’Reilly, 2003). Thus, if there is stronger top-down memory feedback in some locations as compared to others, the cell will fire at a higher rate in those remembered locations rather than in all locations even though the attribute is found at all locations. In brief when faced with constant excitatory drive, the cell accommodates, and becomes sensitive to change in the magnitude of the excitatory drive. In the model simulation, this dynamic adaptation is captured by supposing that cells fire 5% of the time on-average across the simulation, regardless of their excitatory inputs.”

(4) Why are grid cells given such a large role for encoding non-spatial attributes? If anything, shouldn't it be lateral EC or perirhinal cortex? Of course, they both could, but there is less reason to think this, at least for rodent mEC.

This is a good point and the following paragraph has been added to the introduction to explain that lateral EC is likely part of the explanation. But even when including lateral EC, it still appears that most of the input to hippocampus is spatial.

“One possible answer to the apparent lack of non-spatial cells in MTL is to highlight the role of the lateral entorhinal cortex (LEC) as the source of non-spatial *what* information for memory encoding (Deshmukh & Knierim, 2011). LEC can be contrasted with mEC, which appears to only provide *where* information (Boccara et al., 2010a; Diehl et al., 2017). Although it is generally true that LEC is involved in non-spatial processing, there is evidence that LEC provides some forms of spatial information (Knierim et al., 2014). The kind of non-spatial information provided by LEC appears to be in relation to objects (Connor & Knierim, 2017; Wilson et al., 2013). However, in a typical rodent spatial navigation study there are no objects within the enclosure. Thus, although the distinction between mEC and LEC is likely part of the explanation, it is still the case that rodent entorhinal input to hippocampus *appears* to heavily favor spatial information.”

(5) Clarification: why do place cells and grid cells differ in terms of stability in the model? Place cells are not stable initially but grid cells come out immediately. They seem directly connected so a bit unclear why; especially if place cell feedback leads to grid cell fields. There is an explanation in the text - based on grid cells coding the on-average memories, but these should be based on place cell inputs as well. So how is it that place fields are unstable then grid fields do not move at all? I wonder if a set of images or videos (gifs) showing the differences in spatial learning would be nice and clarify this point.

In this revision, I provide a new video focused on learning of place cell memories that include head direction. This second video is in relation to the results reported in Figure 9. The short answer is that the grid fields for the non-spatial cell are based on the average across several view-dependent memories (i.e., across several place cells that have head direction sensitivity) and the average is reliable even if the place cells are unstable. The text of this explanation now reads:

“Why was the grid immediately apparent for the non-spatial attribute cell whereas the grid took considerable prior experience for the head direction cells? The answer relates to memory consolidation and the shifting nature of the hippocampal place cells. Head direction cells only produced a reliable grid once the hippocampal place cells (aka, memory cells) assumed stable locations. During the first few sessions, the hippocampal place cells were shifting their positions owing to pattern separation and consolidation. But once the place cells stabilized, they provided reliable top-down memory feedback to the head direction cells in some places but not others, thus producing a reliable grid arrangement to the firing maps of the head direction cells. In other words, for the head direction cells, the grid only appeared once the place cells stabilized. This slow stabilization of place fields is a known property (Bostock et al., 1991; Frank et al., 2004).

In the simulation, the place cells did not stabilize until a sufficient number of place cells were created (Figure 9C). Specifically, these additional memories were located immediately outside the enclosure, around all borders (Figure 9D). These “outside the box” memories served to constrain the interior place cells, locking them in position despite ongoing consolidation. This dynamic can be seen in a movie showing a representative simulation. The movie shows the positions of the head direction sensitive place cells during initial learning, and then during additional sessions of prior experience as the movie speeds up (see link in Figure 9 capture).

Why did the non-spatial grid cell (k) produce a grid immediately, before the place cells stabilized? As discussed in relation to Figure 8, the non-spatial grid cell is the projection through the 3D volume of real-world coordinates that includes X, Y, and head direction. Each grid field of a non-spatial grid cell reflects feedback from several place cells that each have a different head direction sensitivity (see for instance the allocentric pairs of memories illustrated in Figure 8C and 8D). Thus, each grid field is the average across several memories that entail different viewpoints and this averaging across memories provides stability even if the individual memories are not yet stable. This average of unstable memories produces a blurry sort of grid pattern without any prior experience.

A final piece of the puzzle relies on the same mechanism that caused the grid pattern to align with the borders as reported in the results of Figures 6 and 7. Specifically, there are some “sticky” locations with ongoing consolidation because the connection weights are bounded. Because weights cannot go below their minimum or above their maximum, it is slightly more difficult for consolidation to push or pull connection weights over the peak value or under the minimum value of the tuning curve. Thus, the place cells tend to linger in locations that correspond to the peak or trough of a border cell. There are multiple peak and trough locations but for the parameter values in this simulation, the grid pattern seen in Figure 9C shows the set of peak/trough locations that satisfy the desired spacing between memories. Thus, the average across memories shows a reliable grid field at these locations even though the memories are unstable.”

(6) Other predictions. Clearly, the model makes many interesting (and quite specific!) predictions. But does it make some known simple predictions?• More place cells at rewarded (or more visited) locations. Some empirical researchers seem to think this is not as obvious as it seems (e.g., Duvellle et al., 2019; JoN; Nyberg et al., 2021, Neuron Review).• Grid cell field moves toward reward (Butler et al., 2019; Boccera et al., 2019).• Grid cells deform in trapezoid (Krupic et al., 2015) and change in environments like mazes (Derikman et al., 2014).

Thank you for these suggestions and I have added the following paragraph to the discussion:

“In terms of the animal’s internal state, all locations in the enclosure may be viewed as equally aversive and unrewarding, which is a memorable characteristic of the enclosure. Reward, or lack thereof, is arguably one of the most important nonspatial characteristics and application of this model to reward might explain the existence of goal-related activity in place cells (Hok et al., 2007; although see Duvelle et al., 2019), reflecting the need to remember rewarding locations for goal directed behavior. Furthermore, if place cell memories for a rewarding location activate entorhinal grid cells, this may explain the finding that grid cells remap in an enclosure with a rewarded location such that firing fields are attracted to that location (Boccara et al., 2019; Butler et al., 2019). Studies that introduce reward into the enclosure are an important first step in terms of examining what happens to grid cells when the animal is placed in a more varied environment.”

Regarding the changes in shape of the environment, this was discussed in the section of the paper that reads “As seen in Figure 12, because all but one of the place cells was exterior when the simulated animal was constrained to a narrow passage, the hippocampal place cell memories were no longer arranged in a hexagonal grid. This disruption of the grid array for narrow passages might explain the finding that the grid pattern (of grid cells) is disrupted in the thin corner of a trapezoid (Krupic et al., 2015) and disrupted when a previously open enclosure is converted to a hairpin maze by insertion of additional walls within the enclosure (Derdikman et al., 2009).” This particular section of the paper now appears in the Appendix and Figure 12 is now Appendix Figure 2.

**Reviewer #2 (Public Review):**
The manuscript describes a new framework for thinking about the place and grid cell system in the hippocampus and entorhinal cortex in which these cells are fundamentally involved in supporting non-spatial information coding. If this framework were shown to be correct, it could have high impact because it would suggest a completely new way of thinking about the mammalian memory system in which this system is non-spatial. Although this idea is intriguing and thought-provoking, a very significant caveat is that the paper does not provide evidence that specifically supports its framework and rules out the alternate interpretations. Thus, although the work provides interesting new ideas, it leaves the reader with more questions than answers because it does not rule out any earlier ideas.Basically, the strongest claim in the paper, that grid cells are inherently non-spatial, cannot be specifically evaluated versus existing frameworks on the basis of the evidence that is shown here. If, for example, the author had provided behavioral experiments showing that human memory encoding/retrieval performance shifts in relation to the predictions of the model following changes in the environment, it would have been potentially exciting because it could potentially support the author's reconceptualization of this system. But in its current form, the paper merely shows that a new type of model is capable of explaining the existing findings. There is not adequate data or results to show that the new model is a significantly better fit to the data compared to earlier models, which limits the impact of the work. In fact, there are some key data points in which the earlier models seem to better fit the data.Overall, I would be more convinced that the findings from the paper are impactful if the author showed specific animal memory behavioral results that were only supported by their memory model but not by a purely spatial model. Perhaps the author could run new experiments to show that there are specific patterns of human or animal behavior that are only explained by their memory model and not by earlier models. But in its current form, I cannot rule out the existing frameworks and I believe some of the claims in this regard are overstated.

As previously detailed in Box 1 and as explained in the text in several places, the model provides an explanation of several findings that remain unexplained by other theories (see “Results Uniquely Explained by the Memory Model”). But more generally this is a good point, and the initial draft failed to fully articulate why a researcher might choose this model to guide future empirical investigations. A new section in the introduction that deals with these issues, titled ‘Why Model the Rodent Navigation Literature with a Memory Model?’ That section reads:

“Spatial navigation is inherently a memory problem – learning the spatial arrangement of a new enclosure requires memory for the conjunction of what and where. This has long been realized and in the introduction to ‘Hippocampus as a Cognitive Map’, O’Keefe and Nadel (1978) wrote “We shall argue that the hippocampus is the core of a neural memory system providing an objective spatial framework within which the items and events of an organism's experience are located and interrelated” (emphasis added). Furthermore, in the last chapter of their book, they extended cognitive map theory to human memory for non-spatial characteristics. However, in the decades since the development of cognitive map theory, the rodent spatial navigation and human memory literatures have progressed somewhat independently.

The ideas proposed in this model are an attempt to reunify these literatures by returning to the original claim that spatial navigation is inherently a memory problem. The goal of the current study is to explain the rodent spatial navigation literature using a memory model that has the potential to also explain the human memory literature. In contrast, most grid cell models (Bellmund et al., 2016; Bush et al., 2015; Castro & Aguiar, 2014; Hasselmo, 2009; Mhatre et al., 2012; Solstad et al., 2006; Sorscher et al., 2023; Stepanyuk, 2015; Widloski & Fiete, 2014) are domain specific models of spatial navigation and as such, they do not lend themselves to explanations of human memory. Thus, the reason to prefer this model is parsimony. Rather than needing to develop a theory of memory that is separate from a theory of spatial navigation, it might be possible to address both literatures with a unified account.

This study does not attempt to falsify other theories of grid cells. Instead, this model reaches a radically different interpretation regarding the function of grid cells; an interpretation that emerges from viewing spatial navigation as a memory problem. All other grid cell models assume that an entorhinal grid cell displaying a spatially arranged grid of firing fields serves the function of spatial coding (i.e., spatial grid cells exist to support a spatial metric). In contrast, the proposed memory model of grid cells assumes that the hexagonal tiling reflects the need to keep memories separate from each other to minimize confusion and confabulation – the grid pattern is the byproduct of pattern separation between memories rather than the basis of a spatial code.

It is now understood that grid-like firing fields can occur for non-spatial twodimensional spaces. For instance, human entorhinal cortex exhibits grid-like responses to video morph trajectories in a two-dimensional bird neck-length versus bird leg-length space (Constantinescu et al., 2016). As a general theory of learning and memory, the proposed memory model of grid cells is easily extended to explain these results (e.g., relabeling the border cell inputs in the model as neck-length and leg-length inputs). However, there are other grid cell models that can explain both spatial grid cells as well as non-spatial grid-like responses (Mok & Love, 2019; Rodríguez-Domínguez & Caplan, 2019; Stachenfeld et al., 2017; Wei et al., 2015). Similar to this memory model of grid cells, these models are also positioned to explain both the rodent spatial navigation and human memory literatures. Nevertheless, there is a key difference between this model and other grid cell models that generalize to non-spatial representations. Specifically, these other models assume that grid cells exhibiting spatial receptive fields serve the function of identifying positions in the environment (i.e., their function is spatial). As such, these models do not explain why most of the input to rodent hippocampus appears to be spatial (Boccara et al., 2010; Diehl et al., 2017; Grieves & Jeffery, 2017). This memory model of grid cells provides an answer to the apparent paucity of nonspatial cell types in rodent MTL by proposing that grid cells with spatial receptive fields have been misclassified as spatial (they are what cells rather than where cells) and that place cells are fundamentally memory cells that conjoin what and where.”

- The paper does not fully take into account all the findings regarding grid cells, some of which very clearly show spatial processing in this system. For example, findings on grid-bydirection cells (e.g., Sargolini et al. 2006) would seem to suggest that the entorhinal grid system is very specifically spatial and related to path integration. Why would grid-bydirection cells be present and intertwined with grid cells in the author's memory-related reconceptualization? It seems to me that the existence of grid-by-direction cells is strong evidence that at least part of this network is specifically spatial.

Head by direction grid cells were a key part of the reported results. These grid cells naturally arise in the model as the animal forms memories (aka, hippocampal place cells) that conjoin location (as defined by border cells), head direction at the time of memory formation, and one or more non-spatial properties found at that location. In this revision, I have attempted to better explain how including head direction in hippocampal memories naturally gives rise to these cell types. The introduction to the head direction module simulations now reads:

“According to this memory model of spatial navigation, place cells are the conjunction of location, as defined by border cells, and one or more properties that are remembered to exist at that location. Such memories could, for instance, allow an animal to remember the location of a food cache (Payne et al., 2021). The next set of simulations investigates behavior of the model when one of the to-be-remembered properties is head direction at the time when the memory was formed (e.g., the direction of a pathway leading to a food cache). Indicating that head direction is an important part of place cell representations, early work on place cells in mazes found strong sensitivity to head direction, such that the place field is found in one direction of travel but not the other (McNaughton et al., 1983; Muller et al., 1994). Place cells can exhibit a less extreme version of head direction sensitivity in open field recordings (Rubin et al., 2014), but the nature of the sensitivity is more complicated, depending on location of the animal relative to the place field center (Jercog et al., 2019).

It is possible that some place cell memories do not receive head direction input, as was the case for the simulations reported in Figures 6/7 – in those simulations, place cells were entirely insensitive to head direction, owing to a lack of input from head direction cells. However, removal of head direction input to hippocampus affects place cell responses (Calton et al., 2003) and grid cell responses (Winter et al., 2015), suggesting that head direction is a key component of the circuit. Furthermore, if place cells represent episodic memories, it seems natural that they should include head direction (i.e., viewpoint at the time of memory formation).

In the simulations reported next, head direction is simply another property that is conjoined in a hippocampal place cell memory. In this case, a head direction cell should become a head direction conjunctive grid cell (i.e., a grid cell, but only when the animal is heading in a particular direction), owing to memory feedback from the hexagonal array of hippocampal place cell memories. When including head direction, the real-world dimensions of variation are across three dimensions (X, Y, and head direction) rather than two, and consolidation will cause the place cells to arrange in a three-dimensional volume. The simulation reported below demonstrates that this situation provides a “grid module”.”

- I am also concerned that the paper does not do enough to address findings regarding how the elliptical shape of grid fields shifts when boundaries of an environment compress in one direction or change shape/angles (Lever et al., & Krupic et al). Those studies show compression in grid fields based on boundary position, and I don't see how the authors' model would explain these findings.

This finding was covered in the original submission: “For instance, perhaps one egocentric/allocentric pair of mEC grid modules is based on head direction (viewpoint) in remembered positions relative to the enclosure borders whereas a different egocentric/allocentric pair is based on head direction in remembered positions relative to landmarks exterior to the enclosure. This might explain why a deformation of the enclosure (moving in one of the walls to form a rectangle rather than a square) caused some of the grid modules but not others to undergo a deformation of the grid pattern in response to the deformation of the enclosure wall (see also Barry et al., 2007). More specifically, if there is one set of non-orthogonal dimensions for enclosure borders and the movement of one wall is too modest as to cause avoid global remapping, this would deform the grid modules based the enclosure border cells. At the same time, if other grid modules are based on exterior properties (e.g., perhaps border cells in relation to the experimental room rather than the enclosure), then those grid modules would be unperturbed by moving the enclosure wall.”

I apologize for being unclear in describing how the model might explain this result. The paragraph has been rewritten and now reads:

“Consider the possibility that one mEC grid modules is based on head direction (viewpoint) in remembered positions relative to the enclosure borders (e.g., learning the properties of the enclosure, such as the metal surface) while a different grid module is based on head direction in remembered positions relative to landmarks exterior to the enclosure (e.g., learning the properties of the experimental room, such as the sound of electronics that the animal is subject to at all locations). This might explain why a deformation of the enclosure (moving one of the walls to form a rectangle rather than a square) caused some of the grid modules but not others to undergo a deformation of the grid pattern in response to the deformation of the enclosure wall (see also Barry et al., 2007). More specifically, suppose that the movement of one wall is modest and after moving the wall, the animal views the enclosure as being the same enclosure, albeit slightly modified (e.g., when a home is partially renovated, it is still considered the same home). In this case, the set of non-orthogonal dimensions associated with enclosure borders would still be associated with the now-changed borders and any memories in reference to this border-determined space would adjust their positions accordingly in real-world coordinates (i.e., the place cells would subtly shift their positions owing to this deformation of the borders, producing a corresponding deformation of the grid). At the same time, there may be other sets of memories that are in relation to dimensions exterior to the enclosure. Because these exterior properties are unchanged, any place cells and grid cells associated with the exterior-oriented memories would be unchanged by moving the enclosure wall.”

- Are findings regarding speed modulation of grid cells problematic for the paper's memory results?

- A further issue is that the paper does not seem to adequately address developmental findings related to the timecourses of the emergence of different cell types. In their simulation, researchers demonstrate the immediate emergence of grid fields in a novel environment, while noting that the stabilization of place cell positions takes time. However, these simulation findings contradict previous empirical developmental studies (Langston et al., 2010). Those studies showed that head direction cells show the earliest development of spatial response, followed by the appearance of place cells at a similar developmental stage. In contrast, grid cells emerge later in this developmental sequence. The gradual improvement in spatial stability in firing patterns likely plays a crucial role in the developmental trajectory of grid cells. Contrary to the model simulation, grid cells emerge later than place cells and head direction cells, yet they also hold significance in spatial mapping.

- The model simulations suggest that certain grid patterns are acquired more gradually than others. For instance, egocentric grid cells require the stabilization of place cell memories amidst ongoing consolidation, while allocentric grid cells tend to reflect average place field positions. However, these findings seemingly conflict with empirical studies, particularly those on the conjunctive representation of distance and direction in the earliest grid cells. Previous studies show no significant differences were found in grid cells and grid cells with directional correlates across these age groups, relative to adults (Wills et al., 2012). This indicates that the combined representation of distance and direction in single mEC cells is present from the earliest ages at which grid cells emerge.

These are good points and they have been addressed in a new section of the introduction titled ‘The Scope of the Proposed Model’. That section reads:

“The reported simulations explain why most mEC cell types in the rodent literature appear to be spatial (Boccara et al., 2010; Diehl et al., 2017; Grieves & Jeffery, 2017). Assuming that rodents can form non-spatial memories, rodent hippocampus must receive non-spatial input from entorhinal cortex. These simulations suggest that characterization of the rodent mEC cortex as primarily spatial might be incorrect if most grid cells (except perhaps head direction conjunctive grid cells) have been mischaracterized as spatial. Other literatures with other species find non-spatial representations in MTL (Gulli et al., 2020; Quiroga et al., 2005; Wixted et al., 2014) and non-spatial hippocampal memory encoding has been found in rodents (Liu et al., 2012; McEchron & Disterhoft, 1999). The proposed memory model is compatible with these results – the ideas contained in this model could be applied to nonspatial memory representations. However, surveys of cell types in rodent entorhinal cortex seem to indicate that most cells are spatial (Boccara et al., 2010; Diehl et al., 2017; Grieves & Jeffery, 2017). How can the rodent hippocampus encode nonspatial memories if most of its input is spatial? The goal of the reported simulations is to explain the apparent paucity of non-spatial cells in rodent entorhinal cortex by proposing that grid cells have been misclassified as spatial (see also Luo et al., 2024).

Given the simplicity of the proposed model, there are important findings that the model cannot address -- it is not that the model makes the wrong predictions but rather that it makes no predictions. The role of running speed (Kraus et al., 2015) is one such variable for which the model makes no predictions. Similarly, because the model is a rate-coded model rather than a model of oscillating spiking neurons, it makes no predictions regarding theta oscillations (Buzsáki & Moser, 2013). The model is an account of learning and memory for an adult animal, and it makes no predictions regarding the developmental (Langston et al., 2010; Muessig et al., 2015; Wills et al., 2012) or evolutionary (Rodrıguez et al., 2002) time course of different cell types. This model contains several purely spatial representations such as border cells, head direction cells, and head direction conjunctive grid cells and it may be that these purely spatial cell types emerged first, followed by the evolution and/or development of non-spatial cell types. However, this does not invalidate the model. Instead, this is a model for an adult animal that has both episodic memory capabilities and spatial navigation capabilities, irrespective of the order in which these capabilities emerged.

This model has the potential to explain context effects in memory (Godden & Baddeley, 1975; Gulli et al., 2020; Howard et al., 2005). According to this model, different grid cells represent different non-spatial characteristics and place cells represent the combination of these “context” factors and location. In the simulation, just one grid cell is simulated but the same results would emerge when simulating hundreds of different non-spatial inputs provided that all of the simulated non-spatial inputs exist throughout the recording session. However, there is evidence that hippocampus can explicitly represent the passage of time (Eichenbaum, 2014), and time is assuredly an important factor in defining episodic memory (Bright et al., 2020). Thus, although the current model addresses unique combinations of what and where, it is left to future work to incorporate representations of when in the memory model.”

**Reviewer #3 (Public Review):**
A crucial assumption of the model is that the content of experience must be constant in space. It's difficult to imagine a real-world example that satisfies this assumption. Odors and sounds are used as examples. While they are often more spatially diffuse than an objects on the ground, odors and sounds have sources that are readily detectable. Animals can easily navigate to a food source or to a vocalizing conspecific. This assumption is especially problematic because it predicts that all grid cells should become silent when their preferred non-spatial attribute (e.g. a specific odor) is missing. I'm not aware of any experimental data showing that grid cells become silent. On the contrary, grid cells are known to remain active across all contexts that have been tested, including across sleep/wake states. Unlike place cells, grid cells do not seem to turn off. Since grid cells are active in all contexts, their preferred attribute must also be present in all contexts, and therefore they would not convey any information about the specific content of an experience.

These are good points and in this revision I have attempted to explain that there is a great deal of contextual similarity across all recording sessions. One paragraph in the discussion now reads

“In a typical rodent spatial navigation study, the non-spatial attributes are wellcontrolled, existing at all locations regardless of the enclosure used during testing (hence, a grid cell in one enclosure will be a grid cell in a different enclosure). Because labs adopt standard procedures, the surfaces, odors (e.g., from cleaning), external lighting, time of day, human handler, electronic apparatus, hunger/thirst state, etc. might be the same for all recording sessions. Additionally, the animal is not allowed to interact with other animals during recording and this isolation may be an unusual and highly salient property of all recording sessions. Notably, the animal is always attached to wires during recording. The internal state of the animal (fear, aloneness, the noise of electronics, etc.) is likely similar across all recording situations and attributes of this internal state are likely represented in the hippocampus and entorhinal input to hippocampus. According to this model, hippocampal place cells are “marking” all locations in the enclosure as places where these things tend to happen.”

The proposed novelty of this theory is that other models all assume that grid cells encode space. This isn't quite true of models based on continuous attractor networks, the discussion of which is notably absent. More specifically, these models focus on the importance of intrinsic dynamics within the entorhinal cortex in generating the grid pattern. While this firing pattern is aligned to space during navigation and therefore can be used as a representation of that space, the neural dynamics are preserved even during sleep. Similarly, it is because the grid pattern does not strictly encode physical space that gridlike signals are also observed in relation to other two-dimensional continuous variables.

These models were briefly discussed in the general discussion section and in this revision they are further discussed in the introduction in a new section, titled ‘Why Model the Rodent Navigation Literature with a Memory Model?’ That section reads:

“Spatial navigation is inherently a memory problem – learning the spatial arrangement of a new enclosure requires memory for the conjunction of what and where. This has long been realized and in the introduction to ‘Hippocampus as a Cognitive Map’, O’Keefe and Nadel (1978) wrote “We shall argue that the hippocampus is the core of a neural memory system providing an objective spatial framework within which the items and events of an organism's experience are located and interrelated” (emphasis added). Furthermore, in the last chapter of their book, they extended cognitive map theory to human memory for non-spatial characteristics. However, in the decades since the development of cognitive map theory, the rodent spatial navigation and human memory literatures have progressed somewhat independently.

The ideas proposed in this model are an attempt to reunify these literatures by returning to the original claim that spatial navigation is inherently a memory problem. The goal of the current study is to explain the rodent spatial navigation literature using a memory model that has the potential to also explain the human memory literature. In contrast, most grid cell models (Bellmund et al., 2016; Bush et al., 2015; Castro & Aguiar, 2014; Hasselmo, 2009; Mhatre et al., 2012; Solstad et al., 2006; Sorscher et al., 2023; Stepanyuk, 2015; Widloski & Fiete, 2014) are domain specific models of spatial navigation and as such, they do not lend themselves to explanations of human memory. Thus, the reason to prefer this model is parsimony. Rather than needing to develop a theory of memory that is separate from a theory of spatial navigation, it might be possible to address both literatures with a unified account.

This study does not attempt to falsify other theories of grid cells. Instead, this model reaches a radically different interpretation regarding the function of grid cells; an interpretation that emerges from viewing spatial navigation as a memory problem. All other grid cell models assume that an entorhinal grid cell displaying a spatially arranged grid of firing fields serves the function of spatial coding (i.e., spatial grid cells exist to support a spatial metric). In contrast, the proposed memory model of grid cells assumes that the hexagonal tiling reflects the need to keep memories separate from each other to minimize confusion and confabulation – the grid pattern is the byproduct of pattern separation between memories rather than the basis of a spatial code.

It is now understood that grid-like firing fields can occur for non-spatial two dimensional spaces. For instance, human entorhinal cortex exhibits grid-like responses to video morph trajectories in a two-dimensional bird neck-length versus bird leg-length space (Constantinescu et al., 2016). As a general theory of learning and memory, the proposed memory model of grid cells is easily extended to explain these results (e.g., relabeling the border cell inputs in the model as neck-length and leg-length inputs). However, there are other grid cell models that can explain both spatial grid cells as well as non-spatial grid-like responses (Mok & Love, 2019; Rodríguez-Domínguez & Caplan, 2019; Stachenfeld et al., 2017; Wei et al., 2015). Similar to this memory model of grid cells, these models are also positioned to explain both the rodent spatial navigation and human memory literatures. Nevertheless, there is a key difference between this model and other grid cell models that generalize to non-spatial representations. Specifically, these other models assume that grid cells exhibiting spatial receptive fields serve the function of identifying positions in the environment (i.e., their function is spatial). As such, these models do not explain why most of the input to rodent hippocampus appears to be spatial (Boccara et al., 2010; Diehl et al., 2017; Grieves & Jeffery, 2017). This memory model of grid cells provides an answer to the apparent paucity of nonspatial cell types in rodent MTL by proposing that grid cells with spatial receptive fields have been misclassified as spatial (they are what cells rather than where cells) and that place cells are fundamentally memory cells that conjoin what and where.”

The use of border cells or boundary vector cells as the main (or only) source of spatial information in the hippocampus is not well supported by experimental data. Border cells in the entorhinal cortex are not active in the center of an environment. Boundary-vector cells can fire farther away from the walls but are not found in the entorhinal cortex. They are located in the subiculum, a major output of the hippocampus. While the entorhinalhippocampal circuit is a loop, the route from boundary-vector cells to place cells is much less clear than from grid cells. Moreover, both border cells and boundary-vector cells (which are conflated in this paper) comprise a small population of neurons compared to grid cells.

AUTHOR RESPONSE: The model can be built without assuming between-border cells (early simulations with the model did not make this assumption). Regarding this issue, the text reads “Unlike the BVC model, the boundary cell representation is sparsely populated using a basis set of three cells for each of the three dimensions (i.e., 9 cells in total), such that for each of the three non-orthogonal orientations, one cell captures one border, another the opposite border, and the third cell captures positions between the opposing borders (Solstad et al., 2008). However, this is not a core assumption, and it is possible to configure the model with border cell configurations that contain two opponent border cells per dimension, without needing to assume that any cells prefer positions between the borders (with the current parameters, the model predicts there will be two border cells for each between-border cell). Similarly, it is possible to configure the model with more than 3 cells for each dimension (i.e., multiple cells representing positions between the borders).” The Solstad paper found a few cells that responded in positions between borders, but perhaps not as many as 1 out of 3 cells, such as this particular model simulation predicts. If the paucity of between-border cells is a crucial data point, the model can be reconfigured with opponent-border cells without any between border cells. The reason that 3 border cells were used rather than 2 opponent border cells was for simplicity. Because 3 head direction cells were used to capture the face-centered cubic packing of memories, the simulation also used 3 border cells per dimensions to allow a common linear sum metric when conjoining dimensions to form memories. If the border dimensions used 2 cells while head direction used 3 cells, a dimensional weighting scheme would be needed to allow this mixing of “apples and oranges” in terms of distances in the 3D space that includes head direction.

**Recommendations for the authors:**

**Reviewer #1 (Recommendations For The Authors):**
Specific questions/clarifications:(1) Assumption of population-based vs single unit link to biological cells: At the start, the author assumes that each unit here can be associated with a population: "the simulated activation values can be thought of as proportional to the average firing rate of an ensemble of neurons with similar inputs and outputs (O'Reilly & Munakata, 2000)." But is a 'grid cell' found here a single cell or an average of many cells? Does this mean the model assumes many cells that have different fields that are averaged, which become a grid-like unit in the model? But in biology, these are single cells? Or does it mean a grid response is an average of the place cell inputs?

I apologize for being unclear about this. The grid cells in the model are equivalent to real single cells except that the simulation uses a ratecoded cell rather than a spiking cell. The averaging that was mentioned in the paper is across identically behaving spiking cells rather than across cells with different grid field arrangements. To better explain this, I have added the following text:

“For instance, consider a set of several thousand spiking grid cells that are identical in terms of their firing fields. At any moment, some of these identically-behaving cells will produce an action potential while others do not (i.e., the cells are not perfectly synchronized), but a snapshot of their behavior can be extracted by calculating average firing rate across the ensemble. The simulated cells in the model represent this average firing rate of identically-behaving ensembles of spiking neurons.”

This is a mathematical short-cut to avoid simulating many spiking neurons. Because this model was compared to real spike rate maps, this real-valued average firing rate is down-sampled to produce spikes by finding the locations that produced the top 5% of real-valued activation values across the simulation.

(2) It is not clear to me why they are circular border cells/basis sets.

In the initial submission, there was a brief paragraph describing this assumption. In this revision, that paragraph has been expanded and modified for greater clarity. It now reads:

“Because head direction is necessarily a circular dimension, it was assumed that all dimensions are circular (a circular dimension is approximately linear for nearby locations). This assumption of circular dimensions was made to keep the model relatively simple, making it easier to combine dimensions and allowing application of the same processes for all dimensions. For instance, the model requires a weight normalization process to ensure that the pattern of weights for each dimension corresponds to a possible input value along that dimension. However, the normalization for a linear dimension is necessarily different than for a circular dimension. Because the neural tuning functions were assumed to be sine waves, normalization requires that the sum of squared weights add up to a constant value. For a linear dimension, this sum of squares rule only applies to the subset of cells that are relevant to a particular value along the dimension whereas for a circular dimension, this sum of squares rule is over the entire set of cells that represent the dimension (i.e., weight normalization is easier to implement with circular dimensions). Although all dimensions were assumed to be circular for reasons of mathematical convenience and parsimony, circular dimensions may relate to the finding that human observers have difficultly re-orienting themselves in a room depending on the degree of rotational symmetry of the room (Kelly et al., 2008). In addition, this simplifying assumption allows the model to capture the finding that the population of grid cells lies on a torus (Gardner et al., 2022), although I note that the model was developed before this result was known.”

(3) Why is it 3 components? I realise that the number doesn't matter too much, but I believe more is better, so is it just for simplicity?

In this revision, additional text has been added to explain this assumption: “To keep the model simple, the same number of cells was assumed for all dimensions and all dimensions were assumed to be circular (head direction is necessarily circular and because one dimension needed to be circular, all dimensions were assumed to be circular). Three cells per dimensions was chosen because this provides a sparse population code of each dimension, with few border cells responding between borders, with few border cells responding between borders, while allowing three separate phases of grid cells within a grid cell module (in the model, a grid cell module arises from combination of a third dimension, such as head direction, with the real-world X/Y dimensions defined by border cells).”

As a reminder, the text explaining the sparse coding of border cells reads: “However, this is not a core assumption, and it is possible to configure the model with border cell configurations that contain two opponent border cells per dimension, without needing to assume that any cells prefer positions between the borders (with the current parameters, the model predicts there will be two border cells for each between-border cell). Similarly, it is possible to configure the model with more than 3 cells for each dimension (i.e., multiple cells representing positions between the borders).”

The model can work with just two opponent cells or with more than three cells per basis set. In different simulations, I have explored these possibilities. Three was chosen because it is a convenient way to highlight the face-centered cubic packing of memories that tends to occur (FCP produces 3 alternating layers of hexagonally arranged firing fields). Thus, each of the three head direction cells captures a different layer of the FCP arrangement. A more realistic simulation might combine 6 different head direction cells tiling the head direction dimension with opponent border cells (just 2 cells for each border dimensions). Such a combination would produce responses at borders, but no responses between borders and, at the same time, the head direction cells would still reveal the FCP arrangement. However, it is not easy to find the right parameters for such a mix-and-match simulation in which different dimensions have different numbers of tuning functions (e.g., some dimensions having 2 cells while others have 3 or 6 and some dimensions being linear while others are circular). When all of the dimensions are of the same type, the simple sum that arises from multiplying the input by the weight values gives rise to Euclidean distance (see Figure 3B). With a mix-and-match model of different dimension-types, it should be possible to adjust the sum to nevertheless produce a monotonic function with Euclidean distance although I leave this to future work. To keep things simple, I assumed that all dimensions are of the same type (circular, with 3 cells per dimension).

(4) Confusion due to the border cells/box was unclear to me. "If the period of the circular border cells was the same as the width of the box, then a memory pushed outside the box on one side would appear on the opposite side of the box, in which case the partial grid field on one side should match up with its remainder on the other side. This would entail complete confusion between opposite sides of the box, and the representation of the box would be a torus (donut-shaped) rather than a flat two-dimensional surface. To reduce confusion ..." Is this confusion of the model? Of the animal?

This would be confusion of the animal (e.g., a memory field overlapping with one border would also appear at the opposite border in the corresponding location). At one point in model development, I made the assumption that one side of the box wraps to the other side, and I asked Trygve Solstad to run some analyses of real data to see if cells actually wrap around in this manner. He did not find any evidence of this, and so I decided to include outsidethe-box representational area which, as it turned out, allowed the model to capture other behaviors as detailed in the paper.

This section of the paper now reads:

“The cosine tuning curves of the simulated border cells represent distance from the border on both sides of the border (i.e., firing rate increases as the animal approaches the border from either the inside or the outside of the enclosure). Experimental procedures do not allow the animal to experience locations immediately outside the enclosure, but these locations remain an important part of the hypothetic representation, particularly when considering the modification of memories through consolidation (i.e., a memory created inside the enclosure might be moved to a location outside the enclosure). This symmetry about the border cell’s preferred location is needed to maintain an unbiased representation, with a constant sum of squares for the border cell inputs (see methods section). Rather than using linear dimensions, all dimensions were assumed to be circular to keep the model relatively simple. This assumption was made because head direction is necessarily a circular dimension and by having all dimensions be circular, it is easy to combine dimensions in a consistent manner to produce multidimensional hippocampal place cell memories. Thus, the border cells define a torus (or more accurately a three-torus) of possible locations. This provides a hypothetical space of locations that could be represented.

In light of the assumption to represent border cells with a circular dimension, when a memory is pushed outside the East wall of the enclosure, it would necessarily be moved to the West wall of the enclosure if the period of the circular dimension was equal to the width of the enclosure. If this were true, then the partial grid field on one side of the enclosure would match up with its remainder on the other side. Such a situation would cause the animal to become completely confused regarding opposite sides of the enclosure (a location on the West wall would be indistinguishable from the corresponding location on the East wall). To reduce confusion between opposite sides of the enclosure, the width of the enclosure in which the animal navigated (Figure 5) was assumed to be half as wide as the full period of the border cells. In other words, although the space of possible representations was a three-torus, it was assumed that the real-world twodimensional enclosure encompassed a section of the torus (e.g., a square piece of tape stuck onto the surface of a donut). The torus is better thought of as “playing field” in which different sizes and shapes of enclosure can be represented (i.e., different sizes and shapes of tape placed on the donut). Furthermore, this assumption provides representational space that is outside the box without such locations wrapping around to the opposite side of the box.”

(5) Figure 3 - This result seems to be related to whether you use Euclidean or city-block distance. If you use Euclidean distances in two dimensions wouldn't this work out fine?

Euclidean distance was the metric used in the analysis of the two-dimensional simulation, but this did not work out. To make this clear, I have changed the label on the x-axes to read “Euclidean distance” for both the two- and three-dimensional simulations. The two-dimensional simulation produced city block behavior rather than Euclidean behavior because memory retrieval is the sum of the two dimensions, as is standard in neural networks, rather than the Euclidian distance formula, which would require that memory retrieval be the square root of the sum of squares of the two dimensions. One way to address this problem with the two-dimensional simulation would be to use a specific Euclidean-mimicking activation function rather than a simple sum of dimensions. The very first model I developed used such an activation function as applied to opponent border cells with just two dimensions (so 4 cells in total – left/right and top/down). This produced Euclidean behavior, but the activation function was implausible and did not generalize to simulations that also included head direction. In contrast, with three non-orthogonal dimensions, the simple sum of dimensions is approximately Euclidean.

(6) Final sentence of the Discussion: "However, unlike the present model, these models still assume that entorhinal grid cells represent space rather than a non-spatial attribute." I am not sure if the authors of the cited papers will agree with this. They consider the spatial cases, but most argue they can treat non-spatial features as well. What the author might mean is that they assume non-spatial features are in some metric space that, in a way, is spatial. However, I am not sure if the author would argue that non-spatial features cannot be encoded metrically (e.g., Euclidean distance based on the similarity of odours).

In this section, when referring to “entorhinal grid cells” I was specifically referring to traditional grid cells in a rodent spatial navigation experiment. I did not mean to imply that these other theories cannot explain nonspatial grid fields, such as in the two-dimensional bird space grid cells found with humans. The way in which the proposed memory model and these other models differ is in terms of what they assume regarding the function of grid cells that exhibit spatial grid fields. In this revision, I have changed this text to read:

“These models can capture some of the grid cell results presented in the current simulations, including extension to non-spatial grid-like responses (e.g., grid field that cover a two-dimensional neck/leg length bird space). Furthermore, these models may be able to explain memory phenomena similar to the model proposed in this study. However, unlike the proposed model, these models assume that the function of entorhinal grid cells that exhibit spatial X/Y grid fields during navigation is to represent space. In contrast, the memory model proposed in this study assume that the function of spatial X/Y grid cells is to represent a non-spatial attribute; the only reason they exhibit a spatial X/Y grid is because memories of that non-spatial attribute are arranged in a hexagonal grid owing to the uncluttered/unvarying nature of the enclosure. Thus, these model do not explain why most of the input to rodent hippocampus appears to be spatial (Boccara et al., 2010b; Diehl et al., 2017; Grieves & Jeffery, 2017) whereas the proposed model can explain this situation as reflecting the miss-classification of grid cells with a spatial arrangement as providing spatial input to hippocampus.”

(7) It would be interesting to see videos/gifs of the model learning, and an idea of how many steps of trials it takes (is it capturing real-time rodent cell firing whilst foraging, or is it more abstracted, taking more trials).

The short answer is “yes”, the model is capturing real-time rodent cell firing while foraging. This is particularly true when simulating place cell memories in the absence of head direction information, as was shown in a video provided in the initial submission in relation to Figure 4. In this revision, I have provided a second video of learning when simulating place cell memories that include head direction. This second video is in relation to the results reported in Figure 9. This shows that even when learning a three-dimensional real-world space (X, Y, and head direction), the model rapidly produces an on-average hexagonal arrangement of place cells memories owing to the slight tendency of the place cell memories to linger in some locations as compared to others during consolidation. More specifically, they are more likely to linger in the locations that are the intersections of the peaks and/or troughs of the border cells and it is this tendency that supports the immediate appearance of grid cells. However, because the place cell memories are still shifting, head direction conjunctive grid cells are slower to emerge (the head direction conjunctive grid cells require stabilization of the place cells). The video then speeds up the learning process to so how place cells eventually stabilize after sufficient learning of the borders of the enclosure from different head/view directions.

(8) One question is whether all the results have to be presented in the main text. It was difficult to see which key predictions fit the data and do so better than a spatial/navigation account.

Thank you for this suggestion. To make the paper more readable and easier for different readers with different interests to choose different aspects of the results to read, the second half of the results have been put in an appendix. More specifically, the second half of the results concerned place cells rather than grid cells. Thus, in this revision, the main text concerns grid cell results and the appendix concerns place cell results.

**Reviewer #3 (Recommendations For The Authors):**
The title could usefully be shortened to focus on the main argument that observed firing patterns could be consistent with mapping memories instead of space. It's a stretch to argue that memory is the primary role when no such data is presented (i.e., there is no comparison of competing models).

This is a good point (I do not present evidence that conclusively indicates the function of MTL). This original title was chosen to make clear how this account is a radical departure from other accounts of grid cells. The revised title highlights that: (1) a memory model can also explain rodent single cell recording data during navigation; and (2) grid cell may not be non-spatial. The revised title is: “A Memory Model of Rodent Spatial Navigation: Place Cells are Memories Arranged in a Grid and Grid Cells are Non-spatial”

When arguing that the main role of the hippocampus is memory, I strongly suggest engaging with the work of people like Howard Eichenbaum who spent the better part of their career arguing the same (e.g. DOI:10.1152/jn.00005.2017.)

Thank you for pointing out this important oversight. Early in introduction, I now write: “The proposal that hippocampus represents the multimodal conjunctions that define an episode is not new (Marr et al., 1991; Sutherland & Rudy, 1989) and neither is the proposal that hippocampal memory supports spatial/navigation ability (Eichenbaum, 2017). This view of the hippocampus is consistent with “feature in place” results (O’Keefe & Krupic, 2021) in which hippocampal cells respond to the conjunction of a non-spatial attribute affixed to a specific location, rather than responding more generically to any instance of a non-spatial attribute. In other words, the what/where conjunction is unique. Furthermore, the uniqueness of the what/where conjunction may be the fundamental building block of spatial memory and navigation. In reviewing the hippocampal literature, Howard Eichenbaum (2017) concludes that ‘the hippocampal system is not dedicated to spatial cognition and navigation, but organizes experiences in memory, for which spatial mapping and navigation are both a metaphor for and a prominent application of relational memory organization.’”

With a focus on episodic memory, there should be a mention of the temporal component of memory. While it may rightfully be beyond the scope of this model, it's confusing to omit time completely from the discussion.

This issue and several others are now addressed in a new section in the introduction titled ‘The Scope of the Proposed Model’. That section reads:

“The reported simulations explain why most mEC cell types in the rodent literature appear to be spatial (Boccara et al., 2010; Diehl et al., 2017; Grieves & Jeffery, 2017). Assuming that rodents can form non-spatial memories, rodent hippocampus must receive non-spatial input from entorhinal cortex. These simulations suggest that characterization of the rodent mEC cortex as primarily spatial might be incorrect if most grid cells (except perhaps head direction conjunctive grid cells) have been mischaracterized as spatial. Other literatures with other species find non-spatial representations in MTL (Gulli et al., 2020; Quiroga et al., 2005; Wixted et al., 2014) and non-spatial hippocampal memory encoding has been found in rodents (Liu et al., 2012; McEchron & Disterhoft, 1999). The proposed memory model is compatible with these results – the ideas contained in this model could be applied to nonspatial memory representations. However, surveys of cell types in rodent entorhinal cortex seem to indicate that most cells are spatial (Boccara et al., 2010; Diehl et al., 2017; Grieves & Jeffery, 2017). How can the rodent hippocampus encode nonspatial memories if most of its input is spatial? The goal of the reported simulations is to explain the apparent paucity of non-spatial cells in rodent entorhinal cortex by proposing that grid cells have been misclassified as spatial (see also Luo et al., 2024).

Given the simplicity of the proposed model, there are important findings that the model cannot address -- it is not that the model makes the wrong predictions but rather that it makes no predictions. The role of running speed (Kraus et al., 2015) is one such variable for which the model makes no predictions. Similarly, because the model is a rate-coded model rather than a model of oscillating spiking neurons, it makes no predictions regarding theta oscillations (Buzsáki & Moser, 2013). The model is an account of learning and memory for an adult animal, and it makes no predictions regarding the developmental (Langston et al., 2010; Muessig et al., 2015; Wills et al., 2012) or evolutionary (Rodrıguez et al., 2002) time course of different cell types. This model contains several purely spatial representations such as border cells, head direction cells, and head direction conjunctive grid cells and it may be that these purely spatial cell types emerged first, followed by the evolution and/or development of non-spatial cell types. However, this does not invalidate the model. Instead, this is a model for an adult animal that has both episodic memory capabilities and spatial navigation capabilities, irrespective of the order in which these capabilities emerged.

This model has the potential to explain context effects in memory (Godden & Baddeley, 1975; Gulli et al., 2020; Howard et al., 2005). According to this model, different grid cells represent different non-spatial characteristics and place cells represent the combination of these “context” factors and location. In the simulation, just one grid cell is simulated but the same results would emerge when simulating hundreds of different non-spatial inputs provided that all of the simulated non-spatial inputs exist throughout the recording session. However, there is evidence that hippocampus can explicitly represent the passage of time (Eichenbaum, 2014), and time is assuredly an important factor in defining episodic memory (Bright et al., 2020). Thus, although the current model addresses unique combinations of what and where, it is left to future work to incorporate representations of when in the memory model.”

I recommend explaining the motivation of the theory in more detail in the introduction. It reads as "what if it's like this?" It would be helpful to instead highlight the limitations of current theories and argue why this theory is either a better fit for the data or is logically simpler.

This issue and several others are now addressed in the new section in the introduction titled ‘Why Model the Rodent Navigation Literature with a Memory Model?’, which I quoted above in response to the public reviews.

It's worth considering shortening the results section to include only those that most convincingly support the main claim. The manuscript is quite long and appears to lack focus at times.

Thank you for this suggestion. To make the paper more readable and easier for different readers with different interests to choose different aspects of the results to read, the second half of the results have been put in an appendix. More specifically, the second half of the results concerned place cells rather than grid cells. Thus, in this revision, the main text concerns grid cell results and the appendix concerns place cell results.

The discussion of path dependence on the formation of the grid pattern is important but only briefly discussed. It may be useful to add simulations testing whether different paths (not random walks) produce distorted grid patterns.

The short answer is that the path doesn’t affect things in general. The consolidation rule ensures equally spaced memories even if, for instance, one side of the enclosure is explored much more than the other side. As just one example, I have run simulations with a radial arm maze and even though the animal is constrained to only run on the maze arms. The memories still arrange hexagonally as memories become pushed outside the arms. Rather than adding additional simulations to study, I now briefly describe this in the model methods:

“Of note, the ability of the model to produce grid cell responses does not depend on this decision to simulate an animal taking a random walk – the same results emerge if the animal is more systematic in its path. All that matters for producing grid cell responses is that the animal visits all locations and that the animal takes on different head directions for the same location in the case of simulations that also include head direction as an input to hippocampal place cells.”

I struggle to understand in Figure 3 why retrieval strength ought to scale monotonically with Euclidean distance, and why that justifies a more complex model (three non-orthogonal dimensions).

The introduction to this section now reads: “Animals can plan novel straight line paths to reach a known position and evidence suggests they do so by learning Euclidean representations of space (Cheng & Gallistel, 2014; Normand & Boesch, 2009; Wilkie, 1989). Thus, it was assumed that hippocampal place cells represent positions in Euclidean space (as opposed to non-Euclidean space, such a occurs with a city-block metric).”

p.17 "although the representational space is a torus (or more specifically a three-torus), it is assumed that the real-world two-dimensional surface is only a section of the torus (e.g., a square piece of tape stuck onto the surface of a donut)." I fail to understand how the realworld surface is only a part of the torus. In the existing theoretical and experimental work on toroidal topology of grid cell activity, the torus represents a very small fraction of the real world, and repeating activity on the toroidal manifold is a crucial feature of how it maps 2D space in a regular manner. Why then here do you want the torus to be larger than the realworld?

This section has been rewritten to better explain these assumptions. The relevant paragraphs now read:

“The cosine tuning curves of the simulated border cells represent distance from the border on both sides of the border (i.e., firing rate increases as the animal approaches the border from either the inside or the outside of the enclosure). Experimental procedures do not allow the animal to experience locations immediately outside the enclosure, but these locations remain an important part of the hypothetic representation, particularly when considering the modification of memories through consolidation (i.e., a memory created inside the enclosure might be moved to a location outside the enclosure). This symmetry about the border cell’s preferred location is needed to maintain an unbiased representation, with a constant sum of squares for the border cell inputs (see methods section). Rather than using linear dimensions, all dimensions were assumed to be circular to keep the model relatively simple. This assumption was made because head direction is necessarily a circular dimension and by having all dimensions be circular, it is easy to combine dimensions in a consistent manner to produce multidimensional hippocampal place cell memories. Thus, the border cells define a torus (or more accurately a three-torus) of possible locations. This provides a hypothetical space of locations that could be represented.

In light of the assumption to represent border cells with a circular dimension, when a memory is pushed outside the East wall of the enclosure, it would necessarily be moved to the West wall of the enclosure if the period of the circular dimension was equal to the width of the enclosure. If this were true, then the partial grid field on one side of the enclosure would match up with its remainder on the other side. Such a situation would cause the animal to become completely confused regarding opposite sides of the enclosure (a location on the West wall would be indistinguishable from the corresponding location on the East wall). To reduce confusion between opposite sides of the enclosure, the width of the enclosure in which the animal navigated (Figure 5) was assumed to be half as wide as the full period of the border cells. In other words, although the space of possible representations was a three-torus, it was assumed that the real-world twodimensional enclosure encompassed a section of the torus (e.g., a square piece of tape stuck onto the surface of a donut). The torus is better thought of as “playing field” in which different sizes and shapes of enclosure can be represented (i.e., different sizes and shapes of tape placed on the donut). Furthermore, this assumption provides representational space that is outside the box without such locations wrapping around to the opposite side of the box.”

p.28 "More specifically, egocentric grid cells (e.g., head direction conjunctive grid cells) require stabilization of the place cell memories in the face of ongoing consolidation whereas allocentric grid cells reflect on-average place field positions." and p.32 "if place cells represent episodic memories, it seems natural that they should include head direction (an egocentric viewpoint)." But the head direction signal is not egocentric, it is allocentric. I'm unsure whether this is a typo or a potentially more serious conceptual misunderstanding.

Any reference to egocentric has been removed in this revision. In the initial submission, when I used egocentric, I was referring to memories that depended on the head direction of the animal at the time of memory formation. I was using “egocentric” in relation to whether the memory was related to the animal’s personal bodily experience at the time of memory formation. But I concede that this is confusing since the ego/allo distinction is typically used to differentiate angular directions that are relative to the person (left/right) versus earth (East/West). Instead, throughout the manuscript I now refer to these as view-dependent memories since head direction would entail having a different view of the environment at the time of memory formation. I still refer to the stacking of multiple view-dependent memories on the same X/Y location as being the development of an allocentric representation however, since this can be thought of as one way to learn a cognitive map of the enclosure that is view independent.

p.37 "But if the border cells had changed their alignment with the new enclosure (e.g., if the E border dimension aligned with the North-South borders), then the place cells would have appeared to undergo global remapping as their positions rotated by 90 degrees and the grid pattern would have also rotated." But this would not be interpreted as global remapping by standard analyses of place and grid cell responses. A coherent rotation of firing patterns is not interpreted as remapping.

This sentence now reads: “But if the border cells had changed their alignment with the new enclosure (e.g., if the E border dimension aligned with the North-South borders), then the place cells would remain in their same positions relative to the now-rotated borders (i.e., no remapping relative to the enclosure) and the corresponding grid cells would also retain their same alignment relative to the enclosure.”

p.37 "this is more accurately described as partial remapping (nearly all place fields were unaffected)." If nearly all place fields were unaffected, this should be interpreted as a stable map. Partial remapping is a mix of stability, rate remapping, and global remapping within a population of place cells.

This sentence has been removed.

p.40 "The dependence of grid cell responses on memory may help explain why grid cells have been found for bats crawling on a two-dimensional surface (Yartsev et al., 2011), but three-dimensional grid cells have never been observed for flying bats." This is not true. Ginosar et al. (2021) observed 3D grid cells in flying bats.

Thank you for highlighting this issue. In the initial submission I was using “grid cell” to mean a cell that produced a precise hexagonal grid, which is not the case for the 3D grid cells in bats. In this revision, I now discuss grid cell that produce irregular grid fields, writing:

“According to this model, hexagonally arranged grid cells should be the exception rather than the rule when considering more naturalistic environments. In a more ecologically valid situation, such as with landmarks, varied sounds, food sources, threats, and interactions with conspecifics, there may still be remembered locations were events occurred or remembered properties can be found, but because the non-spatial properties are non-uniform in the environment, the arrangement of memory feedback will be irregular, reflecting the varied nature of the environment. This may explain the finding that even in a situation where there are regular hexagonal grid cells, there are often irregular non-grid cells that have a reliable multi-location firing field, but the arrangement of the firing fields is irregular (Diehl et al., 2017). For instance, even when navigating in an enclosure that has uniform properties as dictated by experimental procedures, they may be other properties that were not well-controlled (e.g., a view of exterior lighting in some locations but not others), and these uncontrolled properties may produce an irregular grid (i.e., because the uncontrolled properties are reliably associated with some locations but not others, hippocampal memory feedback triggers retrieval of those properties in the associations locations).

In this memory model, there are other situations in which an irregular but reliable multi-location grid may occur, even when everything is well controlled. In the reported simulations, when the hippocampal place cells were based on variation in X/Y (as defined by Border cells), nothing else changed as a function of location, and the model rapidly produced a precise hexagonal arrangement of hippocampal place cell memories. When head direction was included (i.e., real-world variation in X, Y, and head direction), the model still produced a hexagonal arrangement as per face centered cubic packing of memories, but this precise arrangement was slower to emerge, with place cells continuing to shift their positions until the borders of the enclosure were sufficiently well learned from multiple viewpoints. If there is realworld variation in four or more dimensions, as is likely the case in a more ecologically valid situation, it will be even harder for place cell memories to settle on a precise regular lattice. Furthermore, in the case of four dimensions, mathematicians studying the “sphere packing problem” recently concluded that densest packing is irregular (Campos et al., 2023). This may explain why the multifield grid cells for freely flying bats have a systematic minimum distance between firing fields, but their arrangement is globally irregular (Ginosar et al., 2021). Assuming that the memories encoded by a bat include not just the three realworld dimensions of variation, but also head direction, the grid will likely be irregular even under optimal conditions of laboratory control.”

Multiple typos are found on page 25, end of paragraph 3: "More specifically, if there is one set of non-orthogonal dimensions for enclosure borders and the movement of one wall is too modest as to cause avoid global remapping, this would deform the grid modules based the enclosure border cells."

As detailed above in the response the public reviews, this paragraph has been rewritten.